# Understanding the Dynamics of Forgetting and Generalization in Continual Learning via the Neural Tangent Kernel

**Guodong Zheng**[1,2]\*, **Peng Wang**[3]\*, **Shengchao Hu**[4], **Quan Zheng**[1], **Li Shen**[2,3]†

[1] Huazhong University of Science and Technology
[2] Guangdong Laboratory of Artificial Intelligence and Digital Economy (SZ)
[3] Shenzhen Campus of Sun Yat-sen University
[4] Shanghai Jiao Tong University
`zhenggd943@gmail.com, mathshenli@gmail.com`

## Abstract

Continual learning (CL) enables models to acquire new tasks sequentially while retaining previously learned knowledge. However, most theoretical analyses focus on simplified, converged models or restrictive data distributions and therefore fail to capture how forgetting and generalization evolve during training in more general settings. Current theory faces two fundamental challenges: (i) analyses confined to the converged regime cannot characterize intermediate training dynamics; and (ii) establishing forgetting bounds requires two-sided bounds on the population risk for each task. To address these challenges, we analyze the training-time dynamics of forgetting and generalization in standard CL within the Neural Tangent Kernel (NTK) regime, showing that decreasing the loss's Lipschitz constant and minimizing the cross-task kernel jointly reduce forgetting and improve generalization. Specifically, we (i) characterize intermediate training stages via kernel gradient flow and (ii) employ Rademacher complexity to derive both upper and lower bounds on population risk. Building on these insights, we propose *OGD+*, which projects the current task's gradient onto the orthogonal complement of the subspace spanned by gradients of the most recent task evaluated on all prior samples. We further introduce *Orthogonal Penalized Gradient Descent* (OPGD), which augments OGD+ with gradient-norm penalization to jointly reduce forgetting and enhance generalization. Experiments on multiple benchmarks corroborate our theoretical predictions and demonstrate the effectiveness of OPGD, providing a principled pathway from theory to algorithm design in CL.

## 1 Introduction

Continual learning (CL) trains models on a sequence of tasks with the objective of maintaining strong performance across all of them. Unlike conventional training paradigms that operate on a fixed dataset or a single task, CL typically faces non-stationary data streams and complex task sequences. A major challenge in this setting is that models often experience a substantial performance drop on previously learned tasks when adapting to new ones. This phenomenon, known as *catastrophic forgetting* (McCloskey & Cohen, 1989; McClelland et al., 1995), arises when learning new information interferes with or overwrites prior knowledge.

Despite the considerable empirical success of numerous approaches in CL (Chaudhry et al., 2019a; Farajtabar et al., 2020; Dohare et al., 2024), rigorous theoretical understanding remains limited. Recent studies on the theory of forgetting and generalization error in CL have mainly focused on linear models and often assume restrictive data distributions, such as Gaussian distributions (Evron et al., 2022; Lin et al., 2023; Banayeeanzade et al., 2024; Li et al., 2025b). While the linear regime provides explicit characterizations of forgetting and generalization error, it is not suitable for more

---

\*Equal contribution
†Corresponding author

Table 1: Comparison of our results with Bennani et al. (2020), Doan et al. (2021) and Karakida & Akaho (2022). We summarize whether each work provides theoretical bounds on forgetting and generalization error, and whether it analyzes training dynamics or includes ridge regularization.

| Method | Forgetting Bound | Generalization Error | Training Dynamics | Ridge Regularization |
|---|---|---|---|---|
| Bennani et al. (2020) | ✗ | ✓ | ✗ | ✓ |
| Doan et al. (2021) | ✓ | ✗ | ✗ | ✓ |
| Karakida & Akaho (2022) | ✓ | ✗ | ✗ | ✗ |
| Ours | ✓ | ✓ | ✓ | ✗ |

general models or the non-stationary data streams typical in CL. In contrast, the Neural Tangent Kernel (NTK) regime (Jacot et al., 2018; Lee et al., 2019; Arora et al., 2019b) enables the analysis of more general models without being restricted to special distributions. However, existing NTK-based analyses (Bennani et al., 2020; Doan et al., 2021; Karakida & Akaho, 2022) primarily focus on converged models and therefore cannot characterize the behavior at intermediate training stages.

The main challenges in the theoretical analysis of both forgetting and generalization error are twofold: (1) existing analytical approaches primarily focus on the converged regime, either by obtaining explicit solutions from the loss function or by assuming asymptotic convergence as the number of iterations approaches infinity. However, the dynamics of forgetting and generalization error during training remain underexplored from a theoretical perspective. (2) bounding forgetting requires establishing both upper and lower bounds on the population loss for each task. In particular, while generalization bounds typically provide only an upper bound on the population loss, the forgetting metric involves averaging the discrepancy between the performance of the final model on previous tasks and that of the models trained directly on those tasks. Consequently, deriving two-sided bounds on the population loss in CL remains a significant challenge.

In this work, we theoretically analyze forgetting and generalization in vanilla CL under the NTK regime. We derive bounds at intermediate stages of training, revealing two key insights: (1) both forgetting and generalization error increase with iterations when the Lipschitz constant of the loss with respect to predictions exceeds a threshold, while reducing it consistently mitigates these effects; and (2) decreasing the magnitude of the cross-task kernel similarly alleviates forgetting and improves generalization. To support this analysis, we characterize model dynamics via kernel gradient flow and define forgetting in terms of population loss, deriving upper and lower bounds using Rademacher complexity. These tools allow us to rigorously study the evolution of forgetting and generalization at intermediate training stages. Building on these theoretical findings, we propose OGD+, which projects the current task's gradient onto the orthogonal complement of the subspace spanned by gradients from last task evaluated on all previous samples, reducing forgetting and improving generalization compared to standard OGD. We further show that controlling the Lipschitz constant in OGD+ enhances these benefits. Inspired by this, we introduce *Orthogonal Penalized Gradient Descent* (OPGD), which integrates gradient-norm penalization into OGD+ to jointly reduce forgetting and improve generalization. Finally, we empirically validate our theoretical predictions and demonstrate the effectiveness of OGD+ and OPGD on Permuted MNIST, Rotated MNIST, and Split CIFAR-100, providing a principled pathway from theory to algorithm design in CL.

Our main contributions can be summarized as follows:

- We derive bounds on forgetting and generalization error at intermediate stages of vanilla CL under the NTK regime, showing that both can be mitigated by reducing the Lipschitz constant or driving the cross-task kernel toward zero.
- We define forgetting and generalization error in terms of population loss, and provide upper and lower bounds via Rademacher complexity under the NTK regime. Furthermore, we leverage kernel gradient flow to analyze model dynamics at intermediate training stages.
- Building on these insights, we propose *OGD+*, which projects the current task's gradient onto directions orthogonal to the subspace formed by the last task's gradients on all previous samples. We further introduce *Orthogonal Penalized Gradient Descent* (OPGD), which adds gradient-norm penalization to OGD+ for tighter theoretical bounds.
- We empirically validate our theoretical predictions and demonstrate the effectiveness of OGD+ and OPGD on Permuted MNIST, Rotated MNIST, and Split CIFAR-100.

**Related Work.** Due to space constraints, a comprehensive literature review is provided in Appendix B. In particular, Table 1 presents a detailed comparison between our theoretical results and prior studies (Bennani et al., 2020; Doan et al., 2021; Karakida & Akaho, 2022).

## 2 PRELIMINARY

### 2.1 PROBLEM SETUP

We consider the standard CL setting with $T$ sequential training tasks. For any positive integer $n$, we denote $[n] := \{1, \ldots, n\}$. For each task $\tau \in [T]$, let $\mathcal{D}_\tau$ denote its data distribution, and let $S_\tau = \{X_\tau, Y_\tau\}$ be the corresponding training dataset drawn i.i.d. from $\mathcal{D}_\tau$. Here, $X_\tau = (x_\tau^1, \ldots, x_\tau^{n_\tau})^\top \in \mathbb{R}^{n_\tau \times d}$ is the feature matrix containing $n_\tau$ samples of $d$-dimensional feature vectors, and $Y_\tau \in \mathbb{R}^{n_\tau}$ is the associated label vector. The model trained on task $\tau$ is denoted by $f_\tau$, parameterized by $\theta_\tau$. Its state at iteration $t_\tau$ is written as $f_\tau^{t_\tau}$, and the final model after training is denoted by $f_\tau^*$, where training stops at iteration $t_\tau = t_\tau^*$. The model for task $\tau + 1$ is initialized from the final parameter of task $\tau$, that is, $\theta_{\tau+1}^0 = \theta_\tau^*$.

Let $\ell(f(x), y)$ be a loss function that quantifies the discrepancy between the model prediction $f(x)$ and the true label $y$ for a given sample $(x, y)$. We now introduce two fundamental notions in learning theory: the population loss and the empirical loss.

The population loss with respect to the distribution $\mathcal{D}_\tau$ is defined as:

$$L_{\mathcal{D}_\tau}(f) = \mathbb{E}_{(x_\tau, y_\tau) \sim \mathcal{D}_\tau}[\ell(f(x_\tau), y_\tau)]. \tag{1}$$

The empirical loss over a dataset $S_\tau = \{(x_\tau^i, y_\tau^i)\}_{i=1}^{n_\tau}$ i.i.d. drawn from $\mathcal{D}_\tau$ is defined as:

$$L_{S_\tau}(f) = \frac{1}{n_\tau} \sum_{i=1}^{n_\tau} \ell(f(x_\tau^i), y_\tau^i). \tag{2}$$

As shown in Lopez-Paz & Ranzato (2017); Lin et al. (2023), forgetting and overall generalization error can be defined in terms of the population loss as follows:

(1) *Forgetting*. This metric quantifies the degradation in performance on previously learned tasks after training on the current task at iteration $t_T$:

$$F_{t_T} = \frac{1}{T-1} \sum_{\tau=1}^{T-1} \left( L_{\mathcal{D}_\tau}(f_T^{t_T}) - L_{\mathcal{D}_\tau}(f_\tau^*) \right). \tag{3}$$

(2) *Overall generalization error*. This evaluates the model's generalization performance at iteration $t_T$ by averaging its population loss across all tasks:

$$G_{t_T} = \frac{1}{T} \sum_{\tau=1}^{T} L_{\mathcal{D}_\tau}(f_T^{t_T}). \tag{4}$$

Notably, Doan et al. (2021) define the forgetting metric on discrete datasets, while the definitions in Lin et al. (2023) are restricted to linear models. In contrast, our formulation applies to arbitrary function classes. Specifically, we extend the widely adopted metrics of backward transfer and average accuracy, which are commonly used to characterize forgetting and generalization in CL, by redefining them in terms of population loss. This reformulation enables these metrics to reflect model performance at the level of the underlying data distribution. Moreover, our definitions do not rely on specific model assumptions and evaluate performance over the entire input space, enabling a more comprehensive theoretical analysis of CL.

### 2.2 KERNEL REGIME FOR CONTINUAL LEARNING

We briefly review the basic concepts of the Neural Tangent Kernel (NTK) regime. Throughout this work, we assume that the model is trained using the mean squared error (MSE) loss. Accordingly, the empirical loss over task $\tau$ can be written as:

$$L_{S_\tau}(f) = \frac{1}{2n_\tau} \sum_{i=1}^{n_\tau} (f(x_\tau^i) - y_\tau^i)^2. \tag{5}$$

Before introducing the NTK, we define the neural network kernel (NNK) at training time $t$ as:

$$\hat{K}_\tau^t(x, x') = \langle \nabla_{\theta_\tau} f_\tau^t(x), \nabla_{\theta_\tau} f_\tau^t(x') \rangle \tag{6}$$

Based on this, the gradient flow dynamics of $f_\tau$ with respect to the MSE loss can be written as:

$$\frac{d}{dt} f_\tau^t(x) = -\frac{1}{n_\tau} \hat{K}_\tau^t(x, X_\tau)(f_\tau^t(X_\tau) - Y_\tau) \tag{7}$$

NTK theory states that, in the infinite-width limit, NNK $\hat{K}_\tau^t$ converges to a time-invariant kernel $K_\tau$, known as the NTK (Jacot et al., 2018; Lee et al., 2019; Arora et al., 2019b). In particular, if task $\tau$ is initialized from the trained parameters of task $\tau - 1$, i.e., $\theta_\tau^0 = \theta_{\tau-1}^*$, then each kernel entry satisfies

$$K_\tau(x, x') = \langle \nabla_{\theta_\tau^0} f_\tau^0(x), \nabla_{\theta_\tau^0} f_\tau^0(x') \rangle = \langle \nabla_{\theta_{\tau-1}^*} f_{\tau-1}^*(x), \nabla_{\theta_{\tau-1}^*} f_{\tau-1}^*(x') \rangle \tag{8}$$

For any $x \in \mathbb{R}^d$ and $X = (x_1, \ldots, x_n)^\top \in \mathbb{R}^{n \times d}$, we denote

$$K_\tau(x, X) = (K_\tau(x, x_1), \ldots, K_\tau(x, x_n)), \quad K_\tau(X, X) = [K_\tau(x_i, x_j)]_{n \times n} \tag{9}$$

Under the NTK regime, the training dynamics in Equation (7) reduce to the kernel gradient flow

$$\frac{d}{dt} f_\tau^t(x) = -\frac{1}{n_\tau} K_\tau(x, X_\tau)\big(f_\tau^t(X_\tau) - Y_\tau\big), \tag{10}$$

where $K_\tau$ remains fixed throughout the training process for task $\tau$. Hence, Equation (10) defines an ordinary differential equation (ODE) in the time variable $t$ with an initial condition induced by $\theta_\tau^0 = \theta_{\tau-1}^*$. Furthermore, Equation (10) admits a closed-form solution:

$$f_\tau^t(x) = f_{\tau-1}^*(x) - K_\tau(x, X_\tau)E_{\tau,t}K_\tau(X_\tau, X_\tau)^{-1}\big(f_{\tau-1}^*(X_\tau) - Y_\tau\big), \tag{11}$$

where $E_{\tau,t} := I - \exp\left(-\frac{t}{n_\tau} K_\tau(X_\tau, X_\tau)\right)$, and $f_{\tau-1}^*$ denotes the predictor obtained after completing the training of task $\tau - 1$. The derivation of Equation (11) follows directly from solving the linear ODE in Equation (10) and is provided in Appendix D.1. For notational convenience, we define

$$\tilde{f}_\tau^t(x) := K_\tau(x, X_\tau)E_{\tau,t}K_\tau(X_\tau, X_\tau)^{-1}\tilde{Y}_\tau, \tag{12}$$

where $\tilde{Y}_\tau := Y_\tau - f_{\tau-1}^*(X_\tau)$ represents the residual between the ground-truth labels of task $\tau$ and the predictions of the model trained on task $\tau - 1$. By recursively applying Equation (11) across $\tau$ tasks, the model for task $\tau$ can be expressed as $f_\tau^t(x) = \sum_{i=1}^{\tau-1} \tilde{f}_i^*(x) + \tilde{f}_\tau^t(x)$.

## 3 THEORETICAL RESULT

In this section, we present upper bounds on forgetting and generalization error for vanilla CL under the NTK regime in Theorem 1. For clarity of presentation, we adopt the notational convention $t_T^* = t_T$ for the final task $T$, i.e., the iteration index of task $T$ coincides with its stopping iteration.

**Theorem 1.** *Consider a sequence of $T$ tasks. For each task $\tau \in [T]$, let $\mathcal{D}_\tau$ denote the data distribution, and let $S_\tau = \{X_\tau, Y_\tau\}$ be the corresponding training dataset drawn i.i.d. from $\mathcal{D}_\tau$. Suppose the loss function $\ell(\cdot, \cdot)$ takes values in the interval $[0, c]$ and is $\rho$-Lipschitz in the first argument. Then, with probability at least $1 - \delta$, the following bounds hold:*

$$
\begin{aligned}
F_{t_T} \leq &\frac{1}{T-1} \sum_{\tau=1}^{T-1} \Bigg\{ 2\rho \sum_{k=1}^{T} \frac{\big[\operatorname{Tr}(K_k(X_\tau, X_\tau))\tilde{Y}_k^\top E_{k,t_k^*} K_k(X_k, X_k)^{-1}\tilde{Y}_k\big]^{1/2}}{n_\tau} \\
&+ 2\rho \sum_{k=1}^{\tau} \frac{\big[\operatorname{Tr}(K_k(X_\tau, X_\tau))\tilde{Y}_k^\top E_{k,t_k^*} K_k(X_k, X_k)^{-1}\tilde{Y}_k\big]^{1/2}}{n_\tau} + 6c\sqrt{\frac{log(2/\delta)}{2n_\tau}} \\
&+ \frac{T-\tau}{n_\tau} \sum_{k=\tau+1}^{T} \|K_k(X_\tau, X_k)E_{k,t_k^*} K_k(X_k, X_k)^{-1}\tilde{Y}_k\|^2 + \frac{1}{2n_\tau} \|e^{-\frac{t_\tau^*}{n_\tau} K_\tau(X_\tau, X_\tau)}\tilde{Y}_\tau\|^2 \Bigg\},
\end{aligned}
\tag{13}
$$

$$G_{t_T} \leq \frac{1}{T} \sum_{\tau=1}^{T} \left\{ \frac{T-\tau}{n_\tau} \sum_{k=\tau+1}^{T} \|K_k(X_\tau, X_k) E_{k, t_k^*} K_k(X_k, X_k)^{-1} \tilde{Y}_k\|^2 + \frac{1}{n_\tau} \|e^{-\frac{1}{n_\tau} K_\tau(X_\tau, X_\tau) t_\tau^*} \tilde{Y}_\tau\|^2 \right.$$
$$\left. + 2\rho \sum_{k=1}^{T} \frac{[\text{Tr}(K_k(X_\tau, X_\tau)) \tilde{Y}_k^\top E_{k, t_k^*} K_k(X_k, X_k)^{-1} \tilde{Y}_k]^{1/2}}{n_\tau} + 3c \sqrt{\frac{\log(2/\delta)}{2n_\tau}} \right\}.$$
$$(14)$$

To the best of our knowledge, Theorem 1 provides the first upper bounds on both forgetting and generalization error at intermediate training stages in vanilla CL. The proof is given in Appendix E. Furthermore, our bounds are explicitly dependent on the number of training iterations, allowing us to characterize the evolution of forgetting and generalization errors throughout the training process.

To facilitate the subsequent analysis of the upper bounds of forgetting and generalization errors, we denote $F_{t_T}^{\text{upper}}$ and $G_{t_T}^{\text{upper}}$ as the respective upper bounds of $F_{t_T}$ and $G_{t_T}$ in Theorem 1. Based on Theorem 1, we will provide insights on the following two aspects.

(1) *Lipschitz constant.* The Lipschitz constant $\rho$ characterizes the maximum rate of change of the loss with respect to the model's *predictions*. Formally, for any two predictions $u, v$ in the output space, $|\ell(u, y) - \ell(v, y)| \leq \rho \|u - v\|$. In general, a smaller prediction-Lipschitz constant implies that the loss varies more smoothly with respect to the model outputs (i.e., a flatter landscape in prediction space), which is often associated with improved generalization.

(2) *Cross-task kernel.* Under the NTK regime, we define the cross-task kernel between any two tasks $\tau < k \in [2, T]$ as $K_k(X_\tau, X_k)$. In traditional machine learning, cross kernels characterize the similarity between two datasets (Akaho, 2006; Schölkopf et al., 1997). In the NTK setting, the cross-task kernel instead captures cross-task interactions by measuring the alignment between the model gradients with respect to different task datasets. A larger norm of this matrix indicates stronger task interference, which in turn increases the risk of forgetting and generalization error.

To gain deeper insight into continual learning, we next analyze the effect of these two critical factors.

## 3.1 THE IMPACT OF THE LIPSCHITZ CONSTANT

The role of the Lipschitz constant has been extensively studied in non-CL settings, where smaller values are often linked to improved generalization performance (Bartlett et al., 2017; Miyato et al., 2018; Zhao et al., 2022; Khromov & Singh, 2024). One widely used approach to approximately reduce the Lipschitz constant is to penalize the gradient norm (PGN) of the loss function (Zhao et al., 2022). Moreover, Gradient-norm Aware Minimization (GAM) further penalizes the gradient norm within a neighborhood of the parameters (Zhang et al., 2023), thereby promoting flatter solutions. Although reducing the Lipschitz constant has been both theoretically and empirically shown to improve performance in non-CL scenarios, its effectiveness in CL remains largely unexplored.

> **Q1:** Does the role of the Lipschitz constant in non-CL also hold in the context of CL?

To address **Q1**, we examine how the Lipschitz constant $\rho$ affects the forgetting bound $F_{t_T}^{\text{upper}}$ and the generalization error bound $G_{t_T}^{\text{upper}}$ in Theorem 1. In particular, we analyze the evolution of $F_{t_T}^{\text{upper}}$ and $G_{t_T}^{\text{upper}}$ with respect to the training iteration $t_T$ when the Lipschitz constant exceeds a certain threshold. The detailed proof of Lemma 1 is provided in Section G.1.

**Lemma 1.** *For any fixed $t_T$, a smaller Lipschitz constant $\rho$ leads to smaller values of both $G_{t_T}^{upper}$ and $F_{t_T}^{upper}$. Moreover, there exists a constant $\rho^* > 0$ such that, for all $\rho > \rho^*$, both $G_{t_T}^{upper}$ and $F_{t_T}^{upper}$ increase monotonically with respect to $t_T$.*

**Remark 1.** *Lemma 1 indicates that reducing the Lipschitz constant $\rho$ consistently mitigates both forgetting and generalization error in CL. In contrast to non-CL settings, where reducing the Lipschitz constant $\rho$ primarily improves generalization, in CL it also alleviates forgetting.*

**Remark 2.** *Lemma 1 further implies that, once $\rho$ exceeds a threshold $\rho^*$, the upper bounds $G_{t_T}^{\text{upper}}$ and $F_{t_T}^{\text{upper}}$ increase monotonically with the number of training iterations $t_T$. In the degenerate limit*

*of skipping updates (i.e., $t_T = 0$), these quantities can be made trivially small, but at the cost of no adaptation to the new task—an undesirable, pathological outcome. Therefore, it is crucial to design mechanisms that explicitly control or reduce $\rho$ for each task to improve CL performance in practice.*

As shown above, reducing the Lipschitz constant $\rho$ is beneficial for mitigating forgetting and improving generalization in CL. To implement this in practice, we adopt a penalized gradient-norm framework that approximately reduces $\rho$; further details are provided in Appendix G.2. In the CL setting, the training loss of PGN for any task $\tau \in [T]$ is given by

$$L_{S_\tau}^{PGN}(\theta_\tau) = L_{S_\tau}(\theta_\tau) + \alpha_\tau \|\nabla_{\theta_\tau} L_{S_\tau}(\theta_\tau)\|_2, \tag{15}$$

where $\|\cdot\|_2$ denotes the Euclidean norm and $\alpha_\tau$ is the penalty coefficient.

In practice, we employ GAM rather than PGN, since GAM encourages flatter solutions. In the CL setting, the training loss of GAM for task $\tau$ is defined as

$$L_{S_\tau}^{GAM}(\theta_\tau) = L_{S_\tau}(\theta_\tau) + \alpha_\tau b_\tau \max_{\theta_\tau' \in B(\theta_\tau, b_\tau)} \|\nabla_{\theta_\tau'} L_{S_\tau}(\theta_\tau')\|_2, \tag{16}$$

where the perturbation radius $b_\tau$ controls the neighborhood size, and $B(\theta_\tau, b_\tau)$ denotes the open ball of radius $b_\tau$ centered at $\theta_\tau$ in Euclidean space. Importantly, GAM penalizes the neighborhood Lipschitz constant, thereby avoiding sharp minima and improving robustness. Furthermore, our experimental results in Table 2 empirically demonstrate that GAM effectively mitigates forgetting and enhances generalization compared to vanilla CL (SGD), thereby validating our theoretical analysis.

## 3.2 THE IMPACT OF CROSS-TASK KERNEL

In this section, we examine the influence of the cross-task kernel between any two tasks $\tau < k \in [2, T]$, i.e., $K_k(X_\tau, X_k)$, on the forgetting bound $F_{t_T}^{\text{upper}}$ and the generalization error bound $G_{t_T}^{\text{upper}}$ in Theorem 1. Each entry of $K_k(X_\tau, X_k)$ is the inner product between the model gradients with respect to a sample from $X_\tau$ and a sample from $X_k$. Ideally, to eliminate the adverse effect of $K_k(X_\tau, X_k)$ on $F_{t_T}^{\text{upper}}$ and $G_{t_T}^{\text{upper}}$, all entries should be zero—equivalently, the gradients with respect to different datasets should be mutually orthogonal. A natural and effective approach to enforce such orthogonality is Orthogonal Gradient Descent (OGD) (Farajtabar et al., 2020). In the following, we analyze the behavior of the cross-task kernel to provide a theoretical explanation of how OGD mitigates forgetting and reduces generalization error.

We introduce OGD in the context of CL. For any $\tau \in [T]$, define $v_{\tau,i} := \nabla_\theta f_\tau^*(x_\tau^i)$ and $\mathbb{E}_\tau := \text{span}\{v_{\tau,i}\}_{i=1}^{n_\tau}$, the subspace spanned by the parameter gradients of the converged model $f_\tau^*$ evaluated on the inputs from task $\tau$. The core idea of OGD is to project the gradient of the current task onto the orthogonal complement of the subspaces spanned by all previous tasks, i.e., $\mathbb{E}_1 \oplus \cdots \oplus \mathbb{E}_{\tau-1}$. Let $P_{(\mathbb{E}_1 \oplus \cdots \oplus \mathbb{E}_{\tau-1})^\perp}$ denote the projection operator onto the orthogonal complement of this space, which we write simply as $P_\tau$ for brevity. Under OGD, the gradient flow dynamics for task $\tau$ are given by

$$\frac{d}{dt} f_\tau^t(x) = -\frac{1}{n_\tau} \tilde{K}_\tau(x, X_\tau)\big(f_\tau^t(X_\tau) - Y_\tau\big), \tag{17}$$

where $\tilde{K}_\tau(x, x') = \langle P_\tau \nabla_{\theta_{\tau-1}^*} f_{\tau-1}^*(x), P_\tau \nabla_{\theta_{\tau-1}^*} f_{\tau-1}^*(x') \rangle$. Thus, we obtain a gradient flow analogous to the standard SGD gradient flow in Equation (10), with the key difference lying in the form of the kernel. Additional details are provided in Appendix D.2.

In the following, we demonstrate that OGD reduces the cross-task kernel between the datasets of two adjacent tasks to the zero matrix. The proof of Lemma 2 is provided in Appendix G.3.

**Lemma 2.** *For any $k \in [2, T]$, under OGD we have $\tilde{K}_k(X_{k-1}, X_k) = 0$.*

**Remark 3.** *In Lemma 2, we show that OGD can eliminate the cross-task kernel between two adjacent tasks, thereby yielding tighter bounds on forgetting and generalization error compared to standard SGD. Moreover, we observe that the orthogonality constraints in standard OGD are unnecessarily strong. Specially, if the projector $P_k$ is redefined onto the orthogonal complement of $\mathbb{E}_{k-1}$, rather than $\mathbb{E}_1 \oplus \cdots \oplus \mathbb{E}_{k-1}$, Lemma 2 still holds while avoiding overly restrictive constraints (see Appendix G.3 for details). Empirically, as shown in Table 2, OGD achieves better performance than SGD.*

## 4 OGD+ AND OPGD ALGORITHMS

### 4.1 REFINED ORTHOGONAL GRADIENT DESCENT (OGD+)

In Lemma 2, we theoretically show that OGD exhibits less forgetting and better generalization than SGD in CL by proving that the cross-task kernel between two adjacent tasks is the zero matrix under OGD. Moreover, if this property could be extended so that the cross-task kernel between *any* pair of tasks were zero, CL performance should improve further. This naturally raises the following question:

> **Q2:** How can OGD be improved to eliminate the cross-task kernel between arbitrary task pairs, thereby further reducing forgetting and enhancing generalization?

To address **Q2**, for any $k \in [T]$, we redefine the gradient subspace as $\mathbb{E}'_k := \text{span}\{\nabla_\theta f^*_k(x^m_l) \mid l \in [k], m \in [n_l]\}$ and the projection operator as $P'_k := P_{\mathbb{E}'_{k-1}}^\perp$. We refer to this refined variant of OGD as *OGD+*. Accordingly, the corresponding NTK under OGD+ can be reformulated as

$$\hat{K}_\tau(x, x') = \langle P'_\tau \nabla_{\theta^*_{\tau-1}} f^*_{\tau-1}(x), \, P'_\tau \nabla_{\theta^*_{\tau-1}} f^*_{\tau-1}(x') \rangle. \tag{18}$$

As shown in Lemma 3, OGD+ reduces the cross-task kernel between the datasets of any two tasks to the zero matrix. The detailed proof is provided in Appendix G.4.

**Lemma 3.** *For any $\tau < k \in [2, T]$, under OGD+ we have $\hat{K}_k(X_\tau, X_k) = 0$.*

**Remark 4.** *Lemma 3 demonstrates that OGD+ can eliminate the cross-task kernel between any pair of tasks, thereby yielding lower forgetting and better generalization compared to standard OGD. In particular, we derive upper bounds on both forgetting and generalization error for OGD and OGD+ (see Appendix G.5), and establish that both bounds for OGD+ are strictly tighter.*

**Comparison between OGD and OGD+.** The key difference between OGD and OGD+ lies in how gradient information is stored and released. Specifically, OGD stores the gradients of the model after training on the current task using only the data from that task, and these gradients are retained indefinitely. In contrast, OGD+ stores the gradients of the model after training on the current task using all data from previous tasks, but releases them once training on the subsequent task is completed. As shown in Remark 4, OGD+ provides stronger theoretical guarantees than OGD due to its stricter enforcement of gradient orthogonality. Empirically, Table 2 shows that OGD+ forgets less and generalizes better than OGD on the two MNIST benchmarks, with pronounced improvements in both metrics. However, on Split CIFAR-100, OGD+ slightly underperforms OGD on both metrics, which we attribute to its excessive orthogonality. In particular, overly restrictive orthogonality reduces the feasible gradient subspace, thereby limiting the model's capacity to adequately fit the current task—especially under large distribution shifts between tasks. We next explore strategies to mitigate the negative impact of excessive orthogonality in OGD+.

### 4.2 ORTHOGONAL PENALIZED GRADIENT DESCENT (OPGD)

In Section 4.1, we theoretically demonstrated that OGD+ achieves lower forgetting and better generalization than OGD. However, while OGD+ enhances gradient orthogonality across tasks, it neglects inter-task performance and thus risks reducing plasticity in practice. A straightforward way to enhance inter-task performance is to reduce the Lipschitz constant of each task. Furthermore, in Section 3.1, we theoretically established that reducing the Lipschitz constant consistently mitigates forgetting and improves generalization. These observations naturally motivate the following question:

> **Q3:** Can reducing the Lipschitz constant in OGD+ further mitigate forgetting and enhance generalization compared to standard OGD+?

We denote $F^{\text{upper+}}_{t_T}$ and $G^{\text{upper+}}_{t_T}$ as the upper bounds of forgetting and generalization error for OGD+. To address **Q3**, we analyze how these bounds vary as the Lipschitz constant is reduced, and further examine their dependence on $t_T$ when the Lipschitz constant falls below a certain threshold. The formal results are stated in Lemma 4, with proofs provided in Appendix G.6.

**Lemma 4.** *For any fixed $t_T$, reducing the Lipschitz constant $\rho$ leads to strictly smaller values of $G_{t_T}^{\mathrm{upper}+}$ and $F_{t_T}^{\mathrm{upper}+}$ compared to their original values under OGD+. Moreover, there exists a constant $\rho' > 0$ such that, for all $\rho < \rho'$, $G_{t_T}^{\mathrm{upper}+}$ decreases monotonically with respect to $t_T$, while $F_{t_T}^{\mathrm{upper}+}$ increases monotonically with respect to $t_T$.*

**Remark 5.** *Reducing the Lipschitz constant in OGD+ yields tighter bounds on both forgetting and generalization error than standard OGD+. This indicates that incorporating mechanisms to reduce the Lipschitz constant within OGD+ can further mitigate forgetting while improving generalization.*

**Remark 6.** *Lemma 4 further implies that when the Lipschitz constant in OGD+ falls below a certain threshold, it helps avoid the degenerate phenomenon discussed in Remark 1, thereby benefiting generalization as training progresses. At the same time, longer training (larger $t_T$) increases the risk of catastrophic forgetting because $F_{t_T}^{\mathrm{upper}}$ grows with $t_T$, consistent with the behavior of large Lipschitz constants noted in Remark 1. This highlights a trade-off between mitigating forgetting and improving generalization. Importantly, this observation does not conflict with Remark 5: although extended training may increase forgetting, for any fixed iteration the bounds on both forgetting and generalization remain tighter when the Lipschitz constant is reduced.*

**OPGD algorithm:** Leveraging Remark 5 and Remark 6, we establish a principled pathway from theory to algorithm design: integrating OGD+, which enforces cross-task orthogonality, with GAM, which reduces the Lipschitz constant. This unified approach, termed *Orthogonal Penalized Gradient Descent* (OPGD), jointly mitigates forgetting and enhances generalization. As shown in Table 2, OPGD achieves substantial improvements over OGD+.

Next, we present the details of OPGD. For the first task, we update the model parameters of $f_1$ by minimizing the GAM loss (Equation (16)), which effectively reduces the Lipschitz constant of the loss and thereby enhances inter-task performance. The corresponding gradients are then stored. For each subsequent task $\tau \in [2, T]$, at each parameter update iteration we first minimize the GAM loss for $f_\tau$, and then apply OGD+ to the resulting (penalized) gradient, ensuring that the gradient for task $\tau$ is orthogonal to the stored gradients from task $\tau-1$. Finally, we release the stored gradients of task $\tau-1$ and replace them with the gradients of task $\tau$ evaluated on samples from all previous tasks. The full procedure of OPGD is summarized in Algorithm 1.

---

**Algorithm 1:** OPGD

**Input :** Task sequence $T_1, T_2, \ldots$; learning rate $\eta$; balance coefficient $\alpha$; perturbation radius $b$; small constant $\xi$.

$S \leftarrow \varnothing, \mathcal{M} \leftarrow \varnothing, \theta \leftarrow \theta_0$;
**for** *Task ID $\tau = 1, 2, \ldots$* **do**
  **repeat**
    `// GAM`
    $g_1 \leftarrow \nabla_\theta L_{S_\tau}(\theta)$;
    $f \leftarrow \nabla_\theta^2 L_{S_\tau}(\theta) \frac{\nabla_\theta L_{S_\tau}(\theta)}{\|\nabla_\theta L_{S_\tau}(\theta)\| + \xi}$;
    $\theta' \leftarrow \theta + b \cdot \frac{f}{\|f\| + \xi}$;
    $g_2 \leftarrow b \cdot \nabla_{\theta'}^2 L_{S_\tau}(\theta') \frac{\nabla_{\theta'} L_{S_\tau}(\theta')}{\|\nabla_{\theta'} L_{S_\tau}(\theta')\| + \xi}$;
    $g \leftarrow (1 - \alpha) g_1 + \alpha g_2$;
    `// Orthogonal updates`
    $g \leftarrow g - \sum_{v \in S_\tau} \mathrm{proj}_v(g)$;
    $\theta \leftarrow \theta - \eta g$;
  **until** *convergence*;
  $S \leftarrow \varnothing$;
  **for** $(x, y) \in S_\tau \cup \mathcal{M}$ *and $k \in [1, c]$ with $y_k = 1$* **do**
    $u \leftarrow \nabla_\theta f_\tau(x) - \sum_{v \in S} \mathrm{proj}_v(\nabla_\theta f_\tau(x))$;
    $S \leftarrow S \cup \{u\}$;
  **end**
  Sample $M_\tau \subset S_\tau, \quad \mathcal{M} \leftarrow \mathcal{M} \cup M_\tau$;
**end**

---

## 5 EXPERIMENT

In this section, we present extensive experiments to validate our theoretical findings and demonstrate the effectiveness of OGD+ and OPGD. Additional implementation details, further comparisons with baselines, and ablation studies are provided in Appendix C.

**Datasets.** We evaluate our approach on three widely used CL benchmarks: Permuted MNIST (Kirkpatrick et al., 2017), Rotated MNIST (Farajtabar et al., 2020), and Split CIFAR-100 (Chaudhry et al., 2019a). Permuted MNIST and Rotated MNIST are variants of the original MNIST dataset, where each task is defined by a random pixel permutation or a rotation, respectively. For both benchmarks, we construct 15 sequential tasks using different permutations or rotation angles. Split CIFAR-100 is created by partitioning the 100 classes of CIFAR-100 into 20 disjoint tasks, each containing 5 classes.

Table 2: Average accuracy (ACC) and backward transfer (BWT) over all tasks on different datasets. Higher ACC and BWT indicate better generalization and less forgetting. All results are reproduced by us and averaged over 5 runs. The best continual learning results are highlighted in **bold**.

| Dataset | **Permuted MNIST** (15 tasks) | | **Rotated MNIST** (15 tasks) | | **Split CIFAR-100** (20 tasks) | |
| Method | ACC | BWT | ACC | BWT | ACC | BWT |
| --- | --- | --- | --- | --- | --- | --- |
| SGD | $70.29 \pm 1.50$ | $-25.33 \pm 1.57$ | $68.79 \pm 0.43$ | $-28.09 \pm 0.45$ | $52.08 \pm 0.81$ | $-30.63 \pm 1.31$ |
| GAM | $72.61 \pm 1.44$ | $-22.47 \pm 1.57$ | $72.85 \pm 0.44$ | $-20.60 \pm 0.47$ | $61.70 \pm 1.68$ | $-22.63 \pm 1.60$ |
| OGD | $82.17 \pm 0.64$ | $-12.38 \pm 0.66$ | $77.52 \pm 0.69$ | $-18.43 \pm 0.76$ | $63.91 \pm 1.62$ | $-20.57 \pm 1.66$ |
| **OGD+** | $86.22 \pm 0.62$ | $-8.11 \pm 0.62$ | $86.15 \pm 0.49$ | $-9.02 \pm 0.56$ | $61.84 \pm 2.51$ | $-23.47 \pm 2.48$ |
| **OPGD** | $\mathbf{86.27 \pm 0.56}$ | $\mathbf{-7.73 \pm 0.61}$ | $\mathbf{89.15 \pm 0.22}$ | $\mathbf{-3.69 \pm 0.27}$ | $\mathbf{68.17 \pm 0.71}$ | $\mathbf{-12.58 \pm 1.35}$ |

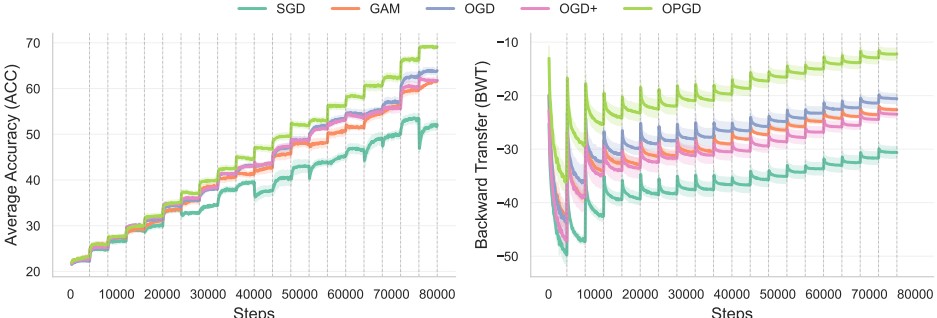

Figure 1: Dynamics of average accuracy (ACC) and backward transfer (BWT) for different methods on Split CIFAR-100. Vertical dotted lines are used to indicate the boundaries between different tasks. For each task, we record ACC and BWT at evenly spaced intervals, performing 40 evaluations per task by measuring performance every 100 training iterations.

**Baselines.** To align with our theoretical analysis, we compare OGD+ and OPGD with three continual learning methods: vanilla CL with SGD, GAM (Zhang et al., 2023), and OGD (Farajtabar et al., 2020). Additional comparisons with other CL methods are provided in Appendix C.3.

**Evaluation metrics.** To align with our theoretical analysis, we adopt *average accuracy* (ACC) and *backward transfer* (BWT) as the evaluation metrics (Lopez-Paz & Ranzato, 2017). Formally, they are defined as

$$ACC = \frac{1}{T} \sum_{i=1}^{T} A_{T,i}, \quad BWT = \frac{1}{T-1} \sum_{i=1}^{T-1} A_{T,i} - A_{i,i},$$

where $A_{t,i}$ denotes the accuracy of the model on task $i$ after completing training on task $t$, and $T$ is the total number of tasks.

**Performance.** As shown in Table 2, OPGD achieves significant improvements in both ACC and BWT over prior methods across all datasets, corroborating our theoretical claim that reducing the Lipschitz constant within OGD+ simultaneously mitigates forgetting and improves generalization. In particular, OPGD yields average relative gains of $+4.59\%$ in ACC and $+36.73\%$ in BWT across three benchmarks. Furthermore, OGD+ consistently forgets less and generalizes better than OGD on Permuted MNIST and Rotated MNIST, with average relative gains of $+4.27\%$ in ACC and $+23.82\%$ in BWT across three benchmarks. However, on Split CIFAR-100—whose distribution is substantially more complex than Permuted MNIST and Rotated MNIST—OGD+ underperforms OGD. We attribute this to excessive orthogonality in OGD+, which reduces model plasticity and consequently degrades inter-task performance, as discussed in Section 4.1; similar observations have been reported by Zhao et al. (2023); Yang et al. (2023). Notably, OPGD mitigates this effect by reducing the Lipschitz constant within OGD+, thereby enhancing inter-task performance.

**Dynamics of forgetting and generalization.** As shown in Figure 1, the ACC of OPGD increases steadily with the number of iterations, indicating that longer training enhances generalization. This

result is consistent with our theoretical analysis in Lemma 4. In contrast, the ACC of SGD does not consistently improve and even declines in the final tasks. This phenomenon aligns with Remark 1, which suggests that without explicit control of the Lipschitz constant, prolonged training may accumulate instability and hinder generalization. Notably, incorporating GAM to reduce the Lipschitz constant helps SGD avoid this degradation, enabling more stable generalization. As shown in Figure 1, the BWT of OPGD decreases within each task interval, indicating that additional iterations increase forgetting—again consistent with Lemma 4. Taken together, these results highlight a fundamental trade-off between forgetting and generalization: *while longer training improves generalization, it simultaneously exacerbates forgetting*, in line with Remark 6. Despite this trade-off, OPGD consistently outperforms competing methods in terms of both ACC and BWT throughout training.

## 6 CONCLUSION

We derived upper bounds on forgetting and generalization error at intermediate training stages in CL under the NTK regime. Our analysis shows that reducing the Lipschitz constant and enforcing gradient orthogonality both help mitigate forgetting and improve generalization. Building on these insights, we proposed OGD+ and OPGD, which refine gradient orthogonality and integrate gradient-norm penalization, respectively. Empirical results on standard benchmarks corroborate our theoretical predictions, providing a principled pathway from theory to algorithm design in CL. We discuss limitations and our use of large language models in Appendix A.

## ACKNOWLEDGMENT

This work is supported by NSFC Grant (No. 62576364), Shenzhen Basic Research Project (Natural Science Foundation) Basic Research Key Project (NO. JCYJ20241202124430041), the Open Research Fund from Guangdong Laboratory of Artificial Intelligence and Digital Economy (SZ) (NO. GML-KF-24-23), CCF-DiDi GAIA Collaborative Research Funds (NO. CCF-DiDi GAIA 202508).

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

# A  ADDITIONAL STATEMENT

**Limitations.**   Our theoretical analysis is developed under the NTK regime, which may not fully reflect the behavior of practical deep networks with finite width or more complex architectures. In addition, our empirical evaluation is limited to classification benchmarks; extending both the theoretical framework and the proposed algorithms to other modalities and learning paradigms remains an important direction for future work.

**The Use of Large Language Models.**   In this work, we exclusively employ large language models (LLMs) to refine the writing and presentation of our manuscript.

# B  RELATED WORKS

**Empirical studies in CL.** Continual learning has achieved substantial empirical progress, with existing approaches broadly categorized into three families: (1) *Regularization-based methods*, which introduce explicit penalty terms to restrict updates on parameters important for previous tasks (Kirkpatrick et al., 2017; Zenke et al., 2017), or employ knowledge distillation by aligning the predictions of the current model (student) with those of the previous model (teacher) to mitigate forgetting (Li & Hoiem, 2017; Dhar et al., 2019; Fostiropoulos et al., 2023); (2) *Replay-based methods*, which either store and replay data from past tasks during training on new tasks (Chaudhry et al., 2019a; Riemer et al., 2018; Buzzega et al., 2020), or retain gradient information from prior tasks and enforce new updates to be orthogonal to past gradients, thereby avoiding explicit data replay (Farajtabar et al., 2020; Saha et al., 2021; Lin et al., 2022); (3) *Architecture-based methods* (Rusu et al., 2016; Yoon et al., 2018; Wang et al., 2022), which dynamically expand or adapt the network architecture to preserve knowledge from earlier tasks.

**Theoretical analysis of CL.** Recent works have begun to lay the theoretical foundations of CL by analyzing forgetting and generalization error under simplified settings. Several studies (Evron et al., 2022; Lin et al., 2023; Banayeeanzade et al., 2024; Zheng et al., 2025; Li et al., 2025b) investigate these phenomena within overparameterized linear models, typically assuming that datasets are drawn from Gaussian distributions. Ding et al. (2024) study forgetting in both overparameterized and underparameterized regimes, while relaxing the data assumptions to fourth-moment conditions. Recently, there has been growing interest in the theoretical analysis of regularization based methods for CL. Heckel (2022) theoretically characterize how the performance of a model in a contrastive CL framework is controlled by the training losses on previous tasks. Li et al. (2023) derive bounds on the average risk over two tasks for an $\ell_2$-regularized CL algorithm. Zhao et al. (2024) provide a statistical analysis of regularization based CL on a sequence of linear regression tasks and highlight how different regularization terms affect model performance. Li et al. (2025a) establish upper and lower bounds on the joint excess risk for a generalized $\ell_2$-regularized CL algorithm. Another major line of theoretical work is based on the NTK regime (Bennani et al., 2020; Doan et al., 2021; Karakida & Akaho, 2022). In particular, Bennani et al. (2020) established generalization error bounds for CL via Rademacher complexity; Doan et al. (2021) analyzed forgetting by introducing the NTK overlap matrix as a task-similarity metric; and Karakida & Akaho (2022), leveraging learning-curve analysis between two tasks (Bordelon et al., 2020), studied both forgetting and generalization error under the assumption that datasets from different tasks are drawn from the same distribution. Although these works provide valuable insights, most rely on simplified settings and converged models, thereby overlooking the evolution of models during training in realistic CL scenarios. In contrast, we theoretically characterize the evolution of both forgetting and generalization error in vanilla CL under the NTK regime, without requiring such restrictive assumptions.

The most relevant works to ours are Bennani et al. (2020); Doan et al. (2021), which theoretically analyze generalization error and forgetting separately under the NTK regime. However, our study differs in several key aspects: (1) Their analyses incorporate a ridge regularization term in the loss function, effectively aligning with regularization-based CL methods. By contrast, our framework makes no such assumption and corresponds to *vanilla* CL, thus serving as a clean theoretical *baseline* against which improved CL methods—e.g., buffer-based and regularization-based approaches—can be directly compared. (2) Their focus lies primarily on the performance of converged models, whereas we characterize the full evolution of forgetting and generalization error throughout the training dynamics. (3) Bennani et al. (2020) study the generalization error of OGD in CL by leveraging the property that model outputs remain consistent between consecutive tasks, while Doan et al. (2021) analyze forgetting in OGD through the NTK overlap matrix. In contrast, we analyze *both* forgetting and generalization error in OGD through the structure of the cross-task kernel.

# C  ADDITIONAL EXPERIMENTAL RESULTS

## C.1  EXPERIMENTAL SETTING

In this section, we provide additional details of our experimental setup, present extended comparisons with more baselines (Section C.3), and conduct an ablation study of OPGD (Section C.4). All experiments are conducted using the NVIDIA RTX 4090 with 24GB GPU memory, CUDA v11.8 and cuDNN v8.7.0 in PyTorch v2.4.1.

**Architecture.** For Permuted MNIST and Rotated MNIST, we adopt a three-layer multilayer perceptron (MLP) with two hidden layers of 100 units each and a final layer with 10 output logits. All layers except the last one use ReLU activation. For Split CIFAR-100, we employ a LeNet architecture for training. Table 3 summarizes the hyperparameter settings used for OPGD.

Table 3: Hyperparameter Settings of OPGD

| Hyperparameter | Permuted MNIST | Rotated MNIST | Split CIFAR-100 |
|---|---|---|---|
| Task nums | 15 | 15 | 20 |
| Network | MLP | MLP | LeNet |
| Epochs | 5 | 5 | 50 |
| Learning rate | $10^{-3}$ | $10^{-3}$ | $10^{-3}$ |
| Batch size | 32 | 32 | 32 |
| Hidden dimension | 100 | 100 | 100 |
| Balance coefficient $\alpha$ | 0.6 | 0.1 | 0.8 |
| Perturbation radius $b$ | 0.02 | 0.2 | 0.05 |
| Sampling size $m$ | 100 | 100 | 100 |

## C.2  ADDITIONAL EXPERIMENTS

To verify that our methods also apply to other types of tasks, we conduct additional experiments on online continual learning. In online continual learning, data arrive sequentially in the form of a stream. Whenever a small batch of data arrives, it is used for a single training iteration (Aljundi et al., 2019b;a). Following the standard online continual learning protocol, we use a batch size of 10 and train for one epoch. We perform experiments on Permuted MNIST, Rotated MNIST, and Split CIFAR-100. We adopt the same hyperparameter settings as those listed in Table 3.

Table 4: Average accuracy (ACC) and backward transfer (BWT) over all tasks on online CL benchmarks. All results are reproduced by us and averaged over 5 runs. The best continual learning performance is highlighted in **bold**.

| Dataset | Permuted MNIST (15 tasks) | | Rotated MNIST (15 tasks) | | Split CIFAR-100 (20 tasks) | |
|---|---|---|---|---|---|---|
| Method | ACC | BWT | ACC | BWT | ACC | BWT |
| SGD | $75.85 \pm 0.78$ | $-12.24 \pm 0.81$ | $67.23 \pm 0.41$ | $-23.62 \pm 0.40$ | $31.90 \pm 1.57$ | $-16.21 \pm 1.42$ |
| GAM | $76.67 \pm 1.02$ | $-10.95 \pm 1.10$ | $70.63 \pm 0.59$ | $-14.86 \pm 0.67$ | $33.56 \pm 1.67$ | $-14.29 \pm 1.65$ |
| OGD | $78.76 \pm 0.50$ | $-8.65 \pm 0.49$ | $79.17 \pm 0.41$ | $-9.53 \pm 0.48$ | $37.66 \pm 1.78$ | $-7.92 \pm 1.66$ |
| **OGD+** | $81.88 \pm 0.33$ | $-5.23 \pm 0.38$ | $86.35 \pm 0.17$ | $-1.02 \pm 0.28$ | $37.68 \pm 1.77$ | $-7.88 \pm 1.68$ |
| **OPGD** | $\mathbf{82.62 \pm 0.51}$ | $\mathbf{-4.93 \pm 0.49}$ | $\mathbf{87.14 \pm 0.22}$ | $\mathbf{2.75 \pm 0.25}$ | $\mathbf{39.88 \pm 1.49}$ | $\mathbf{-6.12 \pm 1.45}$ |

To verify that our theoretical findings can benefit other continual learning algorithms, we integrate GAM or OPGD into both rehearsal based and regularization based methods on the Permuted MNIST dataset. Concretely, for rehearsal based baselines we adopt a naive rehearsal protocol that randomly selects 100 samples from each past task. For regularization based baselines we use an $\ell_2$ regularizer so that the optimization problem for task $\tau$ takes the form

$$\arg\min_{\theta_\tau} \ \ell(\theta_\tau) + \|\theta_\tau - \theta_{\tau-1}\|_2^2.$$

The corresponding results are presented in Table 5. The experimental results show that our theoretical messages also benefit other continual learning algorithms.

Table 5: Average accuracy (ACC) and backward transfer (BWT) over all tasks on Permuted MNIST. All results are reproduced by us and averaged over 5 runs. The best continual learning results are highlighted in **bold**.

| Method | ACC | BWT |
|---|---|---|
| rehearsal | $82.82 \pm 0.78$ | $-12.60 \pm 0.36$ |
| rehearsal + GAM | $86.40 \pm 0.42$ | $-7.64 \pm 0.44$ |
| rehearsal + OPGD | $\mathbf{88.23 \pm 0.16}$ | $\mathbf{-5.62 \pm 0.13}$ |
| $\ell_2$ | $73.30 \pm 1.21$ | $-21.91 \pm 1.22$ |
| $\ell_2$ + GAM | $74.15 \pm 1.33$ | $-19.19 \pm 1.43$ |
| $\ell_2$ + OPGD | $\mathbf{86.45 \pm 0.27}$ | $\mathbf{-7.42 \pm 0.33}$ |

## C.3 COMPARISON WITH ADDITIONAL BASELINES

**Additional baselines.** We additionally compare against two types of CL methods. Regularization-based methods: EWC (Kirkpatrick et al., 2017), MAS (Aljundi et al., 2018), SI (Zenke et al., 2017). Memory-based methods: ER (Chaudhry et al., 2019b), A-GEM (Chaudhry et al., 2019a). For fair comparison, we set the memory buffer size of ER and A-GEM equal to that of OPGD.

Table 6: Average accuracy (ACC) and backward transfer (BWT) over all tasks on different datasets. Higher ACC and BWT indicate better generalization and less forgetting. All results are reproduced by us and averaged over 5 runs. The best continual learning results are highlighted in **bold**.

| Dataset Method | Permuted MNIST | | Rotated MNIST | | Split CIFAR100 | |
|---|---|---|---|---|---|---|
| | ACC | BWT | ACC | BWT | ACC | BWT |
| SGD | $70.29 \pm 1.50$ | $-25.33 \pm 1.57$ | $68.79 \pm 0.43$ | $-28.09 \pm 0.45$ | $52.08 \pm 0.81$ | $-30.63 \pm 1.31$ |
| OGD | $82.17 \pm 0.64$ | $-12.38 \pm 0.66$ | $77.52 \pm 0.69$ | $-18.43 \pm 0.76$ | $63.91 \pm 1.62$ | $-20.57 \pm 1.66$ |
| EWC | $80.11 \pm 1.41$ | $-13.66 \pm 1.57$ | $79.92 \pm 0.75$ | $-2.33 \pm 0.14$ | $56.69 \pm 2.42$ | $-20.87 \pm 2.41$ |
| MAS | $83.82 \pm 0.41$ | $-5.486 \pm 0.36$ | $79.50 \pm 0.16$ | $\mathbf{1.60 \pm 0.48}$ | $66.26 \pm 1.41$ | $\mathbf{-3.54 \pm 1.30}$ |
| SI | $83.30 \pm 0.22$ | $\mathbf{-3.364 \pm 0.23}$ | $77.08 \pm 0.37$ | $-13.78 \pm 0.48$ | $67.45 \pm 2.25$ | $-8.77 \pm 2.38$ |
| GAM | $72.61 \pm 1.44$ | $-22.47 \pm 1.57$ | $72.85 \pm 0.44$ | $-20.60 \pm 0.47$ | $61.70 \pm 1.68$ | $-22.63 \pm 1.60$ |
| ER | $83.35 \pm 0.91$ | $-11.34 \pm 0.91$ | $83.05 \pm 0.34$ | $-12.75 \pm 0.39$ | $66.03 \pm 0.34$ | $-16.89 \pm 0.40$ |
| A-GEM | $84.69 \pm 0.41$ | $-9.92 \pm 0.42$ | $88.30 \pm 0.49$ | $-6.63 \pm 0.55$ | $63.04 \pm 1.59$ | $-18.26 \pm 1.78$ |
| **OGD+** | $86.22 \pm 0.62$ | $-8.11 \pm 0.62$ | $86.15 \pm 0.49$ | $-9.02 \pm 0.56$ | $61.84 \pm 2.51$ | $-23.47 \pm 2.48$ |
| **OGD+GAM** | $83.73 \pm 0.65$ | $-10.41 \pm 0.75$ | $80.31 \pm 0.51$ | $-12.00 \pm 0.55$ | $67.43 \pm 2.10$ | $-13.05 \pm 1.60$ |
| **OPGD** | $\mathbf{86.27 \pm 0.56}$ | $-7.73 \pm 0.61$ | $\mathbf{89.15 \pm 0.22}$ | $-3.69 \pm 0.27$ | $\mathbf{68.17 \pm 0.71}$ | $-12.58 \pm 1.35$ |

**Discussion.** As shown in Table 6, OPGD achieves the highest ACC across all benchmarks, indicating that penalizing the gradient norm within OGD/OGD+ is an effective strategy for improving generalization. Moreover, although OGD+ underperforms OGD on Split CIFAR-100, *combining* OGD+ with GAM outperforms *combining* OGD with GAM on this dataset. On the other hand, SI attains the highest BWT on Permuted MNIST, while MAS achieves the highest BWT on Rotated MNIST and Split CIFAR-100. As highlighted in Remark 6, this pattern reflects an inherent trade-off in OPGD: while it substantially improves generalization during training, it may also increase the risk of forgetting as training progresses. Overall, our main contribution is to provide a theoretical framework for analyzing continual learning and to improve OGD from a theoretical perspective. Although OPGD may not always yield the best BWT, it consistently delivers substantial gains in ACC and exhibits large improvements over standard OGD, thereby validating the effectiveness of our theoretically motivated design.

## C.4 ABLATION STUDY

In this section, we investigate effectiveness of each component in OPGD: balance coefficient $\alpha$ (Table 7), perturbation radius $b$ (Table 8), and per-task sampling size $m$ (Table 9).

**Balance coefficient $\alpha$.** The balance coefficient $\alpha$ controls the strength of the GAM penalty. We perform a grid search over $\{0.1, 0.2, 0.3, 0.4, 0.5, 0.6, 0.7, 0.8, 0.9\}$ using a fixed seed. As shown in Table 7, OPGD is relatively insensitive to $\alpha$, with only modest changes across the range.

**Perturbation radius $b$.** The perturbation radius $b$ controls the neighborhood size in which the GAM penalty is evaluated, thereby governing the magnitude of parameter perturbations. We conduct

Table 7: ACC and BWT of OPGD with different balance coefficients $\alpha$ across datasets.

| $\alpha$ | Permuted MNIST | | Rotated MNIST | | Split CIFAR100 | |
|---|---|---|---|---|---|---|
| | ACC | BWT | ACC | BWT | ACC | BWT |
| 0.1 | 86.52 | −7.77 | **88.79** | −4.85 | 66.90 | −15.33 |
| 0.2 | 86.41 | −7.80 | 88.39 | −4.70 | 67.70 | −14.37 |
| 0.3 | 86.10 | −8.10 | 88.15 | **−4.53** | 67.04 | −14.97 |
| 0.4 | 85.27 | −8.94 | 87.62 | −4.76 | 67.23 | −14.51 |
| 0.5 | 86.48 | **−7.35** | 87.47 | −4.56 | 68.03 | **−13.84** |
| 0.6 | **86.55** | −7.42 | 87.02 | −4.82 | 67.89 | −14.17 |
| 0.7 | 84.76 | −9.30 | 86.15 | −5.58 | 67.98 | −14.21 |
| 0.8 | 85.40 | −8.62 | 85.79 | −5.78 | **68.18** | −14.03 |
| 0.9 | 86.41 | −7.52 | 85.76 | −5.60 | 67.64 | −14.73 |

Table 8: ACC and BWT of OPGD with different perturbation radii $b$ across datasets.

| $b$ | Permuted MNIST | | Rotated MNIST | | Split CIFAR100 | |
|---|---|---|---|---|---|---|
| | ACC | BWT | ACC | BWT | ACC | BWT |
| 0.02 | **86.41** | −7.47 | 88.47 | −5.40 | 68.48 | −13.82 |
| 0.05 | 86.01 | −7.60 | 88.79 | −4.85 | **69.31** | −11.03 |
| 0.1 | 84.79 | −8.17 | 89.01 | −4.35 | 66.02 | −9.90 |
| 0.2 | 85.79 | **−5.96** | **89.09** | −3.76 | 53.13 | −9.45 |
| 0.5 | 79.70 | −7.08 | 87.97 | −3.59 | 50.53 | −8.78 |
| 1.0 | 72.61 | −6.59 | 87.72 | **−2.50** | 48.45 | **−8.39** |
| 2.0 | 70.36 | −8.05 | 77.18 | −2.90 | 47.45 | −9.12 |

a grid search over $\{0.02, 0.05, 0.1, 0.2, 0.5, 1.0, 2.0\}$ using a fixed seed. We observe that enlarging $b$ is not always beneficial for OPGD, especially when $b > 0.2$. Conceptually, OPGD aims to reduce the *local* Lipschitz constant; GAM does so by penalizing the maximal gradient norm within a ball of radius $b$, which serves as a proxy upper bound for the local Lipschitz constant. When $b$ becomes too large, the neighborhood is no longer local, the proxy bound becomes loose—impeding optimization and degrading fit—thereby explaining the observed saturation or decline in performance at large $b$.

Table 9: ACC and BWT of OPGD with different per-task sampling sizes $m$ across datasets.

| $m$ | Permuted MNIST | | Rotated MNIST | | Split CIFAR100 | |
|---|---|---|---|---|---|---|
| | ACC | BWT | ACC | BWT | ACC | BWT |
| 20 | 79.85 | −14.73 | 82.79 | −11.05 | 66.09 | −14.50 |
| 40 | 83.89 | −10.46 | 85.21 | −8.26 | 59.60 | −21.08 |
| 60 | 83.51 | −10.76 | 87.12 | −6.07 | 62.69 | −18.10 |
| 80 | 85.57 | −8.44 | 88.33 | −4.69 | 65.21 | −14.79 |
| 100 | 85.53 | −8.52 | 89.09 | −3.76 | **69.25** | **−11.10** |
| 120 | 86.64 | −7.33 | 89.87 | −2.80 | 66.36 | −13.52 |
| 140 | **87.17** | **−6.71** | **90.23** | **−2.29** | 68.56 | −11.67 |

**Per-task sampling size $m$.** In OPGD, we randomly sample $m$ examples from each task to store in the memory buffer. We perform a grid search over $\{20, 40, 60, 80, 100, 120, 140\}$ using a fixed seed. As shown in Table 9, both ACC and BWT exhibit a clear increasing trend as the buffer size grows. Due to GPU memory constraints, we report results up to $m = 140$; nevertheless, the observed trend indicates that larger buffers would likely yield further gains for OPGD.

# D KERNEL GRADIENT FLOW

## D.1 KERNEL GRADIENT FLOW UNDER SGD

In this section, we derive Equation (11) by solving the linear ODE in Equation (10). We start by evaluating Equation (10) at the training dataset $X_\tau$:

$$\frac{d}{dt} f_\tau^t(X_\tau) = -\frac{1}{n_\tau} K_\tau(X_\tau, X_\tau)\big(f_\tau^t(X_\tau) - Y_\tau\big). \tag{19}$$

Let $g(t) = f_\tau^t(X_\tau) - Y_\tau$. Then, Equation (19) can be rewritten in the simplified form:

$$\frac{d}{dt} g(t) = -\frac{1}{n_\tau} K_\tau(X_\tau, X_\tau) g(t), \qquad g(0) = f_\tau^0(X_\tau) - Y_\tau. \tag{20}$$

Equation (20) is a linear matrix ODE, where $K_\tau(X_\tau, X_\tau)$ is time-invariant and real symmetric. Therefore, the theory of linear ODE guarantees a unique solution:

$$g(t) = \exp\left(-\tfrac{t}{n_\tau}K_\tau(X_\tau, X_\tau)\right)g(0) = \exp\left(-\tfrac{t}{n_\tau}K_\tau(X_\tau, X_\tau)\right)\left(f_\tau^0(X_\tau) - Y_\tau\right). \tag{21}$$

Substituting $g(t) = f_\tau^t(X_\tau) - Y_\tau$ into Equation (21) yields:

$$f_\tau^t(X_\tau) = Y_\tau + \exp\left(-\tfrac{t}{n_\tau}K_\tau(X_\tau, X_\tau)\right)\left(f_\tau^0(X_\tau) - Y_\tau\right). \tag{22}$$

Therefore, we obtain the explicit form of $f_\tau^t(\cdot)$ on the training dataset $X_\tau$. Next, we fix an arbitrary test point $x \in \mathbb{R}^d$. For this $x$, Equation (10) specializes to

$$\frac{d}{dt}f_\tau^t(x) = -\frac{1}{n_\tau}K_\tau(x, X_\tau)\left(f_\tau^t(X_\tau) - Y_\tau\right). \tag{23}$$

Integrating both sides of Equation (23) over the interval $[0, t]$ and applying the initial condition $f_\tau^0(x) = f_{\tau-1}^*(x)$, we obtain

$$f_\tau^t(x) - f_\tau^0(x) = -\frac{1}{n_\tau}K_\tau(x, X_\tau)\int_0^t\left(f_\tau^s(X_\tau) - Y_\tau\right)ds. \tag{24}$$

Substituting Equation (22) into the integral term of Equation (24), we obtain

$$\begin{aligned}\int_0^t\left(f_\tau^s(X_\tau) - Y_\tau\right)ds &= \int_0^t\exp\left(-\tfrac{s}{n_\tau}K_\tau(X_\tau, X_\tau)\right)\left(f_\tau^0(X_\tau) - Y_\tau\right)ds \\ &= \left(\int_0^t\exp\left(-\tfrac{s}{n_\tau}K_\tau(X_\tau, X_\tau)\right)ds\right)\left(f_\tau^0(X_\tau) - Y_\tau\right).\end{aligned} \tag{25}$$

The matrix integral in Equation (25) can be evaluated in closed form by applying the standard identity (valid for any constant matrix $A$ and scalar $\alpha > 0$):

$$\int_0^t\exp(-\alpha sA)\,ds = \alpha^{-1}A^{-1}\left(I - \exp(-\alpha tA)\right), \tag{26}$$

provided that $A$ is invertible.

Applying Equation (26) with $A = K_\tau(X_\tau, X_\tau)$ and $\alpha = 1/n_\tau$ gives

$$\int_0^t\exp\left(-\tfrac{s}{n_\tau}K_\tau(X_\tau, X_\tau)\right)ds = n_\tau K_\tau(X_\tau, X_\tau)^{-1}\left(I - \exp\left(-\tfrac{t}{n_\tau}K_\tau(X_\tau, X_\tau)\right)\right). \tag{27}$$

Plugging Equation (27) into Equation (24), we obtain

$$\begin{aligned}&f_\tau^t(x) - f_\tau^0(x) \\ &= -\frac{1}{n_\tau}K_\tau(x, X_\tau)\left(n_\tau K_\tau(X_\tau, X_\tau)^{-1}\left(I - \exp(-\tfrac{t}{n_\tau}K_\tau(X_\tau, X_\tau))\right)\right)\left(f_\tau^0(X_\tau) - Y_\tau\right) \\ &= -K_\tau(x, X_\tau)K_\tau(X_\tau, X_\tau)^{-1}\left(I - \exp(-\tfrac{t}{n_\tau}K_\tau(X_\tau, X_\tau))\right)\left(f_\tau^0(X_\tau) - Y_\tau\right).\end{aligned} \tag{28}$$

Recalling that $f_\tau^0(\cdot) = f_{\tau-1}^*(\cdot)$, we arrive at the closed-form solution for any $x \in \mathbb{R}^d$:

$$f_\tau^t(x) = f_{\tau-1}^*(x) - K_\tau(x, X_\tau)\left(I - \exp\left(-\tfrac{t}{n_\tau}K_\tau(X_\tau, X_\tau)\right)\right)K_\tau(X_\tau, X_\tau)^{-1}\left(f_{\tau-1}^*(X_\tau) - Y_\tau\right). \tag{29}$$

For notational simplicity, we define

$$E_{\tau,t} := I - \exp\left(-\frac{t}{n_\tau} K_\tau(X_\tau, X_\tau)\right). \tag{30}$$

Thus, the solution can be compactly expressed as

$$f_\tau^t(x) = f_{\tau-1}^*(x) - K_\tau(x, X_\tau) E_{\tau,t} K_\tau(X_\tau, X_\tau)^{-1} \left(f_{\tau-1}^*(X_\tau) - Y_\tau\right). \tag{31}$$

We complete the derivation of Equation (11) by solving the linear ODE in Equation (10). Next, we introduce an important lemma that will be used in the subsequent proofs.

**Lemma 5.** *For any $\tau \in [T]$, both $e^{-\frac{1}{n_\tau} K_\tau(X_\tau, X_\tau) t_\tau^*}$ and $E_{\tau,t_\tau^*}$ are symmetric and positive definite.*

*Proof.* Let $\lambda_{k,n_\tau} > 0$ $(k \in [n_\tau])$ be the eigenvalues of $K_\tau(X_\tau, X_\tau)$. Therefore, there exists an orthogonal matrix $Q_\tau$ such that

$$Q_\tau K_\tau(X_\tau, X_\tau) Q_\tau^\top = diag\{\lambda_{\tau,1}, \ldots, \lambda_{\tau,n_\tau}\}, \tag{32}$$

where $\lambda_{\tau,1}, \ldots, \lambda_{\tau,n_\tau}$ are the eigenvalues of $K_\tau(X_\tau, X_\tau)$.

$$\begin{aligned}
Q_\tau e^{-\frac{1}{n_\tau} K_\tau(X_\tau, X_\tau) t_\tau^*} Q_\tau^\top &= Q_\tau \sum_{k=0}^\infty \frac{1}{k!} \left(-\frac{1}{n_\tau} K_\tau(X_\tau, X_\tau) t_\tau^*\right)^k Q_\tau^\top \\
&= \sum_{k=0}^\infty \frac{1}{k!} \left(-\frac{1}{n_\tau} Q_\tau K_\tau(X_\tau, X_\tau) Q_\tau^\top t_\tau^*\right)^k \\
&= \sum_{k=0}^\infty \frac{1}{k!} \left(-\frac{1}{n_\tau} diag\{\lambda_{\tau,1}, \ldots, \lambda_{\tau,n_\tau}\} t_\tau^*\right)^k \\
&= diag\left\{\sum_{k=0}^\infty \frac{1}{k!} \left(-\frac{\lambda_{\tau,1} t_\tau^*}{n_\tau}\right)^k, \ldots, \sum_{k=0}^\infty \frac{1}{k!} \left(-\frac{\lambda_{\tau,n_\tau} t_\tau^*}{n_\tau}\right)^k\right\} \\
&= diag\left\{e^{-\frac{\lambda_{\tau,1} t_\tau^*}{n_\tau}}, \ldots, e^{-\frac{\lambda_{\tau,n_\tau} t_\tau^*}{n_\tau}}\right\}.
\end{aligned} \tag{33}$$

For any $t_\tau^* > 0$ we have $0 < \exp(-\lambda_{k,n_\tau} t_\tau^*/n_\tau) < 1$. Consequently the matrix exponential $\exp\left(-\frac{1}{n_\tau} K_\tau(X_\tau, X_\tau) t_\tau^*\right)$ is symmetric positive definite, and thus $E_{\tau,t_\tau^*}$ is also symmetric positive definite with eigenvalues $1 - \exp(-\lambda_{k,n_\tau} t_\tau^*/n_\tau) \in (0,1)$. Even if we relax the condition to $K_\tau(\cdot, \cdot)$ being only positive semi-definite, the matrix exponential $\exp\left(-\frac{1}{n_\tau} K_\tau(X_\tau, X_\tau) t_\tau^*\right)$ remains symmetric positive definite, since the exponential of any symmetric matrix with nonnegative eigenvalues yields strictly positive eigenvalues.

$\square$

### D.2 KERNEL GRADIENT FLOW UNDER OGD

In this section, we derive Equation (17), which characterizes the gradient flow dynamics under OGD. For any task $\tau \in [T]$, the parameter $\theta_\tau$ evolves according to the differential equation

$$\frac{d\theta_\tau^t}{dt} = -P_\tau \nabla_{\theta_\tau^t} \ell(\theta_\tau^t) = -P_\tau \frac{1}{n_\tau} \sum_{j=1}^{n_\tau} \left(f_\tau(x_\tau^j) - y_\tau^j\right) \nabla_{\theta_\tau^t} f_\tau^t(x_\tau^j), \tag{34}$$

where $t \geq 0$ denotes continuous time.

Based on Equation (34), the evolution of the network output satisfies

$$\begin{aligned}
\frac{d}{dt} f_\tau^t(x) = \nabla_{\theta_\tau^t} f_\tau^t(x) \frac{d\theta_\tau}{dt} &= -\frac{1}{n_\tau} \sum_{j=1}^{n_\tau} \left(f_\tau^t(x_\tau^j) - y_\tau^j\right) \left\langle \nabla_{\theta_\tau^t} f_\tau^t(x), P_\tau \nabla_{\theta_\tau^t} f_\tau^t(x_\tau^j)\right\rangle \\
&= -\frac{1}{n_\tau} \sum_{j=1}^{n_\tau} \left(f_\tau^t(x_\tau^j) - y_\tau^j\right) \left\langle P_\tau \nabla_{\theta_\tau^t} f_\tau^t(x), P_\tau \nabla_{\theta_\tau^t} f_\tau^t(x_\tau^j)\right\rangle.
\end{aligned} \tag{35}$$

Therefore, under the NTK regime, the kernel gradient flow takes the following form:

$$\frac{d}{dt}f_\tau^t(x) = -\frac{1}{n_\tau}\tilde{K}_\tau(x, X_\tau)\big(f_\tau^t(X_\tau) - Y_\tau\big), \tag{36}$$

where $\tilde{K}_\tau(x, x') = \langle P_\tau \nabla_{\theta_{\tau-1}^*} f_{\tau-1}^*(x), P_\tau \nabla_{\theta_{\tau-1}^*} f_{\tau-1}^*(x')\rangle$. Therefore, the resulting kernel coincides with the one derived in Bennani et al. (2020).

## E    PROOF OF THEOREM 1

In this section, we provide the proof of Theorem 1, which establishes the upper bounds of forgetting $F_T$ and generalization error $G_T$. We first introduce the notion of Rademacher complexity in Subsection E.1. We then derive the upper bound of the generalization error in Subsection E.2, followed by the proof of the upper bound of forgetting in Subsection E.3.

### E.1    GENERALIZATION AND RADEMACHER COMPLEXITY

There are several ways to quantify the complexity of a function class $\mathcal{F}$, one important and widely used measure is the Rademacher complexity. Following the notation in Arora et al. (2019a), we define the empirical Rademacher complexity as follows:

**Definition 1.** *Given a sample set $S_\tau = \{(x_\tau^i, y_\tau^i)\}_{i=1}^{n_\tau}$, the empirical Rademacher complexity of a function class $\mathcal{F}$ is defined as:*

$$\mathcal{R}_{S_\tau}(\mathcal{F}) = \frac{1}{n_\tau}\mathbb{E}_{\boldsymbol{\epsilon}}\left[\sup_{f\in\mathcal{F}}\sum_{i=1}^{n_\tau}\epsilon_i f(x_\tau^i)\right], \tag{37}$$

*where $\boldsymbol{\epsilon} = (\epsilon_1, \ldots, \epsilon_n)^\top$ is a vector of i.i.d. random variables drawn from the Rademacher distribution, i.e., $\epsilon_i \sim \mathrm{Unif}(-1, +1)$.*

Rademacher complexity provides a data-dependent upper bound on the generalization error of a learning algorithm (Bartlett & Mendelson, 2002).

**Theorem 2.** *Suppose the loss function $\ell(\cdot, \cdot)$ is bounded in [0,c] and is $\rho-$Lipschitz in the first argument. Then, with probability at least $1 - \delta$, for all $f \in \mathcal{F}$, it holds that*

$$L_\mathcal{D}(f) - L_S(f) \le 2\rho\mathcal{R}_S(\mathcal{F}) + 3c\sqrt{\frac{log(2/\delta)}{2n}}. \tag{38}$$

Based on Theorem 2, we state the following corollary:

**Corollary 1.** *Suppose the loss function $\ell(\cdot, \cdot)$ is bounded in [0,c] and is $\rho-$Lipschitz in the first argument. Then, with probability at least $1 - \delta$, for all $f \in \mathcal{F}$, it holds that*

$$L_S(f) - L_\mathcal{D}(f) \le 2\rho\mathcal{R}_S(\mathcal{F}) + 3c\sqrt{\frac{log(2/\delta)}{2n}}. \tag{39}$$

*Proof.* Let $\mathcal{G} := \{g_f(z) = \ell(f(x), y) : f \in \mathcal{F}\}$, where each $g_f$ takes values in $[0, c]$. For any $f \in \mathcal{F}$, define the population risk $L_D(f) = \mathbb{E}_D[g_f]$ and the empirical risk $L_S(f) = \mathbb{E}_S[g_f]$.

From the standard Rademacher generalization bound, with probability at least $1 - \delta$, it holds that

$$L_\mathcal{D}(f) \le L_S(f) + 2\mathcal{R}_S(\mathcal{G}) + 3c\sqrt{\frac{\log(2/\delta)}{2n}}. \tag{40}$$

To obtain the reverse direction, consider the shifted function class

$$\mathcal{G}' := \{g_f'(z) = c - g_f(z) : f \in \mathcal{F}\},$$

which also takes values in $[0, c]$ and satisfies $\mathcal{R}_S(\mathcal{G}') = \mathcal{R}_S(\mathcal{G})$. Applying the same bound to $\mathcal{G}'$ yields

$$L_S(f) - L_\mathcal{D}(f) \le 2\mathcal{R}_S(\mathcal{G}) + 3c\sqrt{\frac{\log(2/\delta)}{2n}}.$$

Finally, by the contraction lemma, since $\ell(\cdot, y)$ is $\rho$-Lipschitz in its first argument, we have

$$\mathcal{R}_S(\mathcal{G}) \le \rho \mathcal{R}_S(\mathcal{F}).$$

Substituting this completes the proof. □

Next, we provide an upper bound on the Rademacher complexity of a specific form of function class, as stated in Lemma 6.

**Lemma 6.** *Let $\{K_t : \mathcal{X}_t \times \mathcal{X}_t \to \mathbb{R}\}_{t=1}^T$ be a sequence of positive semi-definite kernels such that $\sup_{x \in \mathcal{X}} \|K_t(x, x)\| < \infty$ for all $t \in [T]$. For each $t \in [T]$, let $\mathcal{H}_t$ be the reproducing kernel Hilbert space (RKHS) associated with $K_t$, equipped with inner product $\langle \cdot, \cdot \rangle_{\mathcal{H}_t}$. Given a sequence of positive constants $\{B_t\}_{t=1}^T$, we define the function class $\mathcal{F}_T$ as*

$$\mathcal{F}_T = \left\{ f : \mathcal{X} \to \mathbb{R} \,\middle|\, x \to \sum_{t=1}^T f_t(x), f_t(x) = K_t(x, X_t)\alpha_t, \|f_t\|_{\mathcal{H}_t} \le B_t, \forall t \in [T] \right\}. \tag{41}$$

*Then the empirical Rademacher complexity of $\mathcal{F}_T$ over $S_\tau$ satisfies*

$$\mathcal{R}_{S_\tau}(\mathcal{F}_T) \le \sum_{t=1}^T \frac{B_t}{n_\tau} (\mathrm{Tr}(K_t(X_\tau, X_\tau)))^{1/2}. \tag{42}$$

*Proof.* For any kernel $K_t$, there exists an associated feature map $\Phi_t : \mathcal{X}_t \to \mathcal{H}_t$ such that for all $x_1, x_2 \in \mathcal{X}_t$, we have $K_t(x_1, x_2) = \langle \Phi_t(x_1), \Phi_t(x_2) \rangle_{\mathcal{H}_t}$. In particular, the kernel vector $K_t(x, X_t)$ is defined as $K_t(x, X_t) = (K_t(x, x_t^1), \ldots, K_t(x, x_t^{n_t}))^\top$ and the coefficient vector is given by $\alpha_t = (\alpha_t^1, \ldots, \alpha_t^{n_t})^\top$. Consequently, for any $f \in \mathcal{F}_T$, we have

$$\begin{aligned}
f(x) &= \sum_{t=1}^T f_t(x) \\
&= \sum_{t=1}^T K_t(x, X_t)\alpha_t \\
&= \sum_{t=1}^T \sum_{i=1}^{n_t} \alpha_t^i K_t(x, x_t^i) \\
&= \sum_{t=1}^T \sum_{i=1}^{n_t} \alpha_t^i \langle \Phi_t(x), \Phi_t(x_t^i) \rangle_{\mathcal{H}_t} \\
&= \sum_{t=1}^T \langle \Phi_t(x), \sum_{i=1}^{n_t} \alpha_t^i \Phi_t(x_t^i) \rangle_{\mathcal{H}_t}.
\end{aligned} \tag{43}$$

Let $w_t = \sum_{i=1}^{n_t} \alpha_t^i \Phi_t(x_t^i)$. Then the function $f$ can be represented as:

$$f(x) = \sum_{t=1}^T \langle w_t, \Phi_t(x) \rangle_{\mathcal{H}_t}. \tag{44}$$

Moreover, the squared norm of $w_t$ in $\mathcal{H}_t$ satisfies:

$$\begin{aligned}
\|w_t\|_{\mathcal{H}_t}^2 &= \sum_{i,j}^{n_t} \alpha_t^i \alpha_t^j \langle \Phi_t(x_t^i), \Phi_t(x_t^j) \rangle_{\mathcal{H}_t} \\
&= \sum_{i,j}^{n_t} \alpha_t^i \alpha_t^j K_t(x_t^i, x_t^j) \\
&= \alpha_t^\top K_t(X_t, X_t)\alpha_t \\
&= \|f_t\|_{\mathcal{H}_t}^2.
\end{aligned} \tag{45}$$

We define the function class $\tilde{\mathcal{F}}_T$ as follows:

$$\tilde{\mathcal{F}}_T = \left\{ f : \mathcal{X} \to \mathbb{R} \,\middle|\, x \to \sum_{t=1}^{T} \langle w_t, \Phi_t(x) \rangle_{\mathcal{H}_t}, \|w_t\|_{\mathcal{H}_t} \leq B_t, \forall t \in [T] \right\}. \tag{46}$$

By construction, we have $\mathcal{F}_T \subset \tilde{\mathcal{F}}_T$. Consequently, the empirical Rademacher complexity of $\mathcal{F}_T$ over $S_\tau$ can be upper bounded by that of $\tilde{\mathcal{F}}_T$, i.e.,

$$
\begin{aligned}
\mathcal{R}_{S_\tau}(\mathcal{F}_T) \leq \mathcal{R}_{S_\tau}(\tilde{\mathcal{F}}_T) =& \frac{1}{n_\tau} \mathbb{E}_\epsilon \Big[ \sup_{\|w_t\|_{\mathcal{H}_t} \leq B_t, \forall t \in [T]} \sum_{i=1}^{n_\tau} \epsilon_i \sum_{t=1}^{T} \langle w_t, \Phi_t(x_\tau^i) \rangle_{\mathcal{H}_t} \Big] \\
=& \frac{1}{n_\tau} \mathbb{E}_\epsilon \Big[ \sup_{\|w_t\|_{\mathcal{H}_t} \leq B_t, \forall t \in [T]} \sum_{t=1}^{T} \langle w_t, \sum_{i=1}^{n_\tau} \epsilon_i \Phi_t(x_\tau^i) \rangle_{\mathcal{H}_t} \Big] \\
\leq& \frac{1}{n_\tau} \sum_{t=1}^{T} \mathbb{E}_\epsilon \Big[ \sup_{\|w_t\|_{\mathcal{H}_t} \leq B_t} \langle w_t, \sum_{i=1}^{n_\tau} \epsilon_i \Phi_t(x_\tau^i) \rangle_{\mathcal{H}_t} \Big] \\
=& \sum_{t=1}^{T} \frac{B_t}{n_\tau} \mathbb{E}_\epsilon \Big[ \| \sum_{i=1}^{n_\tau} \epsilon_i \Phi_t(x_\tau^i) \|_{\mathcal{H}_t} \Big] \\
=& \sum_{t=1}^{T} \frac{B_t}{n_\tau} \mathbb{E}_\epsilon \Big[ \sqrt{\sum_{i,j} \epsilon_i \epsilon_j K_t(x_\tau^i, x_\tau^j)} \Big] \\
\leq& \sum_{t=1}^{T} \frac{B_t}{n_\tau} \sqrt{\sum_{i,j} \mathbb{E}_\epsilon [\epsilon_i \epsilon_j K_t(x_\tau^i, x_\tau^j)]} \\
=& \sum_{t=1}^{T} \frac{B_t}{n_\tau} \sqrt{\sum_{i} \mathbb{E}_\epsilon [\epsilon_i^2 K_t(x_\tau^i, x_\tau^i)]} \\
=& \sum_{t=1}^{T} \frac{B_t}{n_\tau} \sqrt{\sum_{i} K_t(x_\tau^i, x_\tau^i)} \\
=& \sum_{t=1}^{T} \frac{B_t}{n_\tau} (\mathrm{Tr}(K_t(X_\tau, X_\tau)))^{1/2}.
\end{aligned}
\tag{47}
$$

$\square$

### E.2 BOUND ON THE GENERALIZATION ERROR $G_{t_T}$

In order to derive an upper bound on the generalization error defined in Equation (4), we utilize the inequality provided in Equation (38). To proceed, we will separately bound the empirical loss term $L_{S_\tau}(f_T^*)$ and the Rademacher complexity term $\mathcal{R}_{S_\tau}(\mathcal{F}_T)$.

*Proof.* (1) For the term $L_{S_\tau}(f_T^*)$ for any $\tau \in [T]$, we have:

$$
\begin{aligned}
L_{S_\tau}(f_T^*) =& \frac{1}{2n_\tau} \| f_T^*(X_\tau) - Y_\tau \|^2 \\
=& \frac{1}{2n_\tau} \| f_\tau^*(X_\tau) + \sum_{k=\tau+1}^{T} \tilde{f}_k^*(X_\tau) - Y_\tau \|^2 \\
\leq& \frac{1}{n_\tau} \| f_\tau^*(X_\tau) - Y_\tau \|^2 + \frac{1}{n_\tau} \| \sum_{k=\tau+1}^{T} \tilde{f}_k^*(X_\tau) \|^2.
\end{aligned}
\tag{48}
$$

Notably, the convention $\sum_{k=T+1}^{T} \cdot = 0$ always holds, which is known as the empty sum convention. Therefore, when $\tau = T$, Equation (48) remains valid.

Next, we compute the term $\|f_\tau^*(X_\tau) - Y_\tau\|^2$ as follows:

$$
\begin{aligned}
\|f_\tau^*(X_\tau) - Y_\tau\|^2 &= \|f_{\tau-1}^*(X_\tau) + \tilde{f}_\tau^*(X_\tau) - Y_\tau\|^2 \\
&= \|\tilde{f}_\tau^*(X_\tau) - \tilde{Y}_\tau\|^2 \\
&= \|K_\tau(X_\tau, X_\tau)E_{\tau,t_\tau^*}K_\tau(X_\tau, X_\tau)^{-1}\tilde{Y}_\tau - \tilde{Y}_\tau\|^2 \\
&= \|\tilde{Y}_\tau - K_\tau(X_\tau, X_\tau)e^{-\frac{t_\tau^*}{n_\tau}K_\tau(X_\tau,X_\tau)}K_\tau(X_\tau, X_\tau)^{-1}\tilde{Y}_\tau - \tilde{Y}_\tau\|^2 \\
&= \|K_\tau(X_\tau, X_\tau)e^{-\frac{t_\tau^*}{n_\tau}K_\tau(X_\tau,X_\tau)}K_\tau(X_\tau, X_\tau)^{-1}\tilde{Y}_\tau\|^2.
\end{aligned}
\tag{49}
$$

In order to simplify the result in Equation (49), we use the Taylor expansion of the exponential function, i.e., $e^X = \sum_k^\infty \frac{1}{k!}X^k$. Therefore, we have:

$$
\begin{aligned}
\|f_\tau^*(X_\tau) - Y_\tau\|^2 &= \|K_\tau(X_\tau, X_\tau)\sum_k^\infty \frac{1}{k!}(-\frac{t_\tau^*}{n_\tau}K_\tau(X_\tau, X_\tau)*)^k K_\tau(X_\tau, X_\tau)^{-1}\tilde{Y}_\tau\|^2 \\
&= \|\sum_k^\infty \frac{1}{k!}K_\tau(X_\tau, X_\tau)(-\frac{t_\tau^*}{n_\tau}K_\tau(X_\tau, X_\tau))^k K_\tau(X_\tau, X_\tau)^{-1}\tilde{Y}_\tau\|^2 \\
&= \|\sum_k^\infty \frac{1}{k!}(-\frac{t_\tau^*}{n_\tau}K_\tau(X_\tau, X_\tau))^k \tilde{Y}_\tau\|^2 \\
&= \|e^{-\frac{t_\tau^*}{n_\tau}K_\tau(X_\tau,X_\tau)}\tilde{Y}_\tau\|^2.
\end{aligned}
\tag{50}
$$

Using the Cauchy-Schwarz inequality $\|\sum_{i=1}^m \mathbf{v}_i\|^2 \leq m\sum_{i=1}^m \|\mathbf{v}_i\|^2$, we obtain the following upper bound:

$$
L_{S_\tau}(f_T^*) \leq \frac{T-\tau}{n_\tau}\sum_{k=\tau+1}^T \|K_k(X_\tau, X_k)E_{k,t_k^*}K_k(X_k, X_k)^{-1}\tilde{Y}_k\|^2 + \frac{1}{n_\tau}\|e^{-\frac{t_\tau^*}{n_\tau}K_\tau(X_\tau,X_\tau)}\tilde{Y}_\tau\|^2.
\tag{51}
$$

(2) For the term $\mathcal{R}_{S_\tau}(\mathcal{F}_T)$, we first consider a bound on the reproduced kernel Hilbert space (RKHS) norm of $\tilde{f}_\tau^*$. Let $(\mathcal{H}_{K_\tau}, \|\cdot\|_{\mathcal{H}_{K_\tau}})$ be the RKHS induced by the kernel $K_\tau$. We define

$$
\alpha_\tau := E_{\tau,t_\tau^*}K_\tau(X_\tau, X_\tau)^{-1}\tilde{Y}_\tau.
\tag{52}
$$

Then, $\tilde{f}_\tau^*$ can be written as:

$$
\tilde{f}_\tau^*(x) = K_\tau(x, X_\tau)\alpha_\tau.
\tag{53}
$$

The RKHS norm of $\tilde{f}_\tau^*$ is then given by:

$$
\begin{aligned}
\|\tilde{f}_\tau^*\|_{\mathcal{H}_{K_\tau}}^2 &= \alpha_\tau^\top K_\tau(X_\tau, X_\tau)\alpha_\tau \\
&= \tilde{Y}_\tau^\top K_\tau(X_\tau, X_\tau)^{-1}E_{\tau,t_\tau^*}K_\tau(X_\tau, X_\tau)E_{\tau,t_\tau^*}K_\tau(X_\tau, X_\tau)^{-1}\tilde{Y}_\tau \\
&\leq \tilde{Y}_\tau^\top E_{\tau,t_\tau^*}K_\tau(X_\tau, X_\tau)^{-1}\tilde{Y}_\tau := B_\tau^2.
\end{aligned}
\tag{54}
$$

The final inequality in Equation (54) is easily verified by the following equation:

$$
\begin{aligned}
&\tilde{Y}_\tau^\top K_\tau(X_\tau, X_\tau)^{-1}E_{\tau,t_\tau^*}K_\tau(X_\tau, X_\tau)E_{\tau,t_\tau^*}K_\tau(X_\tau, X_\tau)^{-1}\tilde{Y}_\tau - \tilde{Y}_\tau^\top E_{\tau,t_\tau^*}K_\tau(X_\tau, X_\tau)^{-1}\tilde{Y}_\tau \\
=&\tilde{Y}_\tau^\top [K_\tau(X_\tau, X_\tau)^{-1}E_{\tau,t_\tau^*}K_\tau(X_\tau, X_\tau) - I]E_{\tau,t_\tau^*}K_\tau(X_\tau, X_\tau)^{-1}\tilde{Y}_\tau \\
=&\tilde{Y}_\tau^\top [K_\tau(X_\tau, X_\tau)^{-1}E_{\tau,t_\tau^*}K_\tau(X_\tau, X_\tau) - K_\tau(X_\tau, X_\tau)^{-1}K_\tau(X_\tau, X_\tau)]E_{\tau,t_\tau^*}K_\tau(X_\tau, X_\tau)^{-1}\tilde{Y}_\tau \\
=&\tilde{Y}_\tau^\top K_\tau(X_\tau, X_\tau)^{-1}(E_{\tau,t_\tau^*} - I)K_\tau(X_\tau, X_\tau)E_{\tau,t_\tau^*}K_\tau(X_\tau, X_\tau)^{-1}\tilde{Y}_\tau \\
=&-\tilde{Y}_\tau^\top K_\tau(X_\tau, X_\tau)^{-1}e^{-\frac{t_\tau^*}{n_\tau}K_\tau(X_\tau,X_\tau)}K_\tau(X_\tau, X_\tau)E_{\tau,t_\tau^*}K_\tau(X_\tau, X_\tau)^{-1}\tilde{Y}_\tau.
\end{aligned}
\tag{55}
$$

Based on Lemma 5, since $K_\tau(X_\tau, X_\tau)$, $K_\tau(X_\tau, X_\tau)^{-1}$, $e^{-\frac{1}{n_\tau}K_\tau(X_\tau,X_\tau)t^*_\tau}$, and $E_{\tau,t^*_\tau}$ are all positive semi-definite, it follows that

$$\tilde{Y}_\tau^\top K_\tau(X_\tau, X_\tau)^{-1} E_{\tau,t^*_\tau} K_\tau(X_\tau, X_\tau) E_{\tau,t^*_\tau} K_\tau(X_\tau, X_\tau)^{-1} \tilde{Y}_\tau - \tilde{Y}_\tau^\top E_{\tau,t^*_\tau} K_\tau(X_\tau, X_\tau)^{-1} \tilde{Y}_\tau \leq 0. \tag{56}$$

Therefore, we verify that Equation (54) holds, and we obtain an upper bound for $\|\tilde{f}^*_\tau\|_{\mathcal{H}_{K_\tau}}$, which we denote by $B_\tau$. We define the function class $\mathcal{F}_T$ as follows:

$$\mathcal{F}_T = \left\{ f : \mathcal{X} \to \mathbb{R} \middle| x \to \sum_{\tau=1}^{T} \tilde{f}_\tau(x), \tilde{f}_\tau(x) = K_\tau(x, X_\tau)\alpha_\tau, \|\tilde{f}_\tau\|_{\mathcal{H}_{K_\tau}} \leq B_\tau, \forall \tau \in [T] \right\}. \tag{57}$$

Based on Lemma 6, we obtain the following bound on the empirical Rademacher complexity of $\mathcal{F}_T$ over $S_\tau$:

$$\begin{aligned} \mathcal{R}_{S_\tau}(\mathcal{F}_T) &\leq \sum_{k=1}^{T} \frac{B_k}{n_\tau} (\mathrm{Tr}(K_k(X_\tau, X_\tau)))^{1/2} \\ &\leq \sum_{k=1}^{T} \frac{[\mathrm{Tr}(K_k(X_\tau, X_\tau))\tilde{Y}_k^\top E_{k,t^*_k} K_k(X_k, X_k)^{-1}\tilde{Y}_k]^{1/2}}{n_\tau}. \end{aligned} \tag{58}$$

Based on Theorem 2, and by combining Equation (51) with Equation (58), we have:

$$\begin{aligned} L_{\mathcal{D}_\tau}(f^*_T) \leq & \frac{T-\tau}{n_\tau} \sum_{k=\tau+1}^{T} \|K_k(X_\tau, X_k)E_{k,t^*_k}K_k(X_k, X_k)^{-1}\tilde{Y}_k\|^2 + \frac{1}{n_\tau}\|e^{-\frac{t^*_\tau}{n_\tau}K_\tau(X_\tau,X_\tau)}\tilde{Y}_\tau\|^2 \\ & + 2\rho \sum_{k=1}^{T} \frac{[\mathrm{Tr}(K_k(X_\tau, X_\tau))\tilde{Y}_k^\top E_{k,t^*_k} K_k(X_k, X_k)^{-1}\tilde{Y}_k]^{1/2}}{n_\tau} + 3c\sqrt{\frac{log(2/\delta)}{2n_\tau}}. \end{aligned} \tag{59}$$

By substituting the bound from Equation (59) into Equation (4), we obtain:

$$\begin{aligned} G_{t_T} \leq & \frac{1}{T} \sum_{\tau=1}^{T} \left\{ \frac{T-\tau}{n_\tau} \sum_{k=\tau+1}^{T} \|K_k(X_\tau, X_k)E_{k,t^*_k}K_k(X_k, X_k)^{-1}\tilde{Y}_k\|^2 + \frac{1}{n_\tau}\|e^{-\frac{t^*_\tau}{n_\tau}K_\tau(X_\tau,X_\tau)}\tilde{Y}_\tau\|^2 \right. \\ & \left. + 2\rho \sum_{k=1}^{T} \frac{[\mathrm{Tr}(K_k(X_\tau, X_\tau))\tilde{Y}_k^\top E_{k,t^*_k} K_k(X_k, X_k)^{-1}\tilde{Y}_k]^{1/2}}{n_\tau} + 3c\sqrt{\frac{log(2/\delta)}{2n_\tau}} \right\}. \end{aligned} \tag{60}$$

$\square$

### E.3 BOUND ON FORGETTING $F_{t_T}$

In this section, we derive an upper bound on the average forgetting, as presented in Equation (3).

*Proof.* We decompose each term in Equation (3) as follows:

For any $\tau \in [T-1]$, we have:

$$L_{\mathcal{D}_\tau}(f^*_T) - L_{\mathcal{D}_\tau}(f^*_\tau) = \underbrace{L_{\mathcal{D}_\tau}(f^*_T) - L_{S_\tau}(f^*_T)}_{(a)} + \underbrace{L_{S_\tau}(f^*_T) - L_{S_\tau}(f^*_\tau)}_{(b)} + \underbrace{L_{S_\tau}(f^*_\tau) - L_{\mathcal{D}_\tau}(f^*_\tau)}_{(c)}. \tag{61}$$

Next, we derive upper bounds for terms (a), (b), and (c), respectively.

For term (a), by applying Theorem 2 together with the bound in Equation (58), we obtain:

$$\begin{aligned} L_{\mathcal{D}_\tau}(f^*_T) - L_{S_\tau}(f^*_T) &\leq 2\rho\mathcal{R}_{S_\tau}(\mathcal{F}_T) + 3c\sqrt{\frac{log(2/\delta)}{2n_\tau}} \\ &\leq 2\rho \sum_{k=1}^{T} \frac{[\mathrm{Tr}(K_k(X_\tau, X_\tau))\tilde{Y}_k^\top E_{k,t^*_k} K_k(X_k, X_k)^{-1}\tilde{Y}_k]^{1/2}}{n_\tau} + 3c\sqrt{\frac{log(2/\delta)}{2n_\tau}}. \end{aligned} \tag{62}$$

For term (b), we begin by deriving explicit expressions for $L_{S_\tau}(f_T^*)$ and $L_{S_\tau}(f_\tau^*)$, as follows:

$$L_{S_\tau}(f_T^*) \leq \frac{1}{n_\tau}\|f_\tau^*(X_\tau) - Y_\tau\|^2 + \frac{1}{n_\tau}\|\sum_{k=\tau+1}^{T} \tilde{f}_k^*(X_\tau)\|^2, \tag{63}$$

$$L_{S_\tau}(f_\tau^*) = \frac{1}{2n_\tau}\|f_\tau^*(X_\tau) - Y_\tau\|^2. \tag{64}$$

Subtracting the two expressions, we obtain:

$$
\begin{aligned}
&L_{S_\tau}(f_T^*) - L_{S_\tau}(f_\tau^*) \\
&\leq \frac{1}{n_\tau}\|\sum_{k=\tau+1}^{T} \tilde{f}_k^*(X_\tau)\|^2 + \frac{1}{2n_\tau}\|f_\tau^*(X_\tau) - Y_\tau\|^2 \\
&\leq \frac{T-\tau}{n_\tau}\sum_{k=\tau+1}^{T}\|K_k(X_\tau, X_k)E_{k,t_k^*}K_k(X_k, X_k)^{-1}\tilde{Y}_k\|^2 + \frac{1}{2n_\tau}\|e^{-\frac{t_\tau^*}{n_\tau}K_\tau(X_\tau, X_\tau)}\tilde{Y}_\tau\|^2.
\end{aligned}
\tag{65}
$$

For term (c), by applying Corollary 1 together with the bound in Equation (58), we obtain:

$$
\begin{aligned}
&L_{S_\tau}(f_\tau^*) - L_{\mathcal{D}_\tau}(f_\tau^*) \\
&\leq 2\rho\mathcal{R}_{S_\tau}(\mathcal{F}_\tau) + 3c\sqrt{\frac{log(2/\delta)}{2n_\tau}} \\
&\leq 2\rho\sum_{k=1}^{\tau}\frac{[\mathrm{Tr}(K_k(X_\tau, X_\tau))\tilde{Y}_k^\top E_{k,t_k^*}K_k(X_k, X_k)^{-1}\tilde{Y}_k]^{1/2}}{n_\tau} + 3c\sqrt{\frac{log(2/\delta)}{2n_\tau}}.
\end{aligned}
\tag{66}
$$

Then, we have:

$$
\begin{aligned}
F_{t_T} \leq &\frac{1}{T-1}\sum_{\tau=1}^{T-1}\left\{2\rho\sum_{k=1}^{T}\frac{[\mathrm{Tr}(K_k(X_\tau, X_\tau))\tilde{Y}_k^\top E_{k,t_k^*}K_k(X_k, X_k)^{-1}\tilde{Y}_k]^{1/2}}{n_\tau}\right. \\
&+ 2\rho\sum_{k=1}^{\tau}\frac{[\mathrm{Tr}(K_k(X_\tau, X_\tau))\tilde{Y}_k^\top E_{k,t_k^*}K_k(X_k, X_k)^{-1}\tilde{Y}_k]^{1/2}}{n_\tau} + 6c\sqrt{\frac{log(2/\delta)}{2n_\tau}} \\
&\left.+ \frac{T-\tau}{n_\tau}\sum_{k=\tau+1}^{T}\|K_k(X_\tau, X_k)E_{k,t_k^*}K_k(X_k, X_k)^{-1}\tilde{Y}_k\|^2 + \frac{1}{2n_\tau}\|e^{-\frac{t_\tau^*}{n_\tau}K_\tau(X_\tau, X_\tau)}\tilde{Y}_\tau\|^2\right\}.
\end{aligned}
\tag{67}
$$

$\square$

## F  FORGETTING AND GENERALIZATION ERROR OF PGN

We first present the main result in Theorem 3. To help readers quickly understand the proof strategy, we also provide a proof sketch. The detailed proofs are given in Appendix F.1, Appendix F.2, Appendix F.3, and Appendix F.4.

**Theorem 3.** *Consider a sequence of $T$ tasks. For each task $\tau \in [T]$, let $\mathcal{D}_\tau$ denote the data distribution, and let $S_\tau = \{X_\tau, Y_\tau\}$ be the corresponding training dataset drawn i.i.d. from $\mathcal{D}_\tau$. Suppose the loss function $\ell(\cdot, \cdot)$ takes values in the interval $[0, c]$ and is $\rho$-Lipschitz in the first argument. Then, with probability at least $1 - \delta$, the following bounds hold:*

$$
\begin{aligned}
F_{t_T} \leq &\frac{1}{T-1} \sum_{\tau=1}^{T-1} \left\{ 2\rho \sum_{k=1}^{T} \frac{\left[ \operatorname{Tr}(K_k(X_\tau, X_\tau)) \tilde{Y}_k^\top (E_{k,t_k^*}^{\mathrm{PGN}})^2 K_k(X_k, X_k)^{-1} \tilde{Y}_k \right]^{1/2}}{n_\tau} \right.\\
&+ 2\rho \sum_{k=1}^{\tau} \frac{\left[ \operatorname{Tr}(K_k(X_\tau, X_\tau)) \tilde{Y}_k^\top (E_{k,t_k^*}^{\mathrm{PGN}})^2 K_k(X_k, X_k)^{-1} \tilde{Y}_k \right]^{1/2}}{n_\tau} + 6c\sqrt{\frac{log(2/\delta)}{2n_\tau}}\\
&+ \frac{T-\tau}{n_\tau} \sum_{k=\tau+1}^{T} \| K_k(X_\tau, X_k) E_{k,t_k^*}^{\mathrm{PGN}} K_k(X_k, X_k)^{-1} \tilde{Y}_k \|^2\\
&\left. + \frac{1}{2n_\tau} \| e^{-\frac{t_\tau^*}{n_\tau} K_\tau(X_\tau, X_\tau) - \frac{\Phi_\tau(t_\tau^*)}{n_\tau} [K_\tau(X_\tau, X_\tau)]^2} \tilde{Y}_\tau \|^2 \right\},
\end{aligned} \tag{68}
$$

$$
\begin{aligned}
G_{t_T} \leq &\frac{1}{T} \sum_{\tau=1}^{T} \left\{ \frac{T-\tau}{n_\tau} \sum_{k=\tau+1}^{T} \| K_k(X_\tau, X_k) E_{k,t_k^*}^{\mathrm{PGN}} K_k(X_k, X_k)^{-1} \tilde{Y}_k \|^2 \right.\\
&+ \frac{1}{n_\tau} \| e^{-\frac{t_\tau^*}{n_\tau} K_\tau(X_\tau, X_\tau) - \frac{\Phi_\tau(t_\tau^*)}{n_\tau} [K_\tau(X_\tau, X_\tau)]^2} \tilde{Y}_\tau \|^2\\
&\left. + 2\rho \sum_{k=1}^{T} \frac{[\operatorname{Tr}(K_k(X_\tau, X_\tau)) \tilde{Y}_k^\top (E_{k,t_k^*}^{\mathrm{PGN}})^2 K_k(X_k, X_k)^{-1} \tilde{Y}_k]^{1/2}}{n_\tau} + 3c\sqrt{\frac{log(2/\delta)}{2n_\tau}} \right\},
\end{aligned} \tag{69}
$$

*where $E_{\tau,t}^{\mathrm{PGN}} = I - \exp\left( -\frac{t}{n_\tau} K_\tau(X_\tau, X_\tau) - \frac{\Phi_\tau(t)}{n_\tau} K_\tau^2(X_\tau, X_\tau) \right)$ and $\Phi_\tau(t)$ satisfy $\Phi_\tau(t) = \int_0^t \frac{\alpha_\tau}{\sqrt{[f_\tau^s t(X_\tau) - Y_\tau]^\top K_\tau(X_\tau, X_\tau) [f_\tau^s(X_\tau) - Y_\tau]}} ds$.*

*proof sketch.* Our proof consists of four main parts.

(1) *Gradient flow of PGN.* We first compute the gradient of the PGN loss in Equation (15) and apply the chain rule $\frac{d}{dt} f_\tau^t(x) = \nabla_{\theta_\tau^t} f_\tau^t(x) \frac{d\theta_\tau^t}{dt}$ to derive the kernel gradient flow of PGN:

$$
\frac{d}{dt} f_\tau^t(x) = -\frac{1}{n_\tau} K_\tau(x, X_\tau) \big(f_\tau^t(X_\tau) - Y_\tau\big) - \frac{\alpha_\tau}{n_\tau} \frac{K_\tau(x, X_\tau) K_\tau(X_\tau, X_\tau) \big(f_\tau^t(X_\tau) - Y_\tau\big)}{\sqrt{\big(f_\tau^t(X_\tau) - Y_\tau\big)^\top K_\tau(X_\tau, X_\tau) \big(f_\tau^t(X_\tau) - Y_\tau\big)}}. \tag{70}
$$

The detailed derivation is provided in Appendix F.1.

(2) *Solution of the kernel gradient flow for PGN.* We adopt a similar approach to Appendix D.1 to solve the ODE. We first derive the solution on the training set $X_\tau$:

$$
f_\tau^t(X_\tau) = Y_\tau + \exp\left( -\frac{t}{n_\tau} K_\tau(X_\tau, X_\tau) - \frac{\Phi_\tau(t)}{n_\tau} K_\tau^2(X_\tau, X_\tau) \right) \big(f_\tau^0(X_\tau) - Y_\tau\big). \tag{71}
$$

We then obtain the solution at an arbitrary point $x$:

$$
f_\tau^t(x) = f_{\tau-1}^*(x) - K_\tau(x, X_\tau) E_{\tau,t}^{\mathrm{PGN}} K_\tau^{-1}(X_\tau, X_\tau) \big(f_{\tau-1}^*(X_\tau) - Y_\tau\big). \tag{72}
$$

The detailed derivation is provided in Appendix F.2.

(3) *Bound on the generalization error.* We use standard techniques from statistical learning theory to bound the generalization error of the regularization based method via Rademacher complexity (Kakade et al., 2008; Cortes et al., 2010). According to Theorem 2, we need to control the empirical loss $L_{S_\tau}(f_T^*)$ and the Rademacher complexity of the function class $\mathcal{R}_{S_\tau}(\mathcal{F}_T)$.

The empirical loss $L_{S_\tau}(f_T^*)$ can be bounded using Equation (72), which yields

$$
\begin{aligned}
L_{S_\tau}(f_T^*) \leq & \frac{T-\tau}{n_\tau} \sum_{k=\tau+1}^{T} \left\| K_k(X_\tau, X_k) E_{k,t_k^*}^{\mathrm{PGN}} K_k(X_k, X_k)^{-1} \tilde{Y}_k \right\|^2 \\
& + \frac{1}{n_\tau} \left\| e^{-\frac{t_\tau^*}{n_\tau} K_\tau(X_\tau, X_\tau) - \frac{\Phi_\tau(t)}{n_\tau} [K_\tau(X_\tau, X_\tau)]^2} \tilde{Y}_\tau \right\|^2.
\end{aligned}
\tag{73}
$$

We bound the Rademacher complexity through the RKHS norm of $\tilde{f}_\tau^*$ and Lemma 6:

$$
\mathcal{R}_{S_\tau}(\mathcal{F}_T) \leq \sum_{k=1}^{T} \frac{\left[ \mathrm{Tr}\left( K_k(X_\tau, X_\tau) \right) \tilde{Y}_k^\top (E_{k,t_k^*}^{\mathrm{PGN}})^2 K_k(X_k, X_k)^{-1} \tilde{Y}_k \right]^{1/2}}{n_\tau}.
\tag{74}
$$

By combining Equations (73) and (74) with Theorem 2, we obtain the desired upper bound on the generalization error $G_{t_T}$. The detailed derivation is provided in Appendix F.3.

(4) *Bound on forgetting.* We decompose each term in the forgetting metric in Equation (3) as

$$
L_{\mathcal{D}_\tau}(f_T^*) - L_{\mathcal{D}_\tau}(f_\tau^*) = \underbrace{L_{\mathcal{D}_\tau}(f_T^*) - L_{S_\tau}(f_T^*)}_{(a)} + \underbrace{L_{S_\tau}(f_T^*) - L_{S_\tau}(f_\tau^*)}_{(b)} + \underbrace{L_{S_\tau}(f_\tau^*) - L_{\mathcal{D}_\tau}(f_\tau^*)}_{(c)},
\tag{75}
$$

for any $\tau \in [T-1]$.

Theorem 2 and Corollary 1 imply that terms $(a)$ and $(c)$ are controlled by the Rademacher complexities $\mathcal{R}_{S_\tau}(\mathcal{F}_T)$ and $\mathcal{R}_{S_\tau}(\mathcal{F}_\tau)$ respectively. The bounds on $\mathcal{R}_{S_\tau}(\mathcal{F}_T)$ and $\mathcal{R}_{S_\tau}(\mathcal{F}_\tau)$ have already been obtained in step (3). The second term $(b)$ can be bounded as

$$
\begin{aligned}
L_{S_\tau}(f_T^*) - L_{S_\tau}(f_\tau^*) \leq & \frac{1}{n_\tau} \left\| \sum_{k=\tau+1}^{T} \tilde{f}_k^*(X_\tau) \right\|^2 + \frac{1}{2n_\tau} \| f_\tau^*(X_\tau) - Y_\tau \|^2 \\
\leq & \frac{T-\tau}{n_\tau} \sum_{k=\tau+1}^{T} \left\| K_k(X_\tau, X_k) E_{k,t_k^*}^{\mathrm{PGN}} K_k(X_k, X_k)^{-1} \tilde{Y}_k \right\|^2 \\
& + \frac{1}{2n_\tau} \left\| e^{-\frac{t_\tau^*}{n_\tau} K_\tau(X_\tau, X_\tau) - \frac{\Phi_\tau(t_\tau^*)}{n_\tau} [K_\tau(X_\tau, X_\tau)]^2} \tilde{Y}_\tau \right\|^2.
\end{aligned}
\tag{76}
$$

Combining these bounds yields the desired upper bound on the average forgetting. The detailed derivation is provided in Appendix F.4.

$\square$

## F.1 KERNEL GRADIENT FLOW OF PGN

In the CL setting, the training loss of PGN for any task $\tau \in [T]$ is given by

$$
L_{S_\tau}^{PGN}(\theta_\tau) = L_{S_\tau}(\theta_\tau) + \alpha_\tau \| \nabla_{\theta_\tau} L_{S_\tau}(\theta_\tau) \|,
\tag{77}
$$

Therefore, we have

$$
\nabla_\theta L_{S_\tau}^{PGN}(\theta_\tau) = \nabla_{\theta_\tau} L_{S_\tau}(\theta_\tau) + \alpha_\tau \nabla_{\theta_\tau}^2 L_{S_\tau}(\theta_\tau) \frac{\nabla_{\theta_\tau} L_{S_\tau}(\theta_\tau)}{\| \nabla_{\theta_\tau} L_{S_\tau}(\theta_\tau) \|_2}.
\tag{78}
$$

Based on Equation (5), we have

$$\nabla_{\theta_\tau} L_{S_\tau}(\theta_\tau^t) = \frac{1}{n_\tau} [\nabla_{\theta_\tau} f_\tau^t(X_\tau)]^\top \big(f_\tau^t(X_\tau) - Y_\tau\big).$$

For any task $\tau \in [T]$, the parameter $\theta_\tau$ evolves according to the differential equation

$$\begin{aligned}
\frac{d\theta_\tau^t}{dt} = & - \nabla_{\theta_\tau^t} L_{S_\tau}(\theta_\tau^t) - \alpha_\tau \nabla_{\theta_\tau^t}^2 L_{S_\tau}(\theta_\tau^t) \frac{\nabla_{\theta_\tau^t} L_{S_\tau}(\theta_\tau^t)}{\|\nabla_{\theta_\tau^t} L_{S_\tau}(\theta_\tau^t)\|_2} \\
= & - \frac{1}{n_\tau} [\nabla_{\theta_\tau^t} f_\tau^t(X_\tau)]^\top \big(f_\tau^t(X_\tau) - Y_\tau\big) - \alpha_\tau \nabla_{\theta_\tau^t}^2 L_{S_\tau}(\theta_\tau^t) \frac{\frac{1}{n_\tau} [\nabla_{\theta_\tau^t} f_\tau^t(X_\tau)]^\top \big(f_\tau^t(X_\tau) - Y_\tau\big)}{\left\| \frac{1}{n_\tau} [\nabla_{\theta_\tau^t} f_\tau^t(X_\tau)]^\top \big(f_\tau^t(X_\tau) - Y_\tau\big) \right\|},
\end{aligned}$$
(79)

where $t \geq 0$ denotes continuous time.

Under the NTK linearization, we simplify

$$\nabla_{\theta_\tau}^2 L_{S_\tau}(\theta_\tau) = [\nabla_{\theta_\tau} f_\tau^t(X_\tau)]^\top \frac{1}{n_\tau} I \, \nabla_{\theta_\tau} f_\tau^t(X_\tau) = \frac{1}{n_\tau} [\nabla_{\theta_\tau} f_\tau^t(X_\tau)]^\top \nabla_{\theta_\tau} f_\tau^t(X_\tau).$$

Therefore, we have

$$\begin{aligned}
\frac{d\theta_\tau^t}{dt} = & - \frac{1}{n_\tau} [\nabla_{\theta_\tau^t} f_\tau^t(X_\tau)]^\top \big(f_\tau^t(X_\tau) - Y_\tau\big) \\
& - \frac{\alpha_\tau}{n_\tau} [\nabla_{\theta_\tau^t} f_\tau^t(X_\tau)]^\top \nabla_{\theta_\tau^t} f_\tau^t(X_\tau) \frac{[\nabla_{\theta_\tau^t} f_\tau^t(X_\tau)]^\top \big(f_\tau^t(X_\tau) - Y_\tau\big)}{\sqrt{\big(f_\tau^t(X_\tau) - Y_\tau\big)^\top K_\tau(X_\tau, X_\tau)\big(f_\tau^t(X_\tau) - Y_\tau\big)}},
\end{aligned}$$
(80)

where we used $\left\| [\nabla_{\theta_\tau^t} f_\tau^t(X_\tau)]^\top \big(f_\tau^t(X_\tau) - Y_\tau\big) \right\| = \sqrt{\big(f_\tau^t(X_\tau) - Y_\tau\big)^\top K_\tau(X_\tau, X_\tau)\big(f_\tau^t(X_\tau) - Y_\tau\big)}$.

Based on the chain rule, we have

$$\begin{aligned}
\frac{d}{dt} f_\tau^t(x) = & \nabla_{\theta_\tau^t} f_\tau^t(x) \frac{d\theta_\tau^t}{dt} \\
= & - \frac{1}{n_\tau} \nabla_{\theta_\tau^t} f_\tau^t(x) [\nabla_{\theta_\tau^t} f_\tau^t(X_\tau)]^\top \big(f_\tau^t(X_\tau) - Y_\tau\big) \\
& - \frac{\alpha_\tau}{n_\tau} \frac{\nabla_{\theta_\tau^t} f_\tau^t(x) [\nabla_{\theta_\tau^t} f_\tau^t(X_\tau)]^\top \nabla_{\theta_\tau^t} f_\tau^t(X_\tau) [\nabla_{\theta_\tau^t} f_\tau^t(X_\tau)]^\top \big(f_\tau^t(X_\tau) - Y_\tau\big)}{\sqrt{\big(f_\tau^t(X_\tau) - Y_\tau\big)^\top K_\tau(X_\tau, X_\tau)\big(f_\tau^t(X_\tau) - Y_\tau\big)}}.
\end{aligned}$$
(81)

Therefore, under the NTK regime, the kernel gradient flow takes the following form:

$$\frac{d}{dt} f_\tau^t(x) = -\frac{1}{n_\tau} K_\tau(x, X_\tau)\big(f_\tau^t(X_\tau) - Y_\tau\big) - \frac{\alpha_\tau}{n_\tau} \frac{K_\tau(x, X_\tau) K_\tau(X_\tau, X_\tau)\big(f_\tau^t(X_\tau) - Y_\tau\big)}{\sqrt{\big(f_\tau^t(X_\tau) - Y_\tau\big)^\top K_\tau(X_\tau, X_\tau)\big(f_\tau^t(X_\tau) - Y_\tau\big)}},$$
(82)

where $K_\tau(x, x') = \langle \nabla_{\theta_{\tau-1}^*} f_{\tau-1}^*(x), \nabla_{\theta_{\tau-1}^*} f_{\tau-1}^*(x') \rangle$.

### F.2 SOLUTION OF KERNEL GRADIENT FLOW FOR PGN

We follow the approach in Appendix D.1: first derive the solution on the training set $X_\tau$, then extend it to an arbitrary input $x$.

**(1) Solution on the training set $X_\tau$.** Evaluating Equation (82) at $X_\tau$ gives

$$\frac{d}{dt} f_\tau^t(X_\tau) = -\frac{1}{n_\tau} K_\tau(X_\tau, X_\tau)\big(f_\tau^t(X_\tau) - Y_\tau\big) - \frac{\alpha_\tau}{n_\tau} \frac{K_\tau(X_\tau, X_\tau)^2 \big(f_\tau^t(X_\tau) - Y_\tau\big)}{\sqrt{\big(f_\tau^t(X_\tau) - Y_\tau\big)^\top K_\tau(X_\tau, X_\tau)\big(f_\tau^t(X_\tau) - Y_\tau\big)}}.$$
(83)

Let $g(t) = f_\tau^t(X_\tau) - Y_\tau$ and abbreviate $K_\tau := K_\tau(X_\tau, X_\tau)$. Then Equation (83) becomes the matrix ODE

$$\frac{d}{dt}g(t) = -\frac{1}{n_\tau}K_\tau\, g(t) - \frac{\alpha_\tau}{n_\tau}\frac{K_\tau^2\, g(t)}{\sqrt{g(t)^\top K_\tau g(t)}}, \qquad g(0) = f_\tau^0(X_\tau) - Y_\tau. \tag{84}$$

Since $K_\tau$ is real symmetric positive semidefinite and $K_\tau$ commutes with $K_\tau^2$, the theory of linear time-varying ODEs with commuting coefficients yields the solution

$$g(t) = \exp\Big(-\frac{t}{n_\tau}K_\tau - \frac{\Phi_\tau(t)}{n_\tau}K_\tau^2\Big)g(0), \qquad \Phi_\tau(t) = \int_0^t \frac{\alpha_\tau}{\sqrt{g(s)^\top K_\tau g(s)}}ds. \tag{85}$$

Substituting $g(t) = f_\tau^t(X_\tau) - Y_\tau$ into Equation (85) yields

$$f_\tau^t(X_\tau) = Y_\tau + \exp\Big(-\frac{t}{n_\tau}K_\tau - \frac{\Phi_\tau(t)}{n_\tau}K_\tau^2\Big)\big(f_\tau^0(X_\tau) - Y_\tau\big). \tag{86}$$

**(2) Solution at an arbitrary point $x$.** For any $x \in \mathbb{R}^d$, Equation (82) can be rewritten as

$$\frac{d}{dt}f_\tau^t(x) = -\frac{1}{n_\tau}K_\tau(x, X_\tau)\, g(t) - \frac{\alpha_\tau}{n_\tau}\frac{K_\tau(x, X_\tau)\, K_\tau\, g(t)}{\sqrt{g(t)^\top K_\tau g(t)}}. \tag{87}$$

Notably, multiplying $K_\tau(x, X_\tau)\, K_\tau^{-1}$ on both sides of Equation (84) gives

$$K_\tau(x, X_\tau)\, K_\tau^{-1}\frac{d}{dt}g(t) = -\frac{1}{n_\tau}K_\tau(x, X_\tau)\, g(t) - \frac{\alpha_\tau}{n_\tau}\frac{K_\tau(x, X_\tau)\, K_\tau\, g(t)}{\sqrt{g(t)^\top K_\tau g(t)}} = \frac{d}{dt}f_\tau^t(x). \tag{88}$$

Integrating Equation (88) over $[0, t]$ yields

$$f_\tau^t(x) - f_\tau^0(x) = K_\tau(x, X_\tau)\, K_\tau^{-1}\big(g(t) - g(0)\big). \tag{89}$$

Substituting Equation (85) into Equation (89) and using $f_\tau^0(x) = f_{\tau-1}^*(x)$ gives the closed form

$$\begin{aligned}
f_\tau^t(x) &= f_{\tau-1}^*(x) + K_\tau(x, X_\tau)\, K_\tau^{-1}\Big(\exp\Big(-\frac{t}{n_\tau}K_\tau - \frac{\Phi_\tau(t)}{n_\tau}K_\tau^2\Big) - I\Big)\big(f_{\tau-1}^*(X_\tau) - Y_\tau\big) \\
&= f_{\tau-1}^*(x) - K_\tau(x, X_\tau)\, E_{\tau,t}^{\text{PGN}}\, K_\tau^{-1}\big(f_{\tau-1}^*(X_\tau) - Y_\tau\big),
\end{aligned} \tag{90}$$

where $E_{\tau,t}^{\text{PGN}} := I - \exp\Big(-\frac{t}{n_\tau}K_\tau - \frac{\Phi_\tau(t)}{n_\tau}K_\tau^2\Big)$. Therefore, we obtain a form for PGN that is analogous to the SGD solution in Equation (11). The only difference is that PGN uses $E_{\tau,t}^{\text{PGN}}$, whereas SGD uses $E_{\tau,t}$. For notational convenience, define

$$\tilde{f}_\tau^t(x) := K_\tau(x, X_\tau)\, E_{\tau,t}^{\text{PGN}}\, K_\tau(X_\tau, X_\tau)^{-1}\, \tilde{Y}_\tau. \tag{91}$$

Therefore, the predictor for task $\tau$ can be written as $f_\tau^t(x) = \sum_{i=1}^{\tau-1}\tilde{f}_i^*(x) + \tilde{f}_\tau^t(x)$. To simplify the notation, we adopt the notational convention $t_T^* = t_T$ for the final task $T$, that is, the iteration index of task $T$ coincides with its stopping iteration.

### F.3 BOUND ON THE GENERALIZATION ERROR

In this section we derive an upper bound on the generalization error. In particular, we bound the population loss $L_{D_\tau}(f_T^*)$ using the Rademacher complexity of the hypothesis class, the empirical loss $L_{S_\tau}(f_T^*)$, and appropriate constants as shown in Theorem 2. This approach follows standard statistical techniques for regularization based methods as in Kakade et al. (2008); Cortes et al. (2010). Specifically, the regularization term affects the bound through its influence on the Rademacher complexity of the function class.

(1) For the term $L_{S_\tau}(f_T^*)$ for any $\tau \in [T]$, we have:

$$\begin{aligned}
L_{S_\tau}(f_T^*) &= \frac{1}{2n_\tau}\|f_T^*(X_\tau) - Y_\tau\|^2 \\
&= \frac{1}{2n_\tau}\|f_\tau^*(X_\tau) + \sum_{k=\tau+1}^{T} \tilde{f}_k^*(X_\tau) - Y_\tau\|^2.
\end{aligned} \tag{92}$$

Next, we compute the term $\|f_\tau^*(X_\tau) - Y_\tau\|^2$ as follows:

$$\begin{aligned}
&\|f_\tau^*(X_\tau) - Y_\tau\|^2 \\
=&\|f_{\tau-1}^*(X_\tau) + \tilde{f}_\tau^*(X_\tau) - Y_\tau\|^2 \\
=&\|\tilde{f}_\tau^*(X_\tau) - \tilde{Y}_\tau\|^2 \\
=&\|K_\tau(X_\tau, X_\tau)E_{\tau,t_\tau^*}^{\mathrm{PGN}}K_\tau(X_\tau, X_\tau)^{-1}\tilde{Y}_\tau - \tilde{Y}_\tau\|^2 \\
=&\|\tilde{Y}_\tau - K_\tau(X_\tau, X_\tau)e^{-\frac{t_\tau^*}{n_\tau}K_\tau(X_\tau,X_\tau) - \frac{\Phi_\tau(t_\tau^*)}{n_\tau}[K_\tau(X_\tau,X_\tau)]^2}K_\tau(X_\tau, X_\tau)^{-1}\tilde{Y}_\tau - \tilde{Y}_\tau\|^2 \\
=&\|K_\tau(X_\tau, X_\tau)e^{-\frac{t_\tau^*}{n_\tau}K_\tau(X_\tau,X_\tau) - \frac{\Phi_\tau(t_\tau^*)}{n_\tau}[K_\tau(X_\tau,X_\tau)]^2}K_\tau(X_\tau, X_\tau)^{-1}\tilde{Y}_\tau\|^2 \\
=&\|e^{-\frac{t_\tau^*}{n_\tau}K_\tau(X_\tau,X_\tau) - \frac{\Phi_\tau(t_\tau^*)}{n_\tau}[K_\tau(X_\tau,X_\tau)]^2}\tilde{Y}_\tau\|^2.
\end{aligned} \tag{93}$$

Therefore, we obtain the following upper bound:

$$\begin{aligned}
L_{S_\tau}(f_T^*) \leq &\frac{T-\tau}{n_\tau}\sum_{k=\tau+1}^{T}\|K_k(X_\tau, X_k)E_{k,t_k^*}^{\mathrm{PGN}}K_k(X_k, X_k)^{-1}\tilde{Y}_k\|^2 \\
&+ \frac{1}{n_\tau}\|e^{-\frac{t_\tau^*}{n_\tau}K_\tau(X_\tau,X_\tau) - \frac{\Phi_\tau(t_\tau^*)}{n_\tau}[K_\tau(X_\tau,X_\tau)]^2}\tilde{Y}_\tau\|^2.
\end{aligned} \tag{94}$$

(2) For the term $\mathcal{R}(\mathcal{F}_T)$, we first consider a bound on the reproduced kernel Hilbert space (RKHS) norm of $\tilde{f}_\tau^*$. Let $(\mathcal{H}_{K_\tau}, \|\cdot\|_{\mathcal{H}_{K_\tau}})$ be the RKHS induced by the kernel $K_\tau$. We define

$$\hat{\alpha}_\tau := E_{\tau,t_\tau^*}^{\mathrm{PGN}}K_\tau(X_\tau, X_\tau)^{-1}\tilde{Y}_\tau. \tag{95}$$

Then, $\tilde{f}_\tau^*$ can be written as:

$$\tilde{f}_\tau^*(x) = K_\tau(x, X_\tau)\hat{\alpha}_\tau. \tag{96}$$

The RKHS norm of $\tilde{f}_\tau^*$ is then given by:

$$\begin{aligned}
\|\tilde{f}_\tau^*\|_{\mathcal{H}_{K_\tau}}^2 &= \hat{\alpha}_\tau^\top K_\tau(X_\tau, X_\tau)\hat{\alpha}_\tau \\
&= \tilde{Y}_\tau^\top K_\tau(X_\tau, X_\tau)^{-1}E_{\tau,t_\tau^*}^{\mathrm{PGN}}K_\tau(X_\tau, X_\tau)E_{\tau,t_\tau^*}^{\mathrm{PGN}}K_\tau(X_\tau, X_\tau)^{-1}\tilde{Y}_\tau \\
&= \tilde{Y}_\tau^\top (E_{\tau,t_\tau^*}^{\mathrm{PGN}})^2 K_\tau(X_\tau, X_\tau)^{-1}\tilde{Y}_\tau := \hat{B}_\tau^2.
\end{aligned} \tag{97}$$

We define the function class $\mathcal{F}_T$ as follows:

$$\mathcal{F}_T = \left\{ f : \mathcal{X} \to \mathbb{R} \,\middle|\, x \to \sum_{\tau=1}^{T}\tilde{f}_\tau(x), \tilde{f}_\tau(x) = K_\tau(x, X_\tau)\hat{\alpha}_\tau, \|\tilde{f}_\tau\|_{\mathcal{H}_{K_\tau}} \leq \hat{B}_\tau, \forall \tau \in [T] \right\}. \tag{98}$$

Based on Lemma 6, we obtain the following bound on the empirical Rademacher complexity of $\mathcal{F}_T$ over $S_\tau$:

$$\begin{aligned}
\mathcal{R}_{S_\tau}(\mathcal{F}_T) &\leq \sum_{k=1}^{T}\frac{\hat{B}_k}{n_\tau}(\mathrm{Tr}(K_k(X_\tau, X_\tau)))^{1/2} \\
&\leq \sum_{k=1}^{T}\frac{[\mathrm{Tr}(K_k(X_\tau, X_\tau))\tilde{Y}_k^\top(E_{k,t_k^*}^{\mathrm{PGN}})^2 K_k(X_k, X_k)^{-1}\tilde{Y}_k]^{1/2}}{n_\tau}.
\end{aligned} \tag{99}$$

Based on Theorem 2, we have:

$$L_{\mathcal{D}_\tau}(f_T^*) \leq \frac{T-\tau}{n_\tau} \sum_{k=\tau+1}^{T} \|K_k(X_\tau, X_k) E_{k,t_k^*}^{\mathrm{PGN}} K_k(X_k, X_k)^{-1} \tilde{Y}_k\|^2$$
$$+ \frac{1}{n_\tau} \|e^{-\frac{t_\tau^*}{n_\tau} K_\tau(X_\tau, X_\tau) - \frac{\Phi_\tau(t_\tau^*)}{n_\tau} [K_\tau(X_\tau, X_\tau)]^2} \tilde{Y}_\tau\|^2$$
$$+ 2\rho \sum_{k=1}^{T} \frac{[\mathrm{Tr}(K_k(X_\tau, X_\tau)) \tilde{Y}_k^\top (E_{k,t_k^*}^{\mathrm{PGN}})^2 K_k(X_k, X_k)^{-1} \tilde{Y}_k]^{1/2}}{n_\tau} + 3c\sqrt{\frac{log(2/\delta)}{2n_\tau}}. \tag{100}$$

By substituting the bound from Equation (100) into Equation (4), we obtain:

$$G_T \leq \frac{1}{T} \sum_{\tau=1}^{T} \left\{ \frac{T-\tau}{n_\tau} \sum_{k=\tau+1}^{T} \|K_k(X_\tau, X_k) E_{k,t_k^*}^{\mathrm{PGN}} K_k(X_k, X_k)^{-1} \tilde{Y}_k\|^2 \right.$$
$$+ \frac{1}{n_\tau} \|e^{-\frac{t_\tau^*}{n_\tau} K_\tau(X_\tau, X_\tau) - \frac{\Phi_\tau(t_\tau^*)}{n_\tau} [K_\tau(X_\tau, X_\tau)]^2} \tilde{Y}_\tau\|^2 \tag{101}$$
$$\left. + 2\rho \sum_{k=1}^{T} \frac{[Tr(K_k(X_\tau, X_\tau)) \tilde{Y}_k^\top (E_{k,t_k^*}^{\mathrm{PGN}})^2 K_k(X_k, X_k)^{-1} \tilde{Y}_k]^{1/2}}{n_\tau} + 3c\sqrt{\frac{log(2/\delta)}{2n_\tau}} \right\}.$$

### F.4 Bound on forgetting

We decompose each term in Equation (3) as follows:

For any $\tau \in [T-1]$, we have:

$$L_{\mathcal{D}_\tau}(f_T^*) - L_{\mathcal{D}_\tau}(f_\tau^*) = \underbrace{L_{\mathcal{D}_\tau}(f_T^*) - L_{S_\tau}(f_T^*)}_{(a)} + \underbrace{L_{S_\tau}(f_T^*) - L_{S_\tau}(f_\tau^*)}_{(b)} + \underbrace{L_{S_\tau}(f_\tau^*) - L_{\mathcal{D}_\tau}(f_\tau^*)}_{(c)}. \tag{102}$$

Next, we derive upper bounds for terms (a), (b), and (c), respectively.

For term (a), by applying Theorem 2 together with the bound in Equation (99), we obtain:

$$L_{D_\tau}(f_T^*) - L_{S_\tau}(f_T^*)$$
$$\leq 2\rho \mathcal{R}_{S_\tau}(\mathcal{F}_T) + 3c\sqrt{\frac{log(2/\delta)}{2n_\tau}}$$
$$\leq 2\rho \sum_{k=1}^{T} \frac{[Tr(K_k(X_\tau, X_\tau)) \tilde{Y}_k^\top (E_{k,t_k^*}^{\mathrm{PGN}})^2 K_k(X_k, X_k)^{-1} \tilde{Y}_k]^{1/2}}{n_\tau} + 3c\sqrt{\frac{log(2/\delta)}{2n_\tau}}. \tag{103}$$

For term (b), we begin by deriving explicit expressions for $L_{S_\tau}(f_T^*)$ and $L_{S_\tau}(f_\tau^*)$, as follows:

$$L_{S_\tau}(f_T^*) \leq \frac{1}{n_\tau} \|f_\tau^*(X_\tau) - Y_\tau\|^2 + \frac{T-\tau}{n_\tau} \sum_{k=\tau+1}^{T} \|\tilde{f}_k^*(X_\tau)\|^2, \tag{104}$$

$$L_{S_\tau}(f_\tau^*) = \frac{1}{2n_\tau} \|f_\tau^*(X_\tau) - Y_\tau\|^2. \tag{105}$$

Subtracting the two expressions, we obtain:

$$L_{S_\tau}(f_T^*) - L_{S_\tau}(f_\tau^*)$$

$$\leq \frac{T-\tau}{n_\tau} \sum_{k=\tau+1}^{T} \|\tilde{f}_k^*(X_\tau)\|^2 + \frac{1}{2n_\tau} \|f_\tau^*(X_\tau) - Y_\tau\|^2$$

$$\leq \frac{T-\tau}{n_\tau} \sum_{k=\tau+1}^{T} \|K_k(X_\tau, X_k) E_{k,t_k^*}^{\text{PGN}} K_k(X_k, X_k)^{-1} \tilde{Y}_k\|^2 \tag{106}$$

$$+ \frac{1}{2n_\tau} \|e^{-\frac{t_\tau^*}{n_\tau} K_\tau(X_\tau, X_\tau) - \frac{\Phi_\tau(t_\tau^*)}{n_\tau} [K_\tau(X_\tau, X_\tau)]^2} \tilde{Y}_\tau\|^2.$$

For term (c), by applying Corollary 1 together with the bound in Equation (99), we obtain:

$$L_{S_\tau}(f_\tau^*) - L_{D_\tau}(f_\tau^*)$$

$$\leq 2\rho \mathcal{R}_{S_\tau}(\mathcal{F}_\tau) + 3c\sqrt{\frac{log(2/\delta)}{2n_\tau}} \tag{107}$$

$$\leq 2\rho \sum_{k=1}^{\tau} \frac{[Tr(K_k(X_\tau, X_\tau)) \tilde{Y}_k^\top (E_{k,t_k^*}^{\text{PGN}})^2 K_k(X_k, X_k)^{-1} \tilde{Y}_k]^{1/2}}{n_\tau} + 3c\sqrt{\frac{log(2/\delta)}{2n_\tau}}.$$

Then, we have:

$$F_T \leq \frac{1}{T-1} \sum_{\tau=1}^{T-1} \left\{ 2\rho \sum_{k=1}^{T} \frac{[Tr(K_k(X_\tau, X_\tau)) \tilde{Y}_k^\top (E_{k,t_k^*}^{\text{PGN}})^2 K_k(X_k, X_k)^{-1} \tilde{Y}_k]^{1/2}}{n_\tau} \right.$$

$$+ 2\rho \sum_{k=1}^{\tau} \frac{[Tr(K_k(X_\tau, X_\tau)) \tilde{Y}_k^\top (E_{k,t_k^*}^{\text{PGN}})^2 K_k(X_k, X_k)^{-1} \tilde{Y}_k]^{1/2}}{n_\tau} + 6c\sqrt{\frac{log(2/\delta)}{2n_\tau}}$$

$$+ \frac{T-\tau}{n_\tau} \sum_{k=\tau+1}^{T} \|K_k(X_\tau, X_k) E_{k,t_k^*}^{\text{PGN}} K_k(X_k, X_k)^{-1} \tilde{Y}_k\|^2 \tag{108}$$

$$\left. + \frac{1}{2n_\tau} \|e^{-\frac{t_\tau^*}{n_\tau} K_\tau(X_\tau, X_\tau) - \frac{\Phi_\tau(t_\tau^*)}{n_\tau} [K_\tau(X_\tau, X_\tau)]^2} \tilde{Y}_\tau\|^2 \right\}.$$

# G ADDITIONAL PROOFS

## G.1 PROOF OF LEMMA 1

*Proof.* To better reflect practical training scenarios, we consider a finite number of training iterations, denoted by $t_T^{\max}$, and restrict $t_T$ to the interval $[1, t_T^{\max}]$. We exclude the trivial case $t_T = 0$, as it corresponds to the stopping point of task $T-1$. Accordingly, we analyze the evolution of the upper bounds $G_{t_T}^{\text{upper}}$ and $F_{t_T}^{\text{upper}}$ over $[1, t_T^{\max}]$.

(I) We first analyze $G_{t_T}^{\text{upper}}$. Its derivative with respect to $t_T$ can be written as

$$\frac{dG_{t_T}^{\text{upper}}}{dt_T} = \frac{1}{T}(g_1(t_T) + \rho g_2(t_T)), \tag{109}$$

where the functions $g_1$ and $g_2$ are given by

$$g_1(t_T) := \sum_{\tau=1}^{T} \frac{2(T-\tau)}{n_\tau n_T} \tilde{Y}_T^\top e^{-\frac{t_T}{n_T} K_T(X_T, X_T)} K_T(X_T, X_\tau) K_T(X_\tau, X_T) (I - e^{-\frac{t_T}{n_T} K_T(X_T, X_T)}).$$

$$K_T(X_T, X_T)^{-1} \tilde{Y}_T - \frac{2}{n_T^2} \tilde{Y}_T^\top e^{-\frac{2}{n_T} K_T(X_T, X_T) t_T} K_T(X_T, X_T) \tilde{Y}_T, \tag{110}$$

and

$$g_2(t_T) := \sum_{\tau=1}^{T} \frac{\left[\, \mathrm{Tr}(K_T(X_\tau, X_\tau))\right]^{1/2}}{n_\tau n_T} \frac{\tilde{Y}_T^\top e^{-\frac{t_T}{n_T} K_T(X_T, X_T)} \tilde{Y}_T}{[\tilde{Y}_T^\top (I - e^{-\frac{t_T}{n_T} K_T(X_T, X_T)}) K_T(X_T, X_T)^{-1} \tilde{Y}_T]^{1/2}}.$$
(111)

We first assume that $\left[\, \mathrm{Tr}(K_T(X_\tau, X_\tau))\right]^{1/2} \neq 0$ for any $\tau \in [T]$, and that $\tilde{Y}_T \neq 0$. This assumption is mild, since if either term equals zero, the corresponding component can simply be omitted. From the proof of Lemma 5, we know that $e^{-\frac{t_T}{n_T} K_T(X_T, X_T)}$ is positive definite. Consequently, $g_2(t_T) > 0$ holds for all $t_T \in [1, t_T^{\max}]$.

Moreover, both $g_1$ and $g_2$ are continuous with respect to $t_T$ on the interval $[1, t_T^{\max}]$. We therefore define

$$m_1 := \min_{x \in [1, t_T^{\max}]} g_2(x) > 0, \qquad M_1 := \max_{x \in [1, t_T^{\max}]} |g_1(x)|.$$

Let $\rho_g := \frac{M_1}{m_1}$. Then, for any $\rho > \rho_g$, we obtain

$$\begin{aligned}
\frac{dG_{t_T}^{\mathrm{upper}}}{dt_T} &= \frac{1}{T}(g_1(t_T) + \rho g_2(t_T)) \\
&> \frac{1}{T}\left(g_1(t_T) + \frac{M_1}{m_1} g_2(t_T)\right) \\
&\geq \frac{1}{T}(g_1(t_T) + M_1) \\
&\geq 0.
\end{aligned}$$
(112)

Therefore, for any Lipschitz constant $\rho > \rho_g$, $G_{t_T}^{\mathrm{upper}}$ is monotonically increasing with respect to $t_T$.

Since

$$\sum_{k=1}^{T} \frac{[\mathrm{Tr}(K_k(X_\tau, X_\tau)) \tilde{Y}_k^\top E_{k, t_k^*} K_k(X_k, X_k)^{-1} \tilde{Y}_k]^{1/2}}{n_\tau} \geq 0,$$
(113)

it follows that a smaller Lipschitz constant $\rho$ directly results in smaller values of $G_T^{\mathrm{upper}}$ for any fixed $t_T$.

(II) Then, we analyze $F_{t_T}^{\mathrm{upper}}$ similarly. Its derivative with respect to $t_T$ can be written as

$$\frac{dF_{t_T}^{\mathrm{upper}}}{dt_T} = \frac{1}{T-1}(f_1(t_T) + \rho f_2(t_T)),$$
(114)

where the functions $f_1$ and $f_2$ are given by

$$\begin{aligned}
f_1(t_T) := \sum_{\tau=1}^{T-1} \frac{2(T-\tau)}{n_\tau n_T} \tilde{Y}_T^\top e^{-\frac{t_T}{n_T} K_T(X_T, X_T)} K_T(X_T, X_\tau) K_T(X_\tau, X_T) \cdot \\
(I - e^{-\frac{t_T}{n_T} K_T(X_T, X_T)}) K_T(X_T, X_T)^{-1} \tilde{Y}_T,
\end{aligned}$$
(115)

and

$$f_2(t_T) := \sum_{\tau=1}^{T-1} \frac{\left[\, \mathrm{Tr}(K_T(X_\tau, X_\tau))\right]^{1/2}}{n_\tau n_T} \frac{\tilde{Y}_T^\top e^{-\frac{t_T}{n_T} K_T(X_T, X_T)} \tilde{Y}_T}{[\tilde{Y}_T^\top (I - e^{-\frac{t_T}{n_T} K_T(X_T, X_T)}) K_T(X_T, X_T)^{-1} \tilde{Y}_T]^{1/2}}.$$
(116)

We adopt the same assumption as in Part (I). Hence, $f_2(t_T) > 0$ for all $t_T \in [1, t_T^{\max}]$.

Moreover, both $f_1$ and $f_2$ are continuous with respect to $t_T$ on the interval $[1, t_T^{\max}]$. We therefore define

$$m_2 := \min_{x \in [1, t_T^{\max}]} f_2(x) > 0, \qquad M_2 := \max_{x \in [1, t_T^{\max}]} |f_1(x)|.$$

Let $\rho_f := \frac{M_2}{m_2}$. Then, for any $\rho > \rho_f$, we obtain

$$
\begin{aligned}
\frac{dF_{t_T}^{\text{upper}}}{dt_T} &= \frac{1}{T-1}(f_1(t_T) + \rho f_2(t_T)) \\
&> \frac{1}{T-1}\left(f_1(t_T) + \frac{M_2}{m_2}f_2(t_T)\right) \\
&\geq \frac{1}{T-1}(f_1(t_T) + M_2) \\
&\geq 0.
\end{aligned}
\tag{117}
$$

Therefore, for any Lipschitz constant $\rho > \rho_f$, the bound $F_{t_T}^{\text{upper}}$ is monotonically increasing with respect to $t_T$. Let $\rho^* := \max\{\rho_f, \rho_g\}$. It then follows that both $G_{t_T}^{\text{upper}}$ and $F_{t_T}^{\text{upper}}$ are monotonically increasing in $t_T$ whenever $\rho > \rho^*$.

In addition, since

$$
\sum_{k=1}^{T} \frac{\left[\text{Tr}(K_k(X_\tau, X_\tau))\tilde{Y}_k^\top E_{k,t_k^*} K_k(X_k, X_k)^{-1}\tilde{Y}_k\right]^{1/2}}{n_\tau} \geq 0,
\tag{118}
$$

and

$$
\sum_{k=1}^{\tau} \frac{\left[\text{Tr}(K_k(X_\tau, X_\tau))\tilde{Y}_k^\top E_{k,t_k^*} K_k(X_k, X_k)^{-1}\tilde{Y}_k\right]^{1/2}}{n_\tau} \geq 0,
\tag{119}
$$

it follows that a smaller Lipschitz constant $\rho$ directly leads to smaller values of $F_T^{\text{upper}}$ for any fixed $t_T$. □

## G.2 BRIDGING PREDICTION- AND PARAMETER-LIPSCHITZ CONSTANTS

We first introduce two types of Lipschitz constants associated with the loss function. For any fixed label $y$, the *prediction Lipschitz constant* $\rho_f$ measures the smoothness of the loss function $\ell(\cdot, y)$ with respect to the model prediction. Specially, for any $z_1, z_2 \in \mathcal{Z}$,

$$
|\ell(z_1, y) - \ell(z_2, y)| \leq \rho_f \|z_1 - z_2\|.
$$

Similarly, the *parameter Lipschitz constant* $\rho_\theta$ characterizes the variation of the loss with respect to the model parameters. Specifically, for any $\theta_1, \theta_2 \in \Omega$ and any sample $(x, y)$,

$$
|\ell(f_{\theta_1}(x), y) - \ell(f_{\theta_2}(x), y)| \leq \rho_\theta \|\theta_1 - \theta_2\|.
$$

By the mean value theorem and the Cauchy–Schwarz inequality, there exists a point $z'$ on the line segment between $z_1$ and $z_2$, and a point $\theta'$ between $\theta_1$ and $\theta_2$, such that

$$
|\ell(z_1, y) - \ell(z_2, y)| \leq \|\nabla_z \ell(z', y)\| \|z_1 - z_2\|,
$$

$$
|\ell(f_{\theta_1}(x), y) - \ell(f_{\theta_2}(x), y)| \leq \|\nabla_\theta \ell(f_{\theta'}(x), y)\| \|\theta_1 - \theta_2\|.
$$

Therefore, the magnitudes of $\rho_f$ and $\rho_\theta$ are upper bounded by the supremum norms of the prediction gradient $\|\nabla_z \ell\|$ and the parameter gradient $\|\nabla_\theta \ell\|$, respectively.

For any individual sample $x$, let its prediction be $z = f(x)$. By the chain rule,

$$
\nabla_\theta \ell(z, y) = J_f(x)^\top \nabla_z \ell(z, y),
$$

where $J_f(x) = \frac{\partial f(x)}{\partial \theta}$ denotes the Jacobian matrix of the model output with respect to parameters at input $x$. By properties of singular values, the sample-level gradient norm satisfies the following two-sided bound:

$$
\sigma_{\min}(J_f(x)) \|\nabla_z \ell(z, y)\| \leq \|\nabla_\theta \ell(z, y)\| \leq \|J_f(x)\| \|\nabla_z \ell(z, y)\|.
$$

Consequently, in regions where the Jacobian is well conditioned, i.e., there exists a constant $m > 0$ such that $\sigma_{\min}(J_f(x)) \geq m$ for all samples $x$, we obtain

$$\|\nabla_z \ell(z, y)\| \leq \frac{1}{m} \|\nabla_\theta \ell(z, y)\|.$$

This inequality indicates that controlling the parameter gradient norm $\|\nabla_\theta \ell\|$ effectively constrains the prediction gradient norm $\|\nabla_z \ell\|$, thereby locally regulating the prediction Lipschitz constant $\rho_f$.

For simplicity of presentation, we do not distinguish between the prediction and parameter Lipschitz constants in the remainder of this paper and refer to both collectively as the Lipschitz constant.

### G.3   PROOF OF LEMMA 2 AND DISCUSSION

We first present the proof of Lemma 2 as follows.

*Proof.* Consider any entry $\tilde{K}_k(x_{k-1}^i, x_k^j)$ with $i \in [n_{k-1}]$ and $j \in [n_k]$:

$$\tilde{K}_k(x_{k-1}^i, x_k^j) = \langle P_k \nabla_{\theta_{k-1}^*} f_{k-1}^*(x_{k-1}^i), \, P_k \nabla_{\theta_{k-1}^*} f_{k-1}^*(x_k^j) \rangle. \tag{120}$$

By definition, the subspace $\mathbb{E}_{k-1} = \text{span}\{\nabla_{\theta_{k-1}^*} f_{k-1}^*(x_{k-1}^i)\}_{i=1}^{n_{k-1}}$. Since $\nabla_{\theta_{k-1}^*} f_{k-1}^*(x_{k-1}^i) \in \mathbb{E}_{k-1}$, applying the projection operator $P_k$ yields $P_k \nabla_{\theta_{k-1}^*} f_{k-1}^*(x_{k-1}^i) = 0$. Therefore, it follows that

$$\tilde{K}_k(x_{k-1}^i, x_k^j) = \tilde{K}_k(x_{k-1}^i, x_{k-1}^l) = 0, \quad \forall i \in [n_{k-1}], \, j \in [n_k], \, l \in [n_{k-1}].$$

$\square$

**Further discussion of Lemma 2.** If we define the projector $P_k$ as $P_{\mathbb{E}_{k-1}^\perp}$, then Lemma 2 still holds, since $P_k \nabla_{\theta_{k-1}^*} f_{k-1}^*(x_{k-1}^i) = 0$ remains satisfied. This ensures that the gradients of the current task are orthogonal only to those of the immediately preceding task.

### G.4   PROOF OF LEMMA 3

We present the proof of Lemma 3 as follows.

*Proof.* Consider any entry $\hat{K}_k(x_\tau^i, x_k^j)$ with $i \in [n_\tau]$ and $j \in [n_k]$:

$$\hat{K}_k(x_\tau^i, x_k^j) = \langle P_k' \nabla_{\theta_{k-1}^*} f_{k-1}^*(x_\tau^i), \, P_k' \nabla_{\theta_{k-1}^*} f_{k-1}^*(x_k^i) \rangle. \tag{121}$$

By definition, the subspace $\mathbb{E}_k' = \text{span}\{\nabla_{\theta_k^*} f_k^*(x_l^m) \mid l \in [k], \, m \in [n_l]\}$. Since $\nabla_{\theta_{k-1}^*} f_{k-1}^*(x_\tau^i) \in \mathbb{E}_{k-1}'$, applying the projection operator $P_k'$ yields $P_k' \nabla_{\theta_{k-1}^*} f_{k-1}^*(x_\tau^i) = 0$. Therefore, it follows that

$$\hat{K}_k(x_\tau^i, x_k^j) = \hat{K}_k(x_\tau^i, x_\tau^l) = 0, \quad \forall i \in [n_\tau], \, j \in [n_k], \, l \in [n_\tau].$$

$\square$

### G.5   FORGETTING AND GENERALIZATION ERROR BOUNDS FOR OGD AND OGD+

In this section, we derive upper bounds on forgetting and generalization error for both OGD and OGD+. Define $\tilde{E}_{\tau,t} := I - \exp\left(-\frac{t}{n_\tau} \tilde{K}_\tau(X_\tau, X_\tau)\right)$ and $\hat{E}_{\tau,t} := I - \exp\left(-\frac{t}{n_\tau} \hat{K}_\tau(X_\tau, X_\tau)\right)$.

**Theorem 4** (OGD). *Consider a sequence of $T$ tasks. For each task $\tau \in [T]$, let $\mathcal{D}_\tau$ denote the data distribution, and let $S_\tau = \{X_\tau, Y_\tau\}$ be the corresponding training dataset drawn i.i.d. from $\mathcal{D}_\tau$. Suppose the loss function $\ell(\cdot, \cdot)$ takes values in the interval $[0, c]$ and is $\rho$-Lipschitz in the first argument. Then, with probability at least $1 - \delta$, the following bounds hold.*

$$F_{t_T} \leq \frac{1}{T-1} \sum_{\tau=1}^{T-1} \left\{ 2\rho \sum_{k=1}^{T} \frac{\left[ \text{Tr}(\tilde{K}_k(X_\tau, X_\tau)) \tilde{Y}_k^\top \tilde{E}_{k,t_k^*} \tilde{K}_k(X_k, X_k)^{-1} \tilde{Y}_k \right]^{1/2}}{n_\tau} \right.$$

$$+ 2\rho \sum_{k=1}^{\tau} \frac{\left[ \text{Tr}(\tilde{K}_k(X_\tau, X_\tau)) \tilde{Y}_k^\top \tilde{E}_{k,t_k^*} \tilde{K}_k(X_k, X_k)^{-1} \tilde{Y}_k \right]^{1/2}}{n_\tau} + 6c\sqrt{\frac{log(2/\delta)}{2n_\tau}}$$

$$\left. + \frac{T-\tau}{n_\tau} \sum_{k=\tau+2}^{T} \| \tilde{K}_k(X_\tau, X_k) \tilde{E}_{k,t_k^*} \tilde{K}_k(X_k, X_k)^{-1} \tilde{Y}_k \|^2 + \frac{1}{2n_\tau} \| e^{-\frac{t_\tau^*}{n_\tau} \tilde{K}_\tau(X_\tau, X_\tau)} \tilde{Y}_\tau \|^2 \right\}, \tag{122}$$

$$G_{t_T} \leq \frac{1}{T} \sum_{\tau=1}^{T} \left\{ \frac{T-\tau}{n_\tau} \sum_{k=\tau+2}^{T} \| \tilde{K}_k(X_\tau, X_k) \tilde{E}_{k,t_k^*} \tilde{K}_k(X_k, X_k)^{-1} \tilde{Y}_k \|^2 + \frac{1}{n_\tau} \| e^{-\frac{t_\tau^*}{n_\tau} \tilde{K}_\tau(X_\tau, X_\tau)} \tilde{Y}_\tau \|^2 \right.$$

$$\left. + 2\rho \sum_{k=1}^{T} \frac{\left[ \text{Tr}(\tilde{K}_k(X_\tau, X_\tau)) \tilde{Y}_k^\top \tilde{E}_{k,t_k^*} \tilde{K}_k(X_k, X_k)^{-1} \tilde{Y}_k \right]^{1/2}}{n_\tau} \right\} + 3c\sqrt{\frac{log(2/\delta)}{2n_\tau}}. \tag{123}$$

**Theorem 5** (OGD+). *Consider a sequence of $T$ tasks. For each task $\tau \in [T]$, let $\mathcal{D}_\tau$ denote the data distribution, and let $S_\tau = \{X_\tau, Y_\tau\}$ be the corresponding training dataset drawn i.i.d. from $\mathcal{D}_\tau$. Suppose the loss function $\ell(\cdot, \cdot)$ takes values in the interval $[0, c]$ and is $\rho$-Lipschitz in the first argument. Then, with probability at least $1 - \delta$, the following bounds hold.*

$$F_{t_T} \leq \frac{1}{T-1} \sum_{\tau=1}^{T-1} \left\{ 2\rho \sum_{k=1}^{T} \frac{\left[ \text{Tr}(\hat{K}_k(X_\tau, X_\tau)) \tilde{Y}_k^\top \hat{E}_{k,t_k^*} \hat{K}_k(X_k, X_k)^{-1} \tilde{Y}_k \right]^{1/2}}{n_\tau} + 6c\sqrt{\frac{log(2/\delta)}{2n_\tau}} \right.$$

$$\left. + 2\rho \sum_{k=1}^{\tau} \frac{\left[ \text{Tr}(\hat{K}_k(X_\tau, X_\tau)) \tilde{Y}_k^\top \hat{E}_{k,t_k^*} \hat{K}_k(X_k, X_k)^{-1} \tilde{Y}_k \right]^{1/2}}{n_\tau} + \frac{1}{2n_\tau} \| e^{-\frac{t_\tau^*}{n_\tau} \hat{K}_\tau(X_\tau, X_\tau)} \tilde{Y}_\tau \|^2 \right\}, \tag{124}$$

$$G_{t_T} \leq \frac{1}{T} \sum_{\tau=1}^{T} \left\{ 2\rho \sum_{k=1}^{T} \frac{\left[ \text{Tr}(\hat{K}_k(X_\tau, X_\tau)) \tilde{Y}_k^\top \hat{E}_{k,t_k^*} \hat{K}_k(X_k, X_k)^{-1} \tilde{Y}_k \right]^{1/2}}{n_\tau} \right.$$

$$\left. + \frac{1}{n_\tau} \| e^{-\frac{t_\tau^*}{n_\tau} \hat{K}_\tau(X_\tau, X_\tau)} \tilde{Y}_\tau \|^2 + 3c\sqrt{\frac{log(2/\delta)}{2n_\tau}} \right\}. \tag{125}$$

Therefore, the bounds on both forgetting and generalization error for OGD in CL are tighter than those for standard SGD. Furthermore, OGD+ achieves even tighter bounds than OGD, providing stronger theoretical guarantees.

### G.6 PROOF OF LEMMA 4

Based on the bounds established in Theorem 5, we characterize the evolution of $F_{t_T}^{\text{upper}+}$ and $G_{t_T}^{\text{upper}+}$.

*Proof.* Similar to the proof in Appendix G.1, we analyze the evolution of the upper bounds $G_{t_T}^{\text{upper}+}$ and $F_{t_T}^{\text{upper}+}$ over the interval $[1, t_T^{\max}]$.

(I) We first analyze $G_{t_T}^{\text{upper}+}$. Its derivative with respect to $t_T$ can be written as

$$\frac{dG_{t_T}^{\text{upper}+}}{dt_T} = \frac{1}{T} (g_1(t_T) + \rho g_2(t_T)), \tag{126}$$

where the functions $g_1$ and $g_2$ are given by

$$g_1(t_T) := -\frac{2}{n_T^2} \tilde{Y}_T^\top e^{-\frac{2}{n_T}\hat{K}_T(X_T,X_T)t_T} \hat{K}_T(X_T,X_T)\tilde{Y}_T, \tag{127}$$

and

$$g_2(t_T) := \sum_{\tau=1}^{T} \frac{\left[\operatorname{Tr}(\hat{K}_T(X_\tau,X_\tau))\right]^{1/2}}{n_\tau n_T} \frac{\tilde{Y}_T^\top e^{-\frac{t_T}{n_T}\hat{K}_T(X_T,X_T)}\tilde{Y}_T}{[\tilde{Y}_T^\top(I - e^{-\frac{t_T}{n_T}\hat{K}_T(X_T,X_T)})\hat{K}_T(X_T,X_T)^{-1}\tilde{Y}_T]^{1/2}}. \tag{128}$$

We adopt the same assumption as in the proof of Lemma 1. Since $e^{-\frac{t_T}{n_T}\hat{K}_T(X_T,X_T)}$ is positive definite, it follows that $g_2(t_T) > 0$ for all $t_T \in [1, t_T^{\max}]$. Moreover, since $e^{-\frac{2t_T}{n_T}\hat{K}_T(X_T,X_T)}\hat{K}_T(X_T,X_T)$ is positive definite, it follows that $g_1(t_T) < 0$ for all $t_T \in [1, t_T^{\max}]$.

Moreover, both $g_1$ and $g_2$ are continuous with respect to $t_T$ on the interval $[1, t_T^{\max}]$. We therefore define

$$M_1 := -\max_{x \in [1,t_T^{\max}]} g_1(x) > 0, \qquad M_2 := \max_{x \in [1,t_T^{\max}]} g_2(x) > 0.$$

Let $\rho_g' := \frac{M_1}{M_2}$. Then, for any $\rho < \rho_g'$, we obtain

$$\begin{aligned}
\frac{dG_{t_T}^{\text{upper}+}}{dt_T} &= \frac{1}{T}(g_1(t_T) + \rho g_2(t_T)) \\
&< \frac{1}{T}\left(g_1(t_T) + \frac{M_1}{M_2}g_2(t_T)\right) \\
&\leq \frac{1}{T}(g_1(t_T) + M_1) \\
&\leq 0.
\end{aligned} \tag{129}$$

Therefore, for any Lipschitz constant $\rho < \rho_g'$, $G_{t_T}^{\text{upper}+}$ is monotonically decreasing with respect to $t_T$. Since

$$\sum_{k=1}^{T} \frac{[\operatorname{Tr}(\hat{K}_k(X_\tau,X_\tau))\tilde{Y}_k^\top \hat{E}_{k,t_k^*}\hat{K}_k(X_k,X_k)^{-1}\tilde{Y}_k]^{1/2}}{n_\tau} \geq 0, \tag{130}$$

it follows that a smaller Lipschitz constant $\rho$ directly results in smaller values of $G_{t_T}^{\text{upper}+}$ for any fixed $t_T$.

(II) Then, we analyze $F_{t_T}^{\text{upper}+}$ similarly. Its derivative with respect to $t_T$ can be written as

$$\frac{dF_{t_T}^{\text{upper}+}}{dt_T} = \frac{\rho}{T-1} f_2(t_T), \tag{131}$$

where the function $f_2$ is given by

$$f_2(t_T) := \sum_{\tau=1}^{T-1} \frac{\left[\operatorname{Tr}(\hat{K}_T(X_\tau,X_\tau))\right]^{1/2}}{n_\tau n_T} \frac{\tilde{Y}_T^\top e^{-\frac{t_T}{n_T}\hat{K}_T(X_T,X_T)}\tilde{Y}_T}{[\tilde{Y}_T^\top(I - e^{-\frac{t_T}{n_T}\hat{K}_T(X_T,X_T)})\hat{K}_T(X_T,X_T)^{-1}\tilde{Y}_T]^{1/2}}. \tag{132}$$

We adopt the same assumption as in the proof of Lemma 1. Since $e^{-\frac{t_T}{n_T}\hat{K}_T(X_T,X_T)}$ is positive definite, it follows that $f_2(t_T) > 0$ for all $t_T \in [1, t_T^{\max}]$, which leads to $\frac{dF_{t_T}^{\text{upper}+}}{dt_T} > 0$. Therefore, $F_{t_T}^{\text{upper}+}$ is monotonically increasing with respect to $t_T$.

In addition, since

$$\sum_{k=1}^{T} \frac{\left[\operatorname{Tr}(\hat{K}_k(X_\tau,X_\tau))\tilde{Y}_k^\top \hat{E}_{k,t_k^*}\hat{K}_k(X_k,X_k)^{-1}\tilde{Y}_k\right]^{1/2}}{n_\tau} \geq 0, \tag{133}$$

and

$$\sum_{k=1}^{\tau} \frac{\left[\, \mathrm{Tr}(\hat{K}_k(X_\tau, X_\tau))\tilde{Y}_k^\top \hat{E}_{k,t_k^*} \hat{K}_k(X_k, X_k)^{-1}\tilde{Y}_k \right]^{1/2}}{n_\tau} \geq 0, \tag{134}$$

it follows that a smaller Lipschitz constant $\rho$ directly leads to smaller values of $F_{t_T}^{\mathrm{upper}+}$ for any fixed $t_T$. $\square$

