# OpenReview forum: "Understanding the Dynamics of Forgetting and Generalization in Continual Learning via the Neural Tangent Kernel"
_ICLR.cc/2026/Conference — ICLR 2026 Poster_

### Official Review · Reviewer_iWjj · 2025-10-30

**Soundness:** 3
**Presentation:** 3
**Contribution:** 3
**Rating:** 6
**Confidence:** 3

**Summary:**

This paper leverages the Neural Tangent Kernel (NTK) to derive upper bounds for both forgetting and generalization error at any step during training in vanilla CL tasks, while the bounds explicitly depending on the training steps. Two key conclusions are drawn: 1. Reducing the loss function’s Lipschitz constant (ρ) simultaneously decreases forgetting and generalization error. 2. Decreasing the magnitude of the cross-task kernel also mitigates forgetting and improves generalization. Based on these theoretical insights, the paper proposes two methods, OGD+ and OPGD, which project the current task’s gradient onto the orthogonal complement of the subspace spanned by
the gradients of recent tasks on all historical samples. Additionally, a gradient norm penalty (GAM) is introduced to jointly enforce “orthogonal constraints + Lipschitz reduction,” thereby suppressing forgetting while enhancing generalization. Experimental results demonstrate that the proposed methods significantly reduce errors. The writing is clear, the methodology sound, and the experimental outcomes show significant improvement

**Strengths:**

1. This paper models the error upper bound as an explicit function of the training step, allowing it to not only identify the main factors influencing the error but also discuss the effects of “training longer or shorter” on the final performance.

2. The main theorem clearly provides upper bounds for both forgetting and generalization errors, explicitly highlighting their dependence on training steps. By deriving a closed-form solution through kernel gradient flow and incorporating Rademacher complexity for two-sided control of the overall risk, the paper establishes a solid theoretical foundation. It further proposes a corresponding improvement method (OPGD) and proves its superiority, forming a coherent and logically complete framework.

3. The paper is organized around a clear framework: problem → challenges → theory → analysis → algorithm→ experiments → conclusion. The overall exposition is thorough, and the figures and tables effectively present the results in a clear and intuitive manner.

**Weaknesses:**

1. OPGD still requires storing a large amount of gradient data, leading to high computational and memory costs.  This may limit its applicability to large-scale models and tasks.

2. Beyond the current experimental datasets, it would be valuable to introduce more complex and realistic scenarios to further validate the robustness and generalization ability of the proposed algorithm.

3. Theoretical analysis occupies a large portion of the paper, while the main experimental results are relatively limited, making the demonstration of the method’s advantages less compelling.

**Questions:**

1. Is there a plan to evaluate them on more realistic tasks, such as online visual data streams or natural language sequences?

2. If the model deviates from the NTK assumption (e.g., CNNs), have different behaviors been observed in terms of forgetting and generalization mechanisms?

---

> ### Author Response · Authors · 2025-11-20
>
> Thank you for your thorough review and constructive comments. We provide detailed responses to each point below and have made major revisions to the manuscript. Revised text are highlighted in blue for ease of identification.
>
> **Q1.** OPGD still requires storing a large amount of gradient data, leading to high computational and memory costs. This may limit its applicability to large-scale models and tasks.
>
> **Response:** We appreciate the reviewer for raising this practical concern. The memory cost of standard OGD mainly comes from storing the gradients of all previous tasks. In contrast, OPGD stores only a small buffer of training samples for each past task rather than historical gradients. For large scale models, the memory required for storing samples is significantly smaller than that required for storing full gradient vectors. This design substantially reduces memory usage while still preserving the beneficial effect of gradient orthogonality. We further include an ablation study on the number of stored examples per task in Table 9 of Appendix C.4, which shows that a moderate buffer size already yields strong performance.
>
> The dominant computational cost comes from the number of projection steps and the dimensionality of the parameter space. The number of projection steps depends on the number of stored gradients, which in OPGD corresponds to the number of stored samples. Therefore, using a small buffer also reduces computational cost. In addition, recent work by [1] proposes O-LoRA, which learns task specific low rank subspaces that remain approximately orthogonal across tasks; this approach further reduces memory and computational overhead for transformer architectures. Reducing the cost of CL for very large models remains an important direction for  future work.
>
> [1] Wang, Xiao, et al. Orthogonal subspace learning for language model continual learning. EMNLP, 2023.
>
> **Q2.** Beyond the current experimental datasets, it would be valuable to introduce more complex and realistic scenarios to further validate the robustness and generalization ability of the proposed algorithm.
>
> **Response:** Thank you for this constructive suggestion. Following your advice, we extended our empirical evaluation to include online visual data streams; these experiments are described in our response to Q4. The online CL results demonstrate that our algorithm exhibits reduced forgetting and improved generalization in the online setting.
>
> **Q3.** Theoretical analysis occupies a large portion of the paper, while the main experimental results are relatively limited, making the demonstration of the method’s advantages less compelling.
>
> **Response:** We appreciate the reviewer for raising this practical concern. Our main contribution is a theoretical breakthrough that enables analysis of forgetting and generalization dynamics throughout training. Because our algorithms are built upon this theoretical framework, a substantial portion of the paper is devoted to theoretical analysis. Following your advice, we extended our empirical evaluation to include online visual data streams; these experiments are described in our response to Q4.
>
> **Q4.** Is there a plan to evaluate them on more realistic tasks, such as online visual data streams or natural language sequences?
>
> **Response:** Thank you for this valuable suggestion. Training natural language sequence tasks with transformer architectures typically requires substantial GPU resources. These resource requirements exceeded our available compute budget, and therefore we did not conduct full transformer scale experiments. To address the concern about realism we extend our empirical evaluation to online visual data streams.
>
> In the online CL protocol data arrive as a stream and each small batch is used for a single training iteration. Following standard practice in the literature, we set the batch size to 10 and train for one epoch per arrival. We evaluate on Permuted MNIST, Rotated MNIST and Split CIFAR-100. These online experiments show that our algorithms demonstrate strong performance in the online setting.  The result can be reported as follows:
>
> |Method|P-MNIST ACC|P-MNIST BWT|R-MNIST ACC|R-MNIST BWT|CIFAR-100 ACC|CIFAR-100 BWT|
> |-|-:|-:|-:|-:|-:|-:|
> |SGD|75.85±0.78|-12.24±0.81|67.23±0.41|-23.62±0.40|31.90±1.57|-16.21±1.42|
> |GAM|76.67±1.02|-10.95±1.10|70.63±0.59|-14.86±0.67|33.56±1.67|-14.29±1.65|
> |OGD|78.76±0.50|-8.65±0.49|79.17±0.41|-9.53±0.48|37.66±1.78|-7.92±1.66|
> |**OGD+**|81.88±0.33|-5.23±0.38|86.35±0.17|-1.02±0.28|37.68±1.77|-7.88±1.68|
> |**OPGD**|**82.62±0.51**|**-4.93±0.49**|**87.14±0.22**|**2.75±0.25**|**39.88±1.49**|**-6.12±1.45**|

---

> > ### Author Response · Authors · 2025-11-20
> >
> > **Q5.** If the model deviates from the NTK assumption (e.g., CNNs), have different behaviors been observed in terms of forgetting and generalization mechanisms?
> >
> > **Response:** Although our theoretical analysis is developed under the NTK assumption, our empirical results support the relevance of the derived insights beyond this regime. In particular, we empirically demonstrate two points.
> >
> > (1) Incorporating GAM into CL to reduce the Lipschitz constant helps reduce forgetting and improves generalization. A smaller Lipschitz constant indicates a flatter loss landscape, a connection discussed by [1]. This observation is consistent with recent work showing that a flatter loss landscape enhances CL performance, for example [2]  propose c-flat to strengthen CL.
> >
> > (2) We propose OPGD, which augments orthogonal gradient methods with GAM, and show that this combination further improves CL performance. Our experiments confirm that OPGD attains competitive results in both offline and online CL settings.
> >
> > Finally, the only assumption that the NTK regime imposes on the model is that the network is sufficiently wide. Hence the theoretical conclusions derived under this framework remain broadly applicable. Relaxing the NTK assumption to encompass more general model classes is an important and meaningful direction for future work.
> >
> > [1] Zhao, Yang, et al. Penalizing gradient norm for efficiently improving generalization in deep learning. ICML, 2022.
> >
> > [2] Bian, Ang, et al. Make continual learning stronger via c-flat. NeurIPS, 2024.
> >
> > **Finally, if our responses have addressed your concerns to a satisfactory extent, we would be very grateful if you could consider updating your evaluation accordingly. We are also more than happy to clarify any remaining questions during the discussion period. We once again thank the reviewer for the constructive feedback and valuable suggestions on our work.**

---

### Official Review · Reviewer_dpae · 2025-10-30

**Soundness:** 2
**Presentation:** 3
**Contribution:** 2
**Rating:** 2
**Confidence:** 4

**Summary:**

The paper analyzed continual learning in the Neural Tangent Kernel regime and derive intermediate-training bounds for both forgetting and generalization using kernel gradient flow and Rademacher complexity. They show that two quantities govern CL behavior during training: (i) the Lipschitz constant of the loss w.r.t. predictions—smaller is better for both forgetting and generalization, (ii) the cross-task kernel smaller norms reduce interference and forgetting. Building on these findings, they propose OGD+, which projects the current task’s gradient onto the orthogonal complement of a subspace formed by the previous task’s gradients evaluated on all earlier samples (eliminating cross-task kernels between any task pair), and OPGD, which augments OGD+ with gradient-norm penalization to lower the Lipschitz constant.

**Strengths:**

1. The paper is comprehensive, well written and well structured.

**Weaknesses:**

1. What means intermediate training?

2. The continual-training protocol is unclear-for example, how does the transition from task $k$ to task $k+1$ work? After training $t$ steps on task $k$, how is gradient flow for task $k+1$ initialized and defined relative to the state from task $k ?$

3. The assumption also requires a bounded loss function.

4. The paper did not specifically provide the forgetting and generalization error of PGN and PGN, only lemma 2 and lemma 3 provided, which is not enough; it should give the formal results like Theorem 1.

5. The comparison among related work is not sufficient. There are lots of theoretical CL works that consider the regularized setting, but the paper did not discuss.

6. The proposed methods are common in empirical CL studies, while the theoretical CL findings are not enough in my view.

**Questions:**

Above

---

> ### Author Response · Authors · 2025-11-20
>
> Thank you for your thorough review and constructive comments. We provide detailed responses to each point below and have made major revisions to the manuscript. Revised text are highlighted in blue for ease of identification.
>
> **Q1.** What means intermediate training?
>
> **Response：** Intermediate training refers to the evolution of the model during the training process before convergence. Previous works mainly focus on the performance of the final converged model. In contrast, our theoretical analysis characterizes forgetting and generalization at arbitrary training steps along the training trajectory, rather than only at the end of training.
>
> **Q2.** The continual-training protocol is unclear-for example, how does the transition from task $k$  to task $k+1$ work? After training $t$ steps on task $k$, how is gradient flow for task $k+1$ initialized and defined relative to the state from task $k$.
>
> **Response：** We appreciate the reviewer for raising this concern. The CL protocol is already specified in our original submission. After completing training on task $k$, we obtain the final parameter $\theta\_k^\*$. The next task $k+1$ is then initialized from this parameter, that is $\theta\_{k+1}^0 = \theta\_k^\*.$ This initialization rule is stated on line 168 of the manuscript.
>
> Given this initialization, the gradient flow on task $k+1$ is defined as
> \begin{equation}\frac{d}{dt}f_{k+1}^t(x) = -\frac{1}{n_{k+1}}K_{k+1}(x,X_{k+1})\big(f_{k+1}^t(X_{k+1})-Y_{k+1}\big).\end{equation}
> which is an ordinary differential equation in the time variable $t$ with initial condition induced by $\theta\_{k+1}^0 = \theta\_k^*$. Hence the trajectory for task $k+1$ starts from the final state reached after training task $k$.
>
> To make our setting clearer, we have added this clarification to Sections 2.1 and 2.2.
>
> **Q3.** The assumption also requires a bounded loss function.
>
> **Response：** Thank you for this helpful remark. We would like to clarify that this assumption was already included in our original submission. In Theorem 1 we assume that the loss function $\ell(\cdot,\cdot)$ takes values in the interval $[0,c]$, which ensures that the loss is  bounded. This condition is explicitly stated on line 197 of the manuscript.
>
> **Q4.** The comparison among related work is not sufficient. There are lots of theoretical CL works that consider the regularized setting, but the paper did not discuss.
>
> **Response：** Thank you for this thoughtful suggestion. We have added a more comprehensive discussion of theoretical works on regularization based CL in the Related Work section in Appendix B.
>
> Recently, there has been growing interest in the theoretical analysis of regularization based methods for CL. [1] theoretically characterize how the performance of a model in a contrastive CL framework is controlled by the training losses on previous tasks. [2] derive bounds on the average risk over two tasks for an $\ell_2$-regularized CL algorithm. [3] provide a statistical analysis of regularization based CL on a sequence of linear regression tasks and highlight how different regularization terms affect model performance. [4] establish upper and lower bounds on the joint excess risk for a generalized $\ell_2$-regularized CL algorithm.
>
> [1] Reinhard Heckel. Provable continual learning via sketched jacobian approximations. In AISTATS, 2022.
>
> [2] Haoran Li, et al. Fixed design analysis of regularization-based continual learning. In CoLLAs, 2023.
>
> [3] Xuyang Zhao, et al. A statistical theory of regularization-based continual learning. In ICML, 2024.
>
> [4] Haoran Li, et al. Memory-statistics tradeoff in continual learning with structural regularization. In arXiv, 2025.
>
> **Q5.** The proposed methods are common in empirical CL studies, while the theoretical CL findings are not enough in my view.
>
> **Response：** We thank the reviewer for this important concern. Our work advances both algorithmic design and theory in CL. We demonstrate, both theoretically and empirically, that (1) OGD+ reduces forgetting and improves generalization relative to standard OGD, and (2) reducing the Lipschitz constant mitigates forgetting and enhances generalization. While many prior methods were motivated primarily by empirical observations or heuristics, **our algorithms are supported by explicit theoretical guarantees.**
>
> From a theoretical perspective, we provide the first upper bounds on both forgetting and generalization error at intermediate training stages. Previous analyses mostly focus on the performance of the converged model, but CL exhibits significant performance changes throughout training. **This makes our theoretical contribution significant.** Importantly, **our theoretical contribution has also been positively recognized by other reviewers**: 5JLu described the result as novel, Q8vv regarded it as crucial for understanding real-world CL scenarios, and iWjj considered it a solid theoretical foundation.

---

> ### Author Response · Authors · 2025-11-20
>
> **Q6.** The paper did not specifically provide the forgetting and generalization error of PGN and PGN, only lemma 2 and lemma 3 provided, which is not enough; it should give the formal results like Theorem 1.
>
> **Response:** Thank you for your thorough review and constructive comments. In fact, Lemmas 2 and 3 show that the cross-kernel terms for OGD and OGD+ vanish; these statements are not relevant to PGN. Moreover, the formal theorems for OGD and OGD+ appear in Appendix H.5, as stated in Remark 4. Our use of PGN is motivated by Lemma 1, which demonstrates that reducing the Lipschitz constant consistently mitigates both forgetting and generalization error in CL. We have newly added Appendix F to provide the formal results and detailed proofs for the forgetting and generalization errors of PGN.
>
> **Theorem [PGN]** Consider a sequence of $T$ tasks. For each task $\tau\in[T]$, let $\mathcal{D}\_\tau$ denote the data distribution, and let $S\_\tau =\lbrace X\_\tau, Y\_\tau\rbrace$ be the corresponding training dataset drawn i.i.d. from $\mathcal{D}\_\tau$. Suppose the loss function $\ell(\cdot,\cdot)$ takes values in the interval $[0,c]$ and is $\rho$-Lipschitz in the first argument. Then, with probability at least $1-\delta$, the following bounds hold:
> \begin{aligned}F\_{t\_T}\leq&\frac{1}{T-1}\sum\_{\tau=1}^{T-1}\Bigg\lbrace2\rho\sum\_{k=\tau+1}^T \frac{\big[\operatorname{Tr}(K\_k(X\_\tau,X\_\tau))\tilde{Y}\_k^\top(E\_{k,t\_k^\*}^{\mathrm{PGN}})^2K\_k(X\_k,X\_k)^{-1} \tilde{Y}\_k \big]^{1/2} }{n\_\tau} \\\\&+4\rho\sum\_{k=1}^\tau\frac{ \big[ \operatorname{Tr}(K\_k(X\_\tau,X\_\tau))\tilde{Y}\_k^\top(E\_{k,t\_k^\*}^{\mathrm{PGN}})^2 K\_k(X\_k,X\_k)^{-1}\tilde{Y}\_k \big]^{1/2} }{n\_\tau}+6c\sqrt{\frac{\log(2/\delta)}{2n\_\tau}}\\\\&+\frac{1}{n\_\tau}\sum\_{k=\tau+1}^T||K\_k(X\_\tau,X\_k)E\_{k,t\_k^\*}^{\mathrm{PGN}} K\_k(X\_k,X\_k)^{-1} \tilde{Y}\_k ||^2\Bigg\rbrace.\end{aligned}
> \begin{aligned}
> G\_{t\_T} \leq & \frac{1}{T} \sum\_{\tau=1}^T\Bigg\lbrace
> \frac{1}{n\_\tau}\sum\_{k=\tau+1}^T||K\_k(X\_\tau,X\_k)E\_{k,t\_k^\*}^{\mathrm{PGN}}K\_k(X\_k,X\_k)^{-1} \tilde{Y}\_k||^2 +\frac{1}{n\_\tau}||e^{ -\frac{t\_\tau^\*}{n\_\tau} K\_\tau(X\_\tau,X\_\tau)-\frac{\Phi\_\tau(t\_\tau^\*)}{n\_\tau} [K\_\tau(X\_\tau,X\_\tau)]^2 } \tilde{Y}\_\tau||^2\\\\
> &+2\rho\sum\_{k=1}^T\frac{ [ \operatorname{Tr}(K\_k(X\_\tau,X\_\tau))\tilde{Y}\_k^\top (E\_{k,t\_k^\*}^{\mathrm{PGN}})^2 K\_k(X\_k,X\_k)^{-1} \tilde{Y}\_k ]^{1/2} }{n\_\tau}+3c\sqrt{\frac{\log(2/\delta)}{2 n\_\tau}}\Bigg\rbrace.
> \end{aligned}
> where $E_{\tau,t}^{\mathrm{PGN}}=I-\exp\Big(-\tfrac{t}{n_\tau}K_\tau(X_\tau,X_\tau)-\tfrac{\Phi_\tau(t)}{n_\tau}K_\tau^2(X_\tau,X_\tau)\Big)$ and $\Phi_\tau(t)$ satisfy $\Phi_\tau(t)=\int_0^t\frac{\alpha_\tau}{\sqrt{[f_\tau^st(X_\tau)-Y_\tau]^\top K_\tau [f_\tau^s(X_\tau)-Y_\tau}]}ds$.
>
> **Proof Sketch:**
>
> (1) **Gradient flow of PGN.** We first compute the gradient of the PGN loss and apply the chain rule $\frac{d}{dt} f_\tau^t(x)=\nabla_{\theta_\tau^t}f_\tau^t(x)\frac{d\theta_\tau^t}{dt}$ to derive the kernel gradient flow of PGN:
> \begin{equation}\frac{d}{dt}f_\tau^t(x)=-\frac{1}{n_\tau}K_\tau(x, X_\tau)\big(f_\tau^t(X_\tau)-Y_\tau\big)-\frac{\alpha_\tau}{n_\tau}\frac{K_\tau(x, X_\tau)K_\tau(X_\tau,X_\tau)\big(f_\tau^t(X_\tau)-Y_\tau\big)}{\sqrt{\big(f_\tau^t(X_\tau)-Y_\tau\big)^\top K_\tau(X_\tau, X_\tau)\big(f_\tau^t(X_\tau)-Y_\tau\big)}}.\end{equation}
>
> (2) **Solution of the kernel gradient flow for PGN.**  We first derive the  solution on the training set $X_\tau$, and then obtain the solution at an arbitrary point $x$:
>
> \begin{equation}f_\tau^t(x)=f_{\tau-1}^\*(x)-K_\tau(x,X_\tau)E_{\tau,t}^{\mathrm{PGN}}K_\tau^{-1}(X_\tau, X_\tau)\big(f_{\tau-1}^*(X_\tau)-Y_\tau\big).\end{equation}
>
> (3) **Bound on the generalization error.** We continue to use the Rademacher complexity together with the empirical loss to obtain an upper bound on the population loss:
> \begin{equation}L\_{D\_\tau}(f\_{T}^\*)\le L\_{S\_\tau}(f\_{T}^\*)+2\rho\mathcal{R}\_{S\_\tau}(\mathcal{F}\_T)+3c\sqrt{\frac{log(2/\delta)}{2n\_\tau}}.\end{equation}
>
> (4) **Bound on forgetting.** We decompose each term in the forgetting metric as
> \begin{aligned}L\_{D\_\tau}(f\_{T}^\*)-L\_{D\_\tau}(f\_{\tau}^\*)=\underbrace{L\_{D\_\tau}(f\_{T}^\*)-L\_{S\_\tau}(f\_{T}^\*)}\_{(a)} +\underbrace{L\_{S\_\tau}(f\_{T}^\*)-L\_{S\_\tau}(f\_{\tau}^\*)}\_{(b)}+\underbrace{L\_{S\_\tau}(f\_{\tau}^\*)-L\_{D\_\tau}(f\_{\tau}^\*)}\_{(c)}.
> \end{aligned}
>
> Terms $(a)$ and $(c)$ are controlled by the Rademacher complexities $\mathcal{R}\_{S_\tau}(\mathcal{F}\_T)$ and $\mathcal{R}\_{S_\tau}(\mathcal{F}\_\tau)$ respectively. The bounds on $\mathcal{R}\_{S_\tau}(\mathcal{F}\_T)$ and $\mathcal{R}\_{S_\tau}(\mathcal{F}\_\tau)$ have already been obtained in step (3). Term $(b)$ can be bounded as
>
> \begin{aligned}L\_{S_\tau}(f\_{T}^\*)-L\_{S_\tau}(f\_{\tau}^\*)\le \frac{1}{n\_\tau}\sum\_{k=\tau+1}^T|| K\_k(X\_\tau,X\_k)E_{k,t\_k^\*}^{\mathrm{PGN}} K\_k(X\_k,X\_k)^{-1}\tilde{Y}\_{k}||^2.\end{aligned}

---

> > ### Author Response · Authors · 2025-11-27
> >
> > We hope that our responses have addressed your questions and clarified the contributions of our work. As the discussion period is approaching its end, we would greatly appreciate any additional comments or guidance on any remaining concerns. Thank you again for your time and constructive feedback.

---

### Official Review · Reviewer_Q8vv · 2025-10-30

**Soundness:** 3
**Presentation:** 3
**Contribution:** 2
**Rating:** 6
**Confidence:** 4

**Summary:**

The paper addresses the challenges of continual learning (CL), specifically focusing on the dynamics of forgetting and generalization. Continual learning aims to enable models to learn tasks sequentially without forgetting previously acquired knowledge. The authors analyze these dynamics within the Neural Tangent Kernel (NTK) regime, addressing two main challenges: characterizing intermediate training stages and establishing forgetting bounds using population risk bounds. They introduce two novel algorithms, OGD+ and Orthogonal Penalized Gradient Descent (OPGD), both of which project gradients orthogonally to reduce forgetting and enhance generalization. Their empirical studies on benchmarks like Permuted MNIST, Rotated MNIST, and Split CIFAR-100 validate the theoretical predictions and demonstrate the effectiveness of their approaches.

**Strengths:**

1. **Theoretical Contribution**: The paper extends the NTK-based analysis to intermediate training dynamics, which is crucial for understanding real-world continual learning scenarios. By providing both upper and lower bounds on population risk using Rademacher complexity, the authors offer rigorous theoretical insights.

2. **Algorithm Design**: OGD+ and OPGD are innovative in their approach to mitigate forgetting and improve generalization. The use of gradient orthogonality and gradient-norm penalization represents a well-founded strategy for continual learning models.

3. **Empirical Validation**: The experiments conducted on various benchmarks, such as Permuted MNIST and Split CIFAR-100, confirm the theoretical discussions, demonstrating the practical efficacy of OPGD and OGD+ compared to standard methods.

4. **Clear Presentation**: The paper clearly outlines its contributions, setting a solid baseline through comparative analysis with other methods in Table 1. This provides context for the improvements made by the proposed methods.

**Weaknesses:**

1. **Incremental Algorithmic Novelty**:  The algorithms are mainly based on OGD and gradient norm penalty, which seems to be a little bit incremental.

2. **Limited Applicability**: The theoretical framework is developed under the NTK regime, which might not encapsulate the full behavior of practical deep networks with finite width and more complex architectures. This limitation is acknowledged in the paper, suggesting a need for exploring applicability to diversified settings.

3. **Specific Task Benchmarks**: The empirical tests are confined to a specific set of benchmarks, potentially overlooking performance variations across other task types. While benchmarks like Permuted MNIST and Split CIFAR-100 are standard, testing across a broader spectrum could provide more insights.

**Questions:**

1. **Computation Complexity**: Is the projection step computationally heavy for complex NN? How to address this problem?

2. **Expansion Beyond NTK**: How would the authors suggest overcoming the limitations of the NTK-based analysis when dealing with finite-width networks or other types of architecture beyond classification tasks?

---

> ### Author Response · Authors · 2025-11-20
>
> Thank you for your thorough review and constructive comments. We provide detailed responses to each point below and have made major revisions to the manuscript. Revised text  are highlighted in blue for ease of identification.
>
> **Q1.** Incremental Algorithmic Novelty: The algorithms are mainly based on OGD and gradient norm penalty, which seems to be a little bit incremental.
>
> **Response：** Thank you for this insightful comment. Our algorithms indeed build on OGD and gradient norm regularization, yet we believe the contributions go beyond a purely incremental extension in the following aspects:
> (1) We provide a rigorous theoretical analysis of both OGD and gradient norm penalization that explains how these mechanisms mitigate forgetting and improve generalization in CL.
> (2) we introduce OGD+, which provably forgets less and generalizes better than standard OGD. This improvement is supported by both our theoretical guarantees and extensive empirical results.
> (3) We propose OPGD, which further strengthens OGD+ by penalizing the gradient norm in a principled way. We show that this additional penalty leads to tighter bounds on forgetting and generalization, and the experiments confirm the predicted gains.
>
> Overall, our proposed algorithms are supported by strong theoretical guarantees.
>
> **Q2.** Limited Applicability: The theoretical framework is developed under the NTK regime, which might not encapsulate the full behavior of practical deep networks with finite width and more complex architectures. This limitation is acknowledged in the paper, suggesting a need for exploring applicability to diversified settings.
>
> **Response：** Thank you for the careful review and constructive comments. Recent theoretical work in CL has largely focused on linear models, while our analysis in the NTK regime addresses a substantially broader class of architectures by requiring only that the network be sufficiently wide. Extending the current framework beyond the NTK regime to encompass finite-width networks and more complex architectures is both important and challenging. We regard this extension as a key direction for future work.
>
> **Q3.** Specific Task Benchmarks: The empirical tests are confined to a specific set of benchmarks, potentially overlooking performance variations across other task types. While benchmarks like Permuted MNIST and Split CIFAR-100 are standard, testing across a broader spectrum could provide more insights.
>
> **Response：** Thank you for your thorough review and constructive comments. In our original submission, we conducted experiments on offline CL. To extend the evaluation to additional task types, **we additionally perform experiments on online CL**. In this setting, data arrive as a stream, and each small batch is used for a single training iteration. Following standard practice, we set the batch size to 10 and train for one epoch per arrival. We evaluate on Permuted MNIST, Rotated MNIST, and Split CIFAR-100. These online experiments demonstrate that our algorithms maintain strong performance in the online setting. The results are reported as follows:
>
> |Method|P-MNIST ACC|P-MNIST BWT|R-MNIST ACC|R-MNIST BWT|CIFAR-100 ACC|CIFAR-100 BWT|
> |-|-:|-:|-:|-:|-:|-:|
> |SGD|75.85±0.78|-12.24±0.81|67.23±0.41|-23.62±0.40|31.90±1.57|-16.21±1.42|
> |GAM|76.67±1.02|-10.95±1.10|70.63±0.59|-14.86±0.67|33.56±1.67|-14.29±1.65|
> |OGD|78.76±0.50|-8.65±0.49|79.17±0.41|-9.53±0.48|37.66±1.78|-7.92±1.66|
> |**OGD+**|81.88±0.33|-5.23±0.38|86.35±0.17|-1.02±0.28|37.68±1.77|-7.88±1.68|
> |**OPGD**|**82.62±0.51**|**-4.93±0.49**|**87.14±0.22**|**2.75±0.25**|**39.88±1.49**|**-6.12±1.45**|

---

> ### Author Response · Authors · 2025-11-20
>
> **Q4.** Computation Complexity: Is the projection step computationally heavy for complex NN? How to address this problem?
>
> **Response：** We appreciate the reviewer for raising this practical concern. Our method adopts the standard projection used in OGD. The projection of a vector $u$ onto the direction of a vector $v$ is $proj\_v(u)=\frac{\langle u, v\rangle}{\langle v, v\rangle}v$, which only requires a small number of inner products and scalar multiplications. Consequently, a single projection operation does not incur a significant computational cost. The principal source of overhead in standard OGD arises from the number of stored gradients and the dimensionality of the parameter space.
>
> Our method OGD+ stores a small buffer of examples for each past task rather than retaining all historical gradients. Gradients are computed on this reduced buffer using the last converged model, and projections are applied with respect to these computed gradients. This design substantially reduces the number of projection operations. We report an ablation study on the number of stored examples per task in Table 9 of Appendix C.4, which shows that a moderate buffer size per task yields strong empirical performance.
>
> In addition, recent work explores to limit computation by restricting updates to low dimensional subspaces. For example, [1] propose O-LoRA, which learns task-specific low-rank subspaces that are kept approximately orthogonal across tasks; this approach reduces both memory and compute overhead for transformer architectures. Further reductions in computation and memory for very large models and datasets remain an important direction for future work.
>
> [1]Wang, Xiao, et al. "Orthogonal subspace learning for language model continual learning." Findings of the Association for Computational Linguistics: EMNLP 2023.
>
> **Q5.** Expansion Beyond NTK: How would the authors suggest overcoming the limitations of the NTK-based analysis when dealing with finite-width networks or other types of architecture beyond classification tasks?
>
> **Response：** This is an interesting and challenging question. Extending theoretical analysis of CL beyond the NTK regime is an active direction of our current research. Below we outline a decomposition that isolates the principal sources of forgetting and then sketch ideas for how each term might be studied beyond the NTK limit.
>
> We decompose each term in the forgetting metric as:
>
> \begin{aligned}L\_{D\_\tau}(f\_{T}^\*)-L\_{D\_\tau}(f\_{\tau}^\*)=\underbrace{L\_{D\_\tau}(f\_{T}^\*)-L\_{S\_\tau}(f\_{T}^\*)}\_{(a)} +\underbrace{L\_{S\_\tau}(f\_{T}^\*)-L\_{S\_\tau}(f\_{\tau}^\*)}\_{(b)}+\underbrace{L\_{S\_\tau}(f\_{\tau}^\*)-L\_{D\_\tau}(f\_{\tau}^\*)}\_{(c)}.
> \end{aligned}
>
> Term (b) depends only on the empirical loss and can be directly computed from the iteration dynamics of the training algorithm. In contrast, terms (a) and (c) correspond to the generalization error. Stability-based techniques [1] rely solely on the update rules and do not require assumptions on the network architecture, making them suitable for analyzing generalization in non-NTK models. Addressing this question is indeed challenging and may require the development of novel theoretical tools. Nevertheless, advancing this line of research is essential for the theoretical understanding of continual learning. We are actively pursuing this program in future work.
>
> [1]Hardt, Moritz, Ben Recht, and Yoram Singer. "Train faster, generalize better: Stability of stochastic gradient descent." International conference on machine learning. PMLR, 2016.
>
> **Finally, if our responses have addressed your concerns to a satisfactory extent, we would be very grateful if you could consider updating your evaluation accordingly. We are also more than happy to clarify any remaining questions during the discussion period. We once again thank the reviewer for the constructive feedback and valuable suggestions on our work.**

---

> > ### Comment · Reviewer_Q8vv · 2025-11-26
> > **Follow-up question**
> >
> > Thank you to the authors for the detailed response.
> >
> > Per your response to Q5, could you elaborate further on the generalization aspect of the theory?
> >
> > As presented in the submission, the generalization term is currently analyzed through Rademacher complexity, which is known to be somewhat loose. From my perspective, introducing a more refined generalization analysis could strengthen and tighten the theoretical results.

---

> > > ### Author Response · Authors · 2025-11-27
> > >
> > > We appreciate the reviewer’s comment. We adopt stability-based techniques to analyze generalization. Our **motivation** rests on two points:
> > > - Stability-based methods depend on the update rules and are therefore well suited to streaming scenarios. In particular, they naturally capture the interaction between tasks through the initialization rule in CL, namely that the converged parameters from the previous task serve as the initial parameters for the current task.
> > > - Stability-based generalization bounds can be tighter than bounds derived from Rademacher complexity in typical optimization regimes.
> > >
> > > In CL, we define the generalization error of the final model $w\_T$ as
> > > $G\_{T}=\frac{1}{T}\sum\_{\tau=1}^T L\_{\mathcal{D}\_\tau}(w\_T)$. We bound this quantity by controlling each term $L\_{\mathcal{D}\_\tau}(w\_T)$ separately. In particular, we aim to relate the population loss to the empirical loss by establishing
> > > \begin{equation}\mathbb{E}\_{S\_\tau,A^T}\big[L\_{\mathcal{D}\_\tau}(A\_{S\_\tau}^T)-L\_{S\_\tau}(A\_{S\_\tau}^T)\big]<\epsilon,\end{equation}
> > > where $A^T$ denotes a randomized algorithm for the sequence of $T$ tasks and $A\_{S_\tau}^T$ is the model returned when the dataset for task $\tau$ is $S\_\tau$. The expectation is taken over the randomness of both the algorithm and the sample $S\_\tau$, which yields a comprehensive, expectation-based characterization of generalization in the CL setting. Next, we provide a proof sketch of how we employ stability-based techniques to bound generalization. Throughout, we adopt notation and definitions similar to those in [1].
> > >
> > >
> > > **Proof sketch:** (1) Link the **uniform stability** of the learning algorithm to the expected generalization error bound in the CL setting.
> > >
> > > **Definition**[$\epsilon(A^T,\tau)$-Uniformly Stable] A randomized algorithm $A^T$ for a sequence of $T$ tasks is $\epsilon(A^T,\tau)$-uniformly stable if for all pairs of datasets for task $\tau$, $S\_\tau,S\_\tau'\in D\_\tau$ that differ in at most one example, it holds that
> > > \begin{equation}
> > > \sup\_{z}\ \mathbb{E}\_{A^T}\big[\ell(A\_{S\_\tau}^T;z)-\ell(A\_{S\_\tau'}^T;z)\big]\le \epsilon(A^T,\tau).
> > > \end{equation}
> > >
> > >
> > > Specifically, $\epsilon(A^{T},\tau)$-uniform stability quantifies the change in the final model's performance on a task caused by a slight perturbation of that task's dataset.
> > >
> > >
> > > **Theorem**[Theorem 2.2 in [1]] If $A^T$ is $\epsilon(A^T,\tau)$-uniformly stable, then
> > > \begin{equation}
> > > \big|\mathbb{E}\_{S\_\tau,A^T}\big[L\_{\mathcal{D}\_\tau}(A\_{S\_\tau}^T)-L\_{S\_\tau}(A\_{S\_\tau}^T)\big]\big|
> > > \le \epsilon(A^T,\tau).
> > > \end{equation}
> > >
> > > Therefore bounding the uniform stability immediately yields a bound on the corresponding generalization gap.
> > >
> > > (2) Control uniform stability through model discrepancy. If we fix an example $z$ and apply the Lipschitz condition on $\ell(·;z)$ to get
> > > \begin{equation*}\mathbb{E}|\ell(w\_{T,m};z) - \ell(w\_{T,m}';z)|\leq L\mathbb{E}[\delta\_{T,m}],\end{equation*}
> > > where $\delta\_{T,m}=\Vert w\_{T,m} - w\_{T,m}'\Vert$, and $w\_{T,m}$ and $w\_{T,m}'$ denote the models at training step $m$ on task $T$ obtained by running the stochastic algorithm on  $S$ and $S'$, respectively. Therefore uniform stability can be reduced to controlling the expected model discrepancy $\mathbb{E}[\delta\_{T,m}]$.
> > >
> > > (3) Recursively bound $\mathbb{E}[\delta\_{T,m}]$. We consider both convex and nonconvex losses under the standard assumptions that the loss is $\beta$-smooth and $L$-Lipschitz. By Lemma 3.7 in [1], the single-step update map is $1$-expansive for convex losses and $(1+\alpha\beta)$-expansive for nonconvex losses. Therefore, we propagate the discrepancy iteratively along the training trajectory and thus reduce $\mathbb{E}[\delta\_{T,m}]$ to a function of the initial discrepancy $\mathbb{E}[\delta\_{0,0}]$. Consequently, the resulting bound is governed by the Lipschitz constant $L$, the smoothness constant $\beta$, the learning rate $\alpha$, the number of tasks $T$, the sample size $n$ for each task, and the iteration count $m$ within each task.
> > >
> > > [1] Moritz Hardt, et al. Train faster, generalize better: Stability of stochastic gradient descent. ICML,  2016.

---

### Official Review · Reviewer_5JLu · 2025-11-02

**Soundness:** 3
**Presentation:** 3
**Contribution:** 3
**Rating:** 8
**Confidence:** 4

**Summary:**

This paper addresses the theoretical properties of CL in the NTK regime in the intermediate training stage, and give two insights on the role of Lipschitz and cross-task kernel on CL performance.
With these insights, they give two refined versions of the OGD algorithm.
These two algorithms are verified on standard CL datasets.

**Strengths:**

This is a good paper in both theoretical and practical aspects.

On the theoretical side, intermediate-stage CL performance analysis is novel to the theoretical CL literature. Also, the Lipschitz observation is novel to the CL field. The cross-task kernel result is valid and kind of expected.

On the practical side, with the two theoretical messages, the refined algorithms refined a well-used CL algorithm on these aspects and improved its performance on standard basic CL datasets.

The logic is clear. The mathematical tools used are standard.

I am pushing this paper for acceptance.

**Weaknesses:**

I do not see major weaknesses in this paper.

**Questions:**

How do you think the two theoretical messages can benefit other CL algorithms?

---

> ### Author Response · Authors · 2025-11-20
>
> Thank you for your thorough review and constructive comments. We provide detailed responses to each point below and have made major revisions to the manuscript. Revised text are highlighted in blue for ease of identification.
>
> **Q1.** How do you think the two theoretical messages can benefit other CL algorithms?
>
> **Response：** We thank the reviewer for the positive assessment of our work. To verify that our theoretical findings can benefit other CL algorithms, we integrate GAM or OPGD into both rehearsal based and regularization based methods on the Permuted MNIST dataset. Concretely, for rehearsal based baselines we adopt a naive rehearsal protocol that randomly selects 100 samples from each past task. For regularization based baselines we use an $\ell_2$ regularizer so that the optimization problem for task $\tau$ takes the form $\arg\min\_{\theta\_\tau}\ \ell(\theta\_\tau) + ||\theta\_\tau - \theta\_{\tau-1}||\_2^2$. The corresponding results are presented in the following table. The experimental results show that our theoretical messages also benefit other CL algorithms.
>
> |Method|ACC|BWT|
> |-|-|-|
> |rehearsal| 82.82 ± 0.78| -12.60 ± 0.36|
> |rehearsal + GAM| 86.40 ± 0.42| -7.64 ± 0.44|
> |rehearsal + OPGD| **88.23 ± 0.16**| **-5.62 ± 0.13**|
> |$\ell_2$| 73.30 ± 1.21| -21.91 ± 1.22|
> |$\ell_2$ + GAM| 74.15 ± 1.33| -19.19 ± 1.43|
> |$\ell_2$ + OPGD| **86.45 ± 0.27**| **-7.42 ± 0.33**|

---

### Author Response · Authors · 2025-11-22
**Summary**

**We thank all the reviewers for their thoughtful suggestions on our submission and appreciate their positive impressions, including:**

- **Theoretical contributions.** Reviewers praised our intermediate-stage continual learning analysis as “novel” (5JLu), “crucial for understanding real-world continual learning scenarios” (Q8vv), and “a solid theoretical foundation” (iWjj). Reviewer 5JLu also commended the work for its “clear logic and standard mathematical tools.”
- **Empirical validation.** Reviewers 5JLu and Q8vv noted that our refined algorithms improve a well-used continual learning baseline on standard benchmarks. Reviewer Q8vv described our approach as “a well-founded strategy for continual learning models.” Reviewer iWjj observed that “the figures and tables effectively present the results in a clear and intuitive manner.”
- **Writing and presentation.** Reviewers highlighted the clarity and organization of the manuscript: “clear presentation” (Q8vv), “the paper is comprehensive, well written and well structured” (dpae), and “a clear framework and thorough overall exposition” (iWjj).

Below, **we summarize the main concerns and how we addressed them, for conciseness and easy of access.**

### 1. Additional experiments to verify theory

- **Reviewer 5JLu raised the concern of whether our two theoretical messages generalize to other CL algorithms.** We conducted experiments on Permuted MNIST by combining GAM and OPGD with naive rehearsal and $\ell_2$-regularized CL. The results show that both theoretical insights indeed benefit these additional CL methods.
- **Reviewer Q8vv and Reviewer iWjj suggested extending our experiments to more CL scenarios.** We added online-setting experiments for five baselines: SGD, GAM, OGD, OGD+, and OPGD. We evaluated on Permuted MNIST, Rotated MNIST, and Split CIFAR-100. Across all datasets, OGD+ consistently outperforms OGD in mitigating forgetting and improving generalization, while OPGD achieves the best overall performance.

### 2. Additional theoretical results

- **Reviewer dpae suggested including formal theorems that establish upper bounds for the forgetting and generalization error of PGN in CL.** We have incorporated formal theorems for PGN, a proof sketch, and full technical proofs in Appendix F of the revised manuscript.
- **Reviewer Q8vv asked for suggestions on building a more general theoretical analysis beyond the NTK regime.** We propose using stability-based techniques, which rely on the update rules and do not require strong architectural assumptions. We also provide a proof sketch illustrating how this approach can be used to analyze forgetting and generalization.

### 3. Additional explanation

- **Following Reviewer dpae’s suggestion**, we expanded the Related Work section to discuss theoretical CL literature in the regularized setting and refined the description of the CL protocol, clarifying the transition and initialization process from previous to new tasks.
- **Reviewer dpae questioned the contribution of our theoretical results.** We are the first to provide upper bounds on both forgetting and generalization error at intermediate training stages, and the first to theoretically show that reducing the Lipschitz constant mitigates both forgetting and generalization error. These advances highlight the significance of our theoretical contribution.
- **Reviewer Q8vv and Reviewer iWjj raised concerns about computational and memory costs.** We address this by storing only a small batch of samples from previous tasks, instead of storing all past gradients. In addition, recent work [1] proposes O-LoRA, which learns task-specific low-rank subspaces that remain approximately orthogonal across tasks; this method further reduces memory and computational overhead for transformer architectures.
- **Reviewer iWjj: If the model deviates from the NTK assumption (e.g., CNNs), have different behaviors been observed?** Our experiments with CNN show that the empirical results align with our theoretical predictions. Since the NTK regime only assumes sufficient width, the resulting conclusions remain broadly applicable in practice.

[1] Wang, Xiao, et al. Orthogonal subspace learning for language model continual learning. EMNLP, 2023.

---

### Meta-Review · Area_Chair_qypz · 2026-01-07

**Summary:**

This work formulates CL in the NTK regime and provides a novel generalization bound that allows intermediate (early-stopped) training in each task. The analysis leads to the proposal of OPGD, and the authors empirically verify its usefulness.
As some reviewers commented, this work is well balanced between theory and practice, and it will enrich our understanding of intermediate CL dynamics, whose theoretical foundation  is still under construction,
One reviewer gave a rejection-side evaluation, but the issues raised were minor and seem to have been properly addressed in the rebuttal phase. Thus, I evaluate this work as an acceptance.

**Reviewer Concerns:**

*Reviewer dpae is the only reviewer on the rejection side, but all of their concerns seem to be clearly resolved by the rebuttal. In particular,

>paper did not specifically provide the forgetting and generalization error of PGN, only lemma 2 and lemma 3 provided, which is not enough

The authors have added the corresponding theorems for PGN in the revised manuscript.

 *Reviewers Q8vv and iWjj suggested additional experiments, and this was addressed by adding an online CL setup on Split CIFAR-100.

**Reviewer Scores:**

Because of the resolved concerns as mentioned above, I expect that

*Reviewer dpae would increase their score to the accept side, for instance, to 6.

*Reviewer Q8vv or iWjj would increase their score to 8.

---

### Decision · Program_Chairs · 2026-01-26

Accept (Poster)